# Riemannian Networks over Full-Rank Correlation Matrices

**Ziheng Chen** [1 2]   **Xiaojun Wu** [3]   **Bernhard Schölkopf** [2]   **Nicu Sebe** [1]

## Abstract

Representations on the Symmetric Positive Definite (SPD) manifold have garnered significant attention across different applications. In contrast, the manifold of full-rank correlation matrices, a normalized alternative to SPD matrices, remains largely underexplored. This paper introduces Riemannian networks over the correlation manifold, leveraging five recently developed correlation geometries. We systematically extend basic layers, including Multinomial Logistic Regression (MLR), Fully Connected (FC), and convolutional layers, to these geometries. Besides, we present methods for accurate backpropagation for two correlation geometries. Experiments comparing our approach against existing SPD and Grassmannian networks demonstrate its effectiveness.

## 1. Introduction

Covariance matrices in the Symmetric Positive Definite (SPD) manifold have achieved success in various applications, with many deep network architectures adapted to leverage their Riemannian geometries (Huang & Van Gool, 2017; Brooks et al., 2019; Chakraborty et al., 2020; Cruceru et al., 2021; Pan et al., 2022; Kobler et al., 2022; Wang et al., 2023; Chen et al., 2023; Katsman et al., 2024; Li et al., 2025; Pouliquen et al., 2025; Wang et al., 2025; Kang et al., 2025; Hu et al., 2026). In contrast, correlation matrices, despite serving as statistically compact alternatives to covariance matrices (Archakov & Hansen, 2024), remain unexplored in deep learning.

Only recently have Riemannian structures been developed for correlation matrices. David & Gu (2019) identified full-rank correlation matrices as a quotient manifold of the SPD manifold, referred to as the correlation manifold. However, this quotient geometry does not guarantee uniqueness or closed forms of the Riemannian logarithm and Fréchet mean (Thanwerdas & Pennec, 2022b, Sec. 1.1). To close this gap, Thanwerdas & Pennec (2022b) proposed three theoretically and computationally convenient geometries: Euclidean–Cholesky Metric (ECM), Log-Euclidean–Cholesky Metric (LECM), and Poly-Hyperbolic-Cholesky Metric (PHCM). Thanwerdas (2024) further introduced two efficient permutation-invariant metrics: Off-Log Metric (OLM) and Log-Scaled Metric (LSM). These Riemannian structures provide promising foundations for extending Euclidean deep learning to the correlation manifold.

On the other hand, several fundamental layers in Euclidean deep learning, such as Multinomial Logistics Regression (MLR), Fully Connected (FC), and convolutional layers, have been extended to different manifolds by leveraging their rich Riemannian or algebraic structures (Huang & Van Gool, 2017; Huang et al., 2017; 2018; Ganea et al., 2018; Chakraborty et al., 2020; Chen et al., 2022; Shimizu et al., 2021; Bdeir et al., 2024; Chen et al., 2024c; Nguyen et al., 2024). For the SPD manifold, these layers have been extended into the SPD manifold based on bilinear mapping (Huang & Van Gool, 2017), weighted Fréchet mean (Chakraborty et al., 2020), gyrovector spaces (Nguyen & Yang, 2023; Nguyen et al., 2024), Riemannian geometry (Chen et al., 2024a;c), respectively.

Inspired by these advancements, we develop MLR, FC, and convolutional layers for correlation manifolds in a geometrically intrinsic manner. We begin by systematically introducing four types of correlation-based MLR, FC, and convolutional layers, corresponding to ECM, LECM, OLM, and LSM, respectively. Besides, we discuss backpropagation through Riemannian computations over the correlation manifold, with novel approaches for accurate backpropagation under OLM and LSM. As the above four metrics have zero curvature, our next focus is to build correlation layers under the geometry of non-zero curvature. We target PHCM, induced by the product of multiple hyperbolic spaces (Thanwerdas & Pennec, 2022b, Thm. 4.4). By adapting existing Poincaré-based hyperbolic MLR, FC, and convolutional layers designed for a single Poincaré ball (Ganea et al., 2018; Shimizu et al., 2021), we construct their counterparts on the correlation manifold. With these basic

[1]Department of Information Engineering and Computer Science, University of Trento, Trento, Italy [2]Max Planck Institute for Intelligent Systems, Tübingen, Germany [3]School of Artificial Intelligence and Computer Science, Jiangnan University, Wuxi, China. Correspondence to: Ziheng Chen <ziheng_ch@163.com>.

*Proceedings of the 43rd International Conference on Machine Learning*, Seoul, South Korea. PMLR 306, 2026. Copyright 2026 by the author(s).

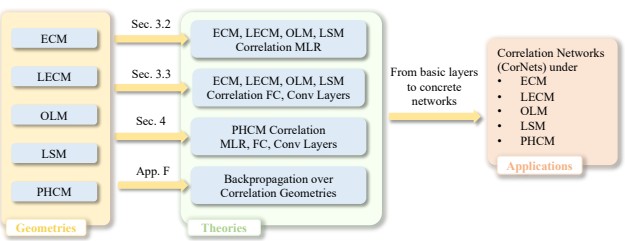

*Figure 1.* Overview of our theoretical derivation.

*Table 1.* Correspondence between Euclidean and correlation-based layers. For convolution, kernel-based FC refers to applying a convolution kernel to a receptive field, which is an FC transformation.

| Space | Euclidean $\mathbb{R}^n$ | Correlation $\mathrm{Cor}^+(n)$ |
|---|---|---|
| $c$-class MLR | $f : \mathbb{R}^n \ni x \mapsto p = \mathrm{Softmax}(Ax + b) \in \mathbb{R}^c$ | $f : \mathrm{Cor}^+(n) \ni C \mapsto p \in \mathbb{R}^c$ |
| FC layer | $\mathcal{F} : \mathbb{R}^n \ni x \mapsto y = Ax + b \in \mathbb{R}^m$ | $\mathcal{F} : \mathrm{Cor}^+(n) \ni C \mapsto Y \in \mathrm{Cor}^+(m)$ |
| Convolution | Kernel-based FC in each receptive field | Kernel-based correlation FC in each receptive field |
| Geometry | Euclidean | ECM, LECM, OLM, LSM and PHCM |

layers, we can construct Correlation Networks (CorNets) under different geometries. The effectiveness is validated by experiments comparing our approach against existing SPD and Grassmannian baselines.

Tab. 1 summarizes the correspondence between Euclidean and our correlation layers, and Fig. 1 illustrates the overview of our theoretical derivation. Due to page limits, all the proofs are presented in App. I. In summary, our **main contributions** are as follows:

1. We systematically extend MLR, FC, and convolutional layers to the correlation manifold under five geometries: four with zero curvature and one with non-zero curvature. The developed layers enable flexible variation of the latent geometry under a consistent network architecture, allowing for straightforward comparisons across different correlation geometries.
2. We develop accurate backpropagation of Riemannian computations under OLM and LSM.
3. We conduct experiments against existing SPD and Grassmannian networks to demonstrate the effectiveness of correlation embeddings and networks.

## 2. Full-Rank Correlation Geometries

**Notations.** For Euclidean spaces, we denote $\langle \cdot, \cdot \rangle$ as the standard inner product over $\mathbb{R}^n$ or $\mathbb{R}^{n \times n}$, with $\|\cdot\|$ as the induced norms, *i.e.*, $L_2$-norm for vectors and Frobenius norm for matrices. The zero vector and matrix are collectively denoted by $\mathbf{0}$. A Riemannian manifold $(\mathcal{M}, g)$ endowed with the Riemannian metric $g$ is abbreviated as $\mathcal{M}$. We denote $\mathrm{Log}_P$, $\mathrm{Exp}_P$, and $\langle \cdot, \cdot \rangle_P = g_P(\cdot, \cdot)$ as the Riemannian logarithm, exponentiation, and inner product at $P \in \mathcal{M}$, respectively. The parallel transport along the geodesic from $P \in \mathcal{M}$ to $Q \in \mathcal{M}$ is denoted by $\Gamma_{P \to Q}$, and the geodesic distance by $\mathrm{d}(\cdot, \cdot)$. App. A summarized our notations.

We briefly review five recently developed geometries on full-rank correlation matrices, with details pro-

vided in App. B. Given a covariance matrix $\Sigma$, its correlation matrix is defined as $C = \mathrm{Cor}(\Sigma) = \mathbb{D}(\Sigma)^{-1/2} \Sigma \mathbb{D}(\Sigma)^{-1/2}$, where $\mathbb{D}(\cdot)$ extracts the diagonal of $\Sigma$. The set of $n \times n$ full-rank correlation matrices, denoted $\mathrm{Cor}^+(n)$, forms a Riemannian manifold (David & Gu, 2019, Thm. 1). As illustrated in Fig. 2, each correlation corresponds to a surface in the SPD manifold. Recent advances introduced five convenient Riemannian metrics: Euclidean–Cholesky Metric (ECM), Log-Euclidean–Cholesky Metric (LECM), Poly-

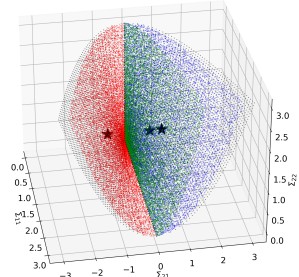

*Figure 2.* The black stars denote $2 \times 2$ correlation matrices, while the red, green, and blue dots denote corresponding SPD matrices. The black dots denote the boundary of SPD matrices.

Hyperbolic-Cholesky Metric (PHCM) (Thanwerdas & Pennec, 2022b), and the permutation-invariant Off-Log Metric (OLM) and Log-Scaled Metric (LSM) (Thanwerdas, 2024). All five are pullback metrics isometric to simpler prototype spaces: PHCM is derived from the product of hyperbolic spaces, while the other four are isometric to Euclidean spaces. We first review the associated prototype spaces before discussing each metric in detail.

- $\mathrm{LT}^1(n)$ ($\mathrm{LT}^0(n)$): Euclidean space of $n \times n$ lower triangular matrices with unit (null) diagonals.
- $\mathcal{L}^n$: Manifold of $n \times n$ lower triangular matrices with positive diagonals and unit row $L_2$-norm.
- $\mathrm{Hol}(n)$: Euclidean space of $n \times n$ symmetric matrices with null diagonals. The tangent space $T_C \mathrm{Cor}^+(n)$ at $C \in \mathrm{Cor}^+(n)$ can be identified with $\mathrm{Hol}(n)$.
- $\mathrm{Row}_0(n)$: Euclidean space of $n \times n$ symmetric matrices with null row sum.

**ECM** is derived from $\mathrm{LT}^1(n)$ by $\mathrm{Cor}^+(n) \xrightleftharpoons[\Theta^{-1}=\mathrm{Cor} \circ \mathrm{Chol}^{-1}]{\Theta = \mathbb{D}^{-1}(\mathrm{Chol}(\cdot)) \, \mathrm{Chol}(\cdot)} \mathrm{LT}^1(n)$, where $\Theta(C) = \mathbb{D}(\mathrm{Chol}(C))^{-1} \mathrm{Chol}(C)$ for any $C \in \mathrm{Cor}^+(n)$. Here, $\mathrm{Chol}(C)$ is the Cholesky decomposition $C = \mathrm{Chol}(C) \, \mathrm{Chol}(C)^\top$ and $\mathbb{D}(\cdot)$ returns a diagonal matrix consisting of the input diagonals. As $\mathrm{LT}^1(n) = I + \mathrm{LT}^0(n)$, ECM is essentially induced from the Euclidean space of $\mathrm{LT}^0(n)$.

**Proposition 2.1** (ECM). *Let* $\phi^{\mathrm{EC}}(C) = \lfloor \Theta(C) \rfloor$, *where* $\lfloor \cdot \rfloor$ *returns a strictly lower triangular matrix. ECM over* $\mathrm{Cor}^+(n)$ *is the pullback metric from the Euclidean space* $\mathrm{LT}^0(n)$ *by* $\phi^{\mathrm{EC}}$.

**LECM** is defined by further pulling back ECM: $\mathrm{Cor}^+(n) \xrightleftharpoons[(\log \circ \Theta)^{-1} = \mathrm{Cor} \circ \mathrm{Chol}^{-1} \circ \exp]{\log \circ \Theta} \mathrm{LT}^0(n)$, where

$\log(\cdot) : \mathrm{LT}^1(n) \to \mathrm{LT}^0(n)$ is the matrix logarithm with the matrix exponentiation $\exp(\cdot)$ as its inverse. Due to the nilpotency of $\mathrm{LT}^0(n)$, the matrix logarithm over $\mathrm{LT}^1(n)$ and exponentiation over $\mathrm{LT}^0(n)$ do not require eigendecomposition, as detailed in App. B.3.1.

**OLM** is derived from a permutation-invariant inner product over $\mathrm{Hol}(n)$ by $\mathrm{Cor}^+(n) \xrightleftharpoons[\mathrm{Exp}^\circ]{\mathrm{Log}^\circ = \mathrm{off} \circ \log} \mathrm{Hol}(n)$. For any symmetric hollow matrix $H \in \mathrm{Hol}(n)$, the operator $\mathcal{D}(H)$ returns a unique diagonal matrix, such that $\mathrm{Exp}^\circ(\cdot) :$ $\mathrm{Hol}(n) \ni H \mapsto \exp(\mathcal{D}(H) + H) \in \mathrm{Cor}^+(n)$ is a diffeomorphism. As shown by Archakov & Hansen (2021, Sec. 5), $\mathcal{D}(H)$ can be computed by the following exponentially converging algorithm: $D_{k+1} = D_k - \log(\mathbb{D}(\exp(D_k + H)))$, with $D_0 = \mathbf{0}$ as the zero matrix.

**LSM** is derived from a permutation-invariant inner product over $\mathrm{Row}_0(n)$ by $\mathrm{Cor}^+(n) \xrightleftharpoons[\mathrm{Exp}^\star = \mathrm{Cor} \circ \exp]{\mathrm{Log}^\star} \mathrm{Row}_0(n)$. For any correlation matrix $C \in \mathrm{Cor}^+(n)$, there exists a unique positive diagonal matrix $\mathcal{D}^\star(C)$ such that $\mathrm{Log}^\star(\cdot) :$ $\mathrm{Cor}^+(n) \ni C \mapsto \log(\mathcal{D}^\star(C)C\mathcal{D}^\star(C)) \in \mathrm{Row}_0(n)$ is a diffeomorphism. As shown by Thanwerdas (2024, Sec. 3.5), $\mathcal{D}^\star(C)$ corresponds to the unique zero of $f : x \in$ $\mathbb{R}^n_+ \longmapsto Cx - \frac{1}{x}$, with $\mathbb{R}^n_+$ as the $n$-dimensional positive vectors and $\frac{1}{x} = \left(\frac{1}{x_1}, \ldots, \frac{1}{x_n}\right)$. This could be solved by damped Newton's method.

**PHCM** is defined by the product of hyperbolic open hemispheres via Cholesky decomposition. Denoting $L = \mathrm{Chol}(C)$ for any correlation matrix $C \in \mathrm{Cor}^+(n)$, the $k$-th row of $L$ is $(L_{k1}, \ldots, L_{k,k-1}, L_{kk}, 0, \ldots, 0)$ with $L_{kk} > 0$, which belongs to the hyperbolic space of open hemisphere $\mathrm{HS}^{k-1} = \left\{x \in \mathbb{R}^k \mid \|x\| = 1 \text{ and } x_k > 0\right\}$. Therefore, $\mathcal{L}^n$ is identified with the product of $n-1$ open hemispheres, denoted as $\mathbb{PHS}^{n-1} = \prod_{i=1}^{n-1} \mathrm{HS}^i$. Here, since $L_{11} = 1$ and $\mathrm{HS}^0 = \{1\}$ are trivial, they are omitted from the product. PHCM is then defined by the pullback of the Cholesky decomposition from $\mathbb{PHS}^{n-1}$.

The Riemannian operators under all five metrics, including the Riemannian logarithm, exponentiation, geodesic, and parallel transport, have closed-form expressions, which are reviewed in App. B. Except for $\mathcal{D}$ and $\mathcal{D}^\star$, all computations involved can be backpropagated by existing techniques. Although the gradients of $\mathcal{D}$ and $\mathcal{D}^\star$ can be approximately backpropagated by PyTorch's autograd through their iterative algorithms, we propose accurate alternatives in App. E. Besides, the Euclidean inner products in the prototype spaces of ECM, LECM, LSM, and OLM are assumed to be standard. For $\mathrm{Cor}^+(n)$ with $n \le 3$, the invariance of OLM and LSM is nuanced and discussed in Rmk. B.5 and App. B.3.2. However, this paper focuses on $n > 3$.

# 3. Log-Euclidean Correlation Layers

Since ECM, LECM, OLM, and LSM are derived via diffeomorphisms from Euclidean spaces, they are collectively termed Log-Euclidean metrics (Thanwerdas, 2024). This motivates the principled development of Multinomial Logistics Regression (MLR), Fully Connected (FC), and convolutional layers under these four geometries. We begin by briefly revisiting the reformulation of MLR, followed by the introduction of correlation-based MLRs, FC layers, and convolutional layers.

## 3.1. Revisiting Multinomial Logistic Regression

As shown by Lebanon & Lafferty (2004, Sec. 5), the Euclidean MLR, $\mathrm{Softmax}(Ax + b)$, which computes the multinomial probability of each class $k \in \{1, \ldots, C\}$ for the input feature vector $x \in \mathbb{R}^n$, can be reformulated as the distances from $x$ to the margin hyperplanes describing the classes:

$$p(y = k \mid x) \propto \exp\left(v_k(x)\right),$$
$$v_k(x) = \mathrm{sign}(\langle a_k, x - p_k \rangle)\|a_k\| \, \mathrm{d}(x, H_{a_k, p_k}), \quad (1)$$
$$H_{a_k, p_k} = \{x \in \mathbb{R}^n \mid \langle a_k, x - p_k \rangle = 0\},$$

where $a_k, p_k \in \mathbb{R}^n$, and $H_{a_k, p_k}$ is a margin hyperplane, with $\mathrm{d}(x, H_{a_k, p_k})$ as the margin distance to the hyperplane. Recently, Chen et al. (2024c) generalized this formulation to general manifolds. Given an $m$-dimensional manifold $\mathcal{M}$, the MLR is defined as

$$p(y = k \mid X) \propto \exp\left(v_k(X)\right), \quad (2)$$
$$v_k(X) = \mathrm{sign}(\langle A_k, \mathrm{Log}_{P_k}(X) \rangle_{P_k})\|A_k\|_{P_k} \, \mathrm{d}(X, H_{A_k, P_k}), \quad (3)$$
$$\mathrm{d}(X, H_{A_k, P_k}) = \inf_{Q \in H_{A_k, P_k}} \mathrm{d}(X, Q), \quad (4)$$
$$H_{A_k, P_k} = \{X \in \mathcal{M} \mid \left\langle \mathrm{Log}_{P_k}(X), A_k \right\rangle_{P_k} = 0\}, \quad (5)$$

where $X \in \mathcal{M}$ is the manifold-valued input and $H_{A_k, P_k}$ is a Riemannian hyperplane, while $P_k \in \mathcal{M}$ and $A_k \in T_{P_k}\mathcal{M}$ for $1 \le k \le C$ are parameters. The key challenge is the optimization problem in Eq. (4). To circumvent this problem, Chen et al. (2024c, Sec. 3.2) relaxed it via Riemannian trigonometry. Unlike their method, this paper directly solves Eq. (4) to more faithfully respect different correlation geometries. In addition, to avoid over-parameterization (Shimizu et al., 2021, Sec. 3.1), we set $P_k = \mathrm{Exp}_E(\gamma_k[Z_k])$ and $A_k = \Gamma_{E \to P_k}(Z_k)$, with $[Z_k] = \frac{Z_k}{\|Z_k\|_E}$ as the unit direction vector of $Z_k$. Here, $E$ is the origin[1] of $\mathcal{M}$, while $\gamma_k \in \mathbb{R}$ and $Z_k \in T_E\mathcal{M} \cong \mathbb{R}^m$ are the MLR parameters. Under this trivialization, each hyperplane $H_{A_k, P_k}$ is denoted as $H_{Z_k, \gamma_k}$. We adopt from Lezcano Casado (2019) the term trivialization, which refers to optimizing manifold-valued parameters via the exponential map. App. C.1 presents a more detailed review of MLR.

---

[1]For the correlation, we define the identity matrix as the origin and will explain the reason later.

## 3.2. Log-Euclidean Correlation MLRs

As all Log-Euclidean metrics are isometric to the Euclidean ones, we can solve the associated MLRs defined by Eqs. (2) and (4) in a principled manner.

**Theorem 3.1.** [↓] *Given an $m$-dimensional manifold $\left(\mathcal{M}, g^{\mathcal{M}}\right)$ isometric to the standard Euclidean space $\mathbb{R}^m$ by the diffeomorphism $\phi : \mathcal{M} \to \mathbb{R}^m$. Denoting $E = \phi^{-1}(\mathbf{0})$ with $\mathbf{0}$ as the zero vector, each $v_k(X)$ and margin hyperplane $H_{Z_k,\gamma_k}$ in the $C$-class Riemannian MLR are $v_k(X) = \langle \phi(X), \phi_{*,E}(Z_k) \rangle - \gamma_k \|\phi_{*,E}(Z_k)\|$ and $H_{Z_k,\gamma_k} = \{X \in \mathcal{M} \mid v_k(X) = 0\}$, respectively. Here, $Z_k \in T_E \mathcal{M} \cong \mathbb{R}^m$ and $\gamma_k \in \mathbb{R}$ for $1 \le k \le C$ are MLR parameters, while $\phi_*$ is the differential.*

Simple computations show that

$$\begin{aligned} \text{ECM: } & \phi^{\text{EC}}(I) = \mathbf{0}, && \text{LECM: } \log \circ \Theta(I) = \mathbf{0}, \\ \text{OLM: } & \text{Log}^\circ(I) = \mathbf{0}, && \text{LSM: } \text{Log}^\star(I) = \mathbf{0}. \end{aligned} \quad (6)$$

Therefore, we define the origin of the correlation manifold under four Log-Euclidean metrics as the identity matrix. Besides, Thm. 3.1 suggests that Log-Euclidean MLRs can be obtained modulo the calculation of diffeomorphisms and their differentials at the identity matrix $I$.

**Proposition 3.2** (Differentials). [↓] *For any tangent vector $V \in T_I \text{Cor}^+(n) \cong \text{Hol}(n)$, the differentials of $\phi^{\text{EC}}$, $\log \circ \Theta$, $\text{Log}^\circ$, and $\text{Log}^\star$ at the identity matrix $I$ are*

$$\begin{aligned} \phi^{\text{EC}}_{*,I}(V) &= \lfloor V \rfloor, & (\log \circ \Theta)_{*,I}(V) &= \lfloor V \rfloor, \\ \text{Log}^\circ_{*,I}(V) &= V, & \text{Log}^\star_{*,I}(V) &= V - \text{diag}(V\mathbf{1}), \end{aligned} \quad (7)$$

*where $\text{diag} : \mathbb{R}^n \to \text{Diag}(n)$ returns a diagonal matrix, and $\mathbf{1} = (1, \cdots, 1)^\top \in \mathbb{R}^n$.*

Putting Prop. 3.2 into Thm. 3.1, we obtain correlation MLRs under four Log-Euclidean metrics.

**Theorem 3.3** (Log-Euclidean MLRs). *Given $C \in \text{Cor}^+(n)$, the logit $v_k(C)$ for the $k$-th class in the correlation MLRs under four Log-Euclidean metrics are*

$$\begin{aligned} v_k^{\text{LE}}(C) &= \langle \lfloor \Theta(C) \rfloor, \lfloor Z_k \rfloor \rangle - \gamma_k \|\lfloor Z_k \rfloor\|, \\ v_k^{\text{LEC}}(C) &= \langle \log \circ \Theta(C), \lfloor Z_k \rfloor \rangle - \gamma_k \|\lfloor Z_k \rfloor\|, \\ v_k^{\text{OL}}(C) &= \langle \text{Log}^\circ(C), Z_k \rangle - \gamma_k \|Z_k\|, \\ v_k^{\text{LS}}(C) &= \langle \text{Log}^\star(C), \text{Log}^\star_{*,I}(Z_k) \rangle - \gamma_k \|\text{Log}^\star_{*,I}(Z_k)\|, \end{aligned} \quad (8)$$

*where $Z_k \in \text{Hol}(n)$ and $\gamma_k \in \mathbb{R}$ are parameters.*

## 3.3. Log-Euclidean FC and Convolutional Layers

In order to build correlation FC layers, we first reformulate the Euclidean FC layer. The Euclidean FC layer is defined as $y = Ax + b$ with $A \in \mathbb{R}^{m \times n}$ and $b \in \mathbb{R}^m$. It can be expressed element-wise as $y_k = v_k(x) = \langle a_k, x - p_k \rangle$ with $a_k, p_k \in \mathbb{R}^n$ and $\langle p_k, a_k \rangle = b_k$. Shimizu et al. (2021, Sec.

3.2) reformulated the Euclidean FC layer as an operation that transforms the input $x \in \mathbb{R}^n$ by $v_k(x)$ in the Euclidean MLR and treats the $k$-th output coordinate $y_k$ as the signed distance from the hyperplane containing the origin and orthogonal to the $k$-th axis of the output space $\mathbb{R}^m$. Based on this, they proposed the Poincaré FC layer between Poincaré balls. We generalize this reformulation to the correlation.

**Definition 3.4** (Correlation FC layers). Given a metric $g$, the correlation FC layer $\mathcal{F} : \text{Cor}^+(n) \ni X \mapsto Y \in \text{Cor}^+(m)$ returns the output $Y$ by solving the following $d = m(m-1)/2$ equations:

$$s_k \, \text{d}(Y, H_{O_k, I}) = v_k(X; Z_k, \gamma_k), \quad 1 \le k \le d, \quad (9)$$

where $s_k = \text{sign}\left(\langle \text{Log}_I(Y), O_k \rangle_I\right)$, $I$ is the identity matrix, $d$ is the dimension of $\text{Cor}^+(m)$, $\{O_k\}_{k=1}^d$ is an orthonormal basis over $T_I \text{Cor}^+(m)$, $\text{d}(\cdot, \cdot)$ is the margin distance to the hyperplane $H_{O_k, I}$, and $v_k^{\mathcal{N}}$ is defined by Eq. (2) for $\text{Cor}^+(n)$. The FC parameters are $\{Z_k \in \text{Hol}(n)\}_{k=1}^d$ and $\{\gamma_k \in \mathbb{R}\}_{k=1}^d$.

App. D.1 details how Def. 3.4 extends the existing SPD, Poincaré, and Euclidean FC layers.

Although Def. 3.4 is implicitly defined by $d$ equations, the FC layers under four Log-Euclidean geometries admit explicit expressions in a principled manner. Analogous to Thm. 3.1, a corresponding result for the FC layer is presented in Lem. I.2, which brings Log-Euclidean FC layers.

**Theorem 3.5** (Log-Euclidean FC layers). [↓] *Given an input correlation $C \in \text{Cor}^+(n)$, the correlation FC layers $\mathcal{F}(\cdot) : \text{Cor}^+(n) \to \text{Cor}^+(m)$ under different Log-Euclidean metrics are*

$$\text{ECM: } Y = \text{Cor} \circ \text{Chol}^{-1}\left(V^{\text{EC}} + I_m\right), \quad (10)$$

$$\text{LECM: } Y = \text{Cor} \circ \text{Chol}^{-1} \circ \exp\left(V^{\text{LEC}}\right), \quad (11)$$

$$\text{OLM: } Y = \text{Exp}^\circ\left(V^{\text{OL}}\right), \quad (12)$$

$$\text{LSM: } Y = \text{Cor} \circ \exp\left(V^{\text{LS}}\right), \quad (13)$$

*where the $(i,j)$-th elements in $V^{\text{EC}} \in \text{LT}^0(m)$, $V^{\text{LEC}} \in \text{LT}^0(m)$, $V^{\text{OL}} \in \text{Hol}(m)$, and $V^{\text{LS}} \in \text{Row}_0(m)$ are*

$$V_{ij}^{\text{EC}} = \begin{cases} v_{ij}^{\text{EC}}(C), & \text{if } i > j \\ 0, & \text{otherwise} \end{cases}$$

$$V_{ij}^{\text{LEC}} = \begin{cases} v_{ij}^{\text{LEC}}(C), & \text{if } i > j \\ 0, & \text{otherwise} \end{cases}$$

$$V_{ij}^{\text{OL}} = \begin{cases} \frac{v_{ij}^{\text{OL}}(C)}{\sqrt{2}}, & \text{if } i > j \\ V_{ji}^{\text{OL}}, & \text{if } i < j \\ 0, & \text{otherwise} \end{cases}$$

$$V_{ij}^{\text{LS}} = \begin{cases} v_{ij}^{\text{LS}}(C)/\sqrt{6}, & \text{if } m > i > j \ge 1 \\ v_{ii}^{\text{LS}}(C)/\sqrt{3}, & \text{if } m > i \ge 1 \\ V_{ji}^{\text{LS}}, & \text{if } i < j \\ -\sum_{k=1}^{m-1} V_{kj}^{\text{LS}}, & \text{if } i = m, 1 \le j < m \\ \sum_{k=1}^{m-1} \sum_{l=1}^{m-1} V_{lk}^{\text{LS}}, & \text{if } i = j = m \end{cases}$$

*Each $v_{ij}^g$ with $g \in \{\text{EC, LEC, OL, LS}\}$ is defined by Eq. (8) with parameters of $Z_{ij} \in \text{Hol}(n)$ and $\gamma_{ij} \in \mathbb{R}$. Each $(i,j)$ index is defined as: For $v_{ij}^{EC}$, $v_{ij}^{LEC}$, and $v_{ij}^{OL}$, the indices are $i, j = 1, \cdots, m$ and $i > j$; For $v_{ij}^{LS}$, the indices are $i, j = 1, \cdots, m-1$ and $i \geq j$.*

**Euclidean convolution.** As shown by Shimizu et al. (2021, Sec. 3.4), the Euclidean convolution takes the FC transformation on each receptive field. Given a $c$-channel concatenated feature vector $x \in (\mathbb{R}^n)^c$ in a receptive field, the $k$-th output of this receptive field can be described as an affine transformation, $y_k = \langle a_k, x \rangle - b_k$. Therefore, the correlation convolution can be defined by the correlation FC layer within each receptive field.

**Correlation convolution.** The $c$-channel correlation matrices $\{C_i \in \text{Cor}^+(n)\}_{i=1}^c$ within a receptive field are first concatenated into $\boldsymbol{C} \in (\text{Cor}^+(n))^c$. For each convolution kernel, $\boldsymbol{C}$ is then fed into a correlation FC layer.[2] Fig. 3 illustrates the above process.

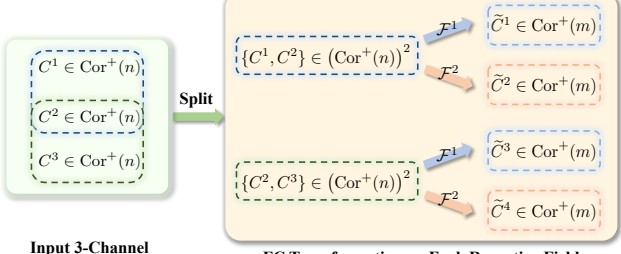

**Input 3-Channel Correlation Matrices**  **FC Transformation on Each Receptive Field**

*Figure 3.* Illustration of the Log-Euclidean 1D convolution with two kernels. The 3-channel input is first split into two receptive fields along the channel dimension. In each receptive field, two kernels are applied to the product space.

# 4. Poly-Hyperbolic-Cholesky Layers

As discussed in Sec. 2, the space $\mathcal{L}^n$, consisting of the Cholesky factors of $\text{Cor}^+(n)$, can be identified with the product of $n-1$ hyperbolic open hemispheres, $\mathbb{PHS}^{n-1} = \prod_{i=1}^{n-1} \text{HS}^i$. As shown by Cannon et al. (1997, Sec. 7), there are five isometric models over the hyperbolic space. A widely used model is the Poincaré ball $\mathbb{P}_K^n = \left\{ x \in \mathbb{R}^n \mid \|x\|^2 < -1/K \right\}$, where the MLR, FC, and convolutional layers have already been well studied (Ganea et al., 2018; Shimizu et al., 2021). In the following, we focus on the canonical Poincaré ball ($K = -1$), denoted as $\mathbb{P}^n$. We first identify the correlation manifold with the poly-Poincaré space $\mathbb{PP}^{n-1} = \prod_{i=1}^{n-1} \mathbb{P}^i$, the product of $n-1$ Poincaré balls. Then, we develop correlation layers from the layers on a single Poincaré space.

---

[2]Thm. 3.5 naturally supports product geometries, which are detailed in App. D.2.

## 4.1. Correlation Geometry via Poincaré Balls

**Proposition 4.1** (Isometries). [↓] *The open hemisphere $\text{HS}^n$ is isometric to the Poincaré ball $\mathbb{P}^n$ by $\psi_{\text{HS}^n \to \mathbb{P}^n}((x^\top, x_{n+1})^\top) = \frac{x}{1+x_{n+1}}$, and $\psi_{\mathbb{P}^n \to \text{HS}^n}(y) = \frac{1}{1+\|y\|^2} \begin{pmatrix} 2y \\ 1 - \|y\|^2 \end{pmatrix}$, with $(x^\top, x_{n+1})^\top \in \text{HS}^n \subset \mathbb{R}^n \times \mathbb{R}^+$ and $y \in \mathbb{P}^n \subset \mathbb{R}^n$.*

Prop. 4.1 indicates that $\text{Cor}^+(n)$ can be identified with $\mathbb{PP}^{n-1} = \prod_{i=1}^{n-1} \mathbb{P}^i$ via the diffeomorphism $\Psi \circ \text{Chol}$:

$$C \xmapsto{\text{Chol}} \begin{pmatrix} 1 & 0 & \cdots & 0 \\ L_{21} & L_{22} & \cdots & 0 \\ \vdots & \vdots & \ddots & \vdots \\ L_{n1} & L_{n2} & \cdots & L_{nn} \end{pmatrix} \xmapsto{\Psi} \begin{matrix} \Psi_1(h_1) \\ \vdots \\ \Psi_{n-1}(h_{n-1}) \end{matrix} \quad (14)$$

with $C \in \text{Cor}^+(n)$, $h_{i-1} = (L_{i1}, \cdots, L_{ii})^\top \in \text{HS}^{i-1}$, and $\Psi_i = \psi_{\text{HS}^i \to \mathbb{P}^i}$. This identification motivates us to construct the correlation layers using the corresponding layers over Poincaré spaces.

## 4.2. Revisiting Poincaré Layers

The Poincaré MLR (Ganea et al., 2018; Shimizu et al., 2021) and FC layers on Poincaré spaces (Shimizu et al., 2021) follow the same logic as Sec. 3.1 and Def. 3.4, respectively. Their closed-form expressions are reviewed in App. C.2.

The Poincaré convolutional layer shares a logic similar to the correlation convolution, except it uses $\beta$-concatenation to concatenate the Poincaré vectors in each receptive field (Shimizu et al., 2021, Secs. 3.3-3.4), which can stabilize the norm of the Poincaré vector. The Poincaré $\beta$-concatenation generalizes the Euclidean concatenation via the scaled concatenation in the tangent space. Given inputs $\{x_i \in \mathbb{P}^{n_i}\}_{i=1}^N$, it is defined as $\text{Exp}_{\mathbf{0}} \left( \beta_n \left( \beta_{n_1}^{-1} v_1^\top, \cdots, \beta_{n_N}^{-1} v_N^\top \right) \right)^\top \in \mathbb{P}^n$, where $v_i = \text{Log}_{\mathbf{0}}(x_i)$ and $n = \sum_{i=1}^N n_i$. Here, $\beta_{n_i}$ and $\beta_n$ are defined by the beta function $\beta_\alpha = \text{B}(\alpha/2, 1/2)$. The inverse is called the Poincaré $\beta$-split. The Poincaré convolution is: (1) $\beta$-concatenating the multi-channel feature in a given receptive field; and (2) performing the Poincaré FC transformation.

## 4.3. Building Poly-Hyperbolic-Cholesky Layers

**PHCM MLR.** The input multi-channel correlation matrices, $\boldsymbol{C} = \{C^i \in \text{Cor}^+(n)\}_{i=1}^c$, are first mapped into poly-Poincaré spaces as $\boldsymbol{x} = \{x^i = \Psi \circ \text{Chol}(C^i) \in \mathbb{PP}^{n-1}\}_{i=1}^c$. The resulting Poincaré vectors are then $\beta$-concatenated into a single Poincaré vector $x \in \mathbb{P}^N$, where $N = c\frac{n(n-1)}{2}$. This concatenated vector is subsequently fed into the Poincaré MLR for classification.

**PHCM convolutional and FC layer.** The convolutional layer follows a logic similar to Log-Euclidean convolution.

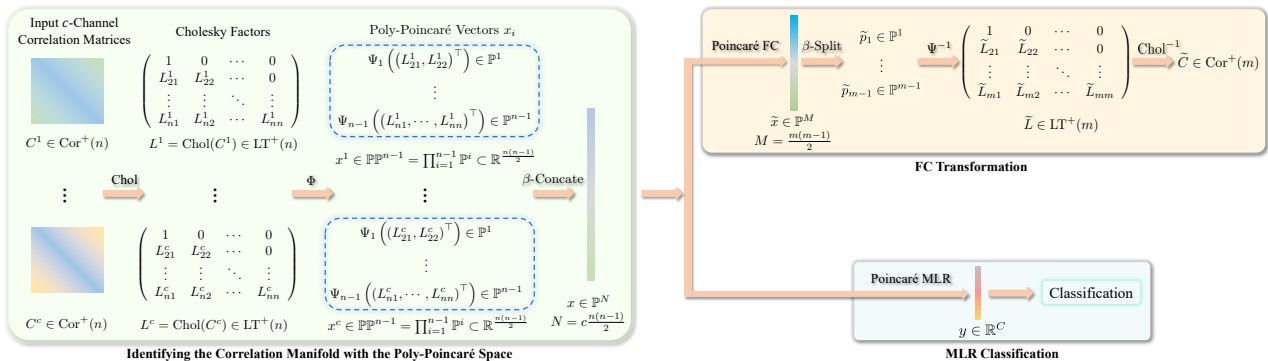

*Figure 4.* Illustration of the PHCM convolution and MLR. The multi-channel input correlation matrices are denoted as $\{C^i\}_{i=1}^c$. For the convolutional layer, the illustration focuses on the transformation within a receptive field and assumes a single-channel output.

The multi-channel correlation matrices within a receptive field $\boldsymbol{C} = \{C^i \in \mathrm{Cor}^+(n)\}_{i=1}^c$ are first mapped to a $\beta$-concatenated Poincaré vector $x \in \mathbb{P}^N$ as in the PHCM MLR, which is then fed into the Poincaré FC layer for dimensionality transformation. This produces a vector $\widetilde{x} \in \mathbb{P}^M$, with $M = k\frac{m(m-1)}{2}$, which is then split using $\beta$-split. Subsequently, applying $\mathrm{Chol}^{-1} \circ \Psi^{-1}$ reconstructs new $k \times m \times m$ correlation matrices. When the input is a single correlation matrix, it is reduced to the correlation FC.

Fig. 4 illustrates the PHCM layers. However, there is an underlying ambiguity in the above discussion. To clarify, we write each $x^i \in \mathbb{PP}^{n-1}$ in $\boldsymbol{x}$ as $x^i = \{p_1^i \in \mathbb{P}^1, \cdots, p_{n-1}^i \in \mathbb{P}^{n-1}\}$, which gives $\boldsymbol{x} = \{p_j^i \in \mathbb{P}^j\}_{i=1,j=1}^{i=c,j=n-1}$. We can either concatenate twice by $i \rightarrow j$ or once along both $i$ and $j$. A similar issue arises with $\beta$-split. We show order-invariance in App. F. Therefore, we always conduct the $\beta$-operation simultaneously along both $i$ and $j$.

## 5. Experiments

We construct Riemannian networks on the correlation manifold, termed CorNets, using the proposed convolutional and MLR layers. Following previous work (Huang & Van Gool, 2017; Brooks et al., 2019; Chen et al., 2024b), we evaluate our approach on the Radar dataset (Brooks et al., 2019) for radar signal classification, along with the HDM05 (Müller et al., 2007), FPHA (Garcia-Hernando et al., 2018) and NTU120 (Liu et al., 2019) datasets for human action recognition. More details are provided in App. H.

**Implementation.** We denote CorNet-Metric as the CorNet composed of correlation convolution and MLR layers under a specified metric. In line with Nguyen et al. (2024), each CorNet consists of one correlation convolutional layer followed by a correlation MLR layer, trained with cross-entropy loss. Following Wang et al. (2024); Nguyen et al. (2024), each raw feature is modeled as a multi-channel $[c, n, n]$ SPD tensor. Since matrix power effectively activates SPD matrices by deforming their geometry (Thanwer-

das & Pennec, 2022a; Chen et al., 2024b;c; 2025), we first apply a matrix power, and then convert the result to correlation matrices as the input of CorNet. Due to trivialization, all parameters lie in Euclidean space and are optimized by standard Euclidean optimizers. We compare CorNets against representative Grassmannian and SPD networks, including GrNet (Huang et al., 2018), GyroGr (Nguyen & Yang, 2023), GyroGr-Scaling (Nguyen & Yang, 2023), SPDNet (Huang & Van Gool, 2017), SPDNetBN (Brooks et al., 2019), RRes-Net (Katsman et al., 2024), LieBN (Chen et al., 2024b), SPD MLR (Chen et al., 2024c), Gyro (Nguyen & Yang, 2023), and GyroSPD++(Nguyen et al., 2024). Please refer to App. H.4 for more details.

**Main Results.** Tab. 2 reports the five-fold results comparing our CorNets against existing SPD and Grassmannian baselines. We summarize the key observations below. **(1) Effectiveness:** CorNets consistently outperform both SPD and Grassmannian networks. Specifically, CorNets surpass the classic SPDNet by *5.15%*, *17.69%*, *6.58%*, and *13.84%* on four datasets, respectively, and outperform the best Grassmannian networks by *7.76%*, *19.07%*, *6.86%*, and *7.45%*. Despite not using batch normalization or residual blocks, CorNets achieve superior performance compared to SPDNetBN, SPDNetLieBN, and RResNet. Notably, although CorNets share the same architecture as GyroSPD++ (one SPD convolutional layer followed by one SPD MLR), CorNets exhibit better performance. These results highlight the effectiveness of correlation embedding and our method for constructing correlation networks. **(2) Optimal metric:** The optimal metric for CorNets varies across datasets, indicating that the choice of geometry is a critical hyperparameter in Riemannian networks. Our framework enables seamless switching among five correlation geometries in a consistent architecture, demonstrating the adaptability of our approach to different tasks. **(3) Efficiency:** CorNets achieve efficiency comparable to or better than several baseline methods. The most efficient CorNet variant is based on ECM, owing to the simplest computations of ECM. Although GyroSPD++ uses the same architecture,

*Table 2.* Five-fold results and training time per epoch on four datasets. The top 3 results are highlighted with **red**, **blue**, and **cyan**. * denotes reproduced results due to missing official code.

| Manifold | Method | Radar Mean±STD | Time | HDM05 Mean±STD | Time | FPHA Mean±STD | Time | NTU120 Mean±STD | Time |
|---|---|---|---|---|---|---|---|---|---|
| Grassmann | GrNet (Huang et al., 2018) | 90.48 ± 0.76 | 1.39 | 63.19 ± 0.70 | 1.64 | 85.31 ± 0.90 | 0.70 | 57.59 ± 0.22 | 50.97 |
| | GyroGr* (Nguyen & Yang, 2023) | 90.64 ± 0.57 | 1.38 | 58.32 ± 1.23 | 2.48 | 79.62 ± 0.49 | 0.70 | 53.76 ± 0.18 | 136.96 |
| | GyroGr-Scaling* (Nguyen & Yang, 2023) | 88.88 ± 1.52 | 1.63 | 39.75 ± 0.93 | 3.52 | 58.62 ± 1.66 | 1.03 | 43.90 ± 0.23 | 338.01 |
| SPD | SPDNet (Huang & Van Gool, 2017) | 93.25 ± 1.10 | 0.66 | 64.57 ± 0.61 | 0.50 | 85.59 ± 0.72 | 0.28 | 51.25 ± 0.36 | 12.77 |
| | SPDNetBN (Brooks et al., 2019) | 94.85 ± 0.99 | 1.25 | 71.28 ± 0.79 | 0.94 | 89.33 ± 0.49 | 0.58 | 54.35 ± 0.43 | 19.78 |
| | SPDResNet-AIM (Katsman et al., 2024) | 95.71 ± 0.37 | 0.96 | 64.95 ± 0.82 | 1.23 | 86.63 ± 0.55 | 0.69 | 57.33 ± 0.35 | 23.84 |
| | SPDResNet-LEM (Katsman et al., 2024) | 95.89 ± 0.86 | 0.77 | 70.12 ± 2.45 | 0.55 | 85.07 ± 0.99 | 0.30 | 61.34 ± 2.02 | 13.00 |
| | SPDNetLieBN-AIM (Chen et al., 2024b) | 95.47 ± 0.90 | 1.21 | 71.83 ± 0.69 | 1.15 | 90.39 ± 0.66 | 0.97 | 58.20 ± 0.46 | 31.10 |
| | SPDNetLieBN-LCM (Chen et al., 2024b) | 94.80 ± 0.71 | 1.10 | 71.78 ± 0.44 | 1.11 | 86.33 ± 0.43 | 0.59 | 57.96 ± 0.43 | 22.06 |
| | SPDNetMLR (Chen et al., 2024c) | 94.59 ± 0.82 | 0.66 | 65.90 ± 0.93 | 5.46 | 86.60 ± 0.43 | 0.88 | 58.59 ± 0.13 | 22.48 |
| | GyroLE* (Nguyen & Yang, 2023) | 96.24 ± 0.24 | 0.79 | 73.17 ± 0.37 | 2.86 | 90.73 ± 0.92 | 1.59 | 59.29 ± 0.42 | 22.08 |
| | GyroLC* (Nguyen & Yang, 2023) | 93.60 ± 1.31 | 0.66 | 67.53 ± 0.85 | 1.49 | 76.10 ± 0.63 | 0.78 | 59.29 ± 0.42 | 14.14 |
| | GyroAI* (Nguyen & Yang, 2023) | 96.29 ± 0.48 | 0.99 | 72.34 ± 1.06 | 22.80 | 89.60 ± 0.37 | 12.62 | 62.21 ± 0.29 | 98.31 |
| | GyroSPD++* (Nguyen et al., 2024) | 95.20 ± 0.88 | 5.09 | 69.82 ± 1.79 | 103.57 | 89.50 ± 0.37 | 66.35 | 61.57 ± 0.30 | 216.46 |
| Correlation | CorNet-ECM | **97.71 ± 0.61** | 1.01 | 81.35 ± 1.27 | 0.60 | **92.17 ± 0.49** | 0.50 | **65.04 ± 0.14** | 12.06 |
| | CorNet-LECM | **98.40 ± 0.70** | 1.12 | 78.05 ± 1.14 | 0.64 | **91.17 ± 0.32** | 0.54 | **65.03 ± 0.10** | 12.68 |
| | CorNet-OLM | **97.57 ± 0.76** | 1.35 | **81.46 ± 0.61** | 0.93 | **91.63 ± 0.12** | 0.79 | **64.41 ± 0.23** | 16.07 |
| | CorNet-LSM | 96.24 ± 1.48 | 1.50 | 74.89 ± 1.07 | 0.98 | 83.43 ± 0.65 | 0.83 | 60.69 ± 0.85 | 16.28 |
| | CorNet-PHCM | 96.56 ± 0.86 | 2.37 | **82.26 ± 0.92** | 1.10 | 90.03 ± 0.63 | 0.77 | 60.01 ± 0.22 | 16.92 |

CorNets achieve significantly greater efficiency, attributed to the heavy computational cost of the AIM-based computations in GyroSPD++ and the lightweight Riemannian computations on the correlation manifold. Particularly, on the largest NTU120 datasets, CorNet-ECM and CorNet-LECM are the top two most efficient ones.

**Ablations on Mixed Geometries.** Our main experiments use the same metric for convolution and MLR. To evaluate mixed geometries, we assign different metrics to the two layers. Tab. 3 reports five-fold results on HDM05 and FPHA. Overall, consistent metrics yield the best accuracy.

**Visualization.** Fig. 5 shows that different metrics induce visibly distinct curved hyperplanes.

**Potential and Necessity.** Although correlation matrices are still SPD, naively treating them as SPD inputs and feeding them into existing SPD networks fails to leverage their intrinsic geometric structures. To illustrate this, we use the classic SPDNet (Huang & Van Gool, 2017) but replace its covariance inputs with correlation matrices. The five-fold average results in Tab. 4 reveal two key insights: (1) on the HDM05 dataset, correlation inputs lead to improved performance, suggesting that correlation embeddings can serve as compact and effective alternatives to covariance representations; and (2) on the other two datasets, the performance degrades, indicating that ignoring the specific geometry of correlation matrices can be detrimental. These findings highlight both the promise and the necessity of designing networks

*Table 4.* SPDNet: SPD vs. correlation.

| Input | Radar | HDM05 | FPHA |
|---|---|---|---|
| SPD | **93.25 ± 1.10** | 64.57 ± 0.61 | **85.59 ± 0.72** |
| Correlation | 89.49 ± 0.67 | **66.81 ± 0.73** | 83.37 ± 0.40 |

respecting the unique geometry of the correlation manifold.

**Ablations on Correlation Embeddings.** To further evaluate the effectiveness of correlation embeddings, we compare the performance of directly classifying raw covariance matrices using SPDMLR (Chen et al., 2024c, Thm. 4.2) with that of classifying corresponding raw correlation matrices using correlation MLR (CorMLR). The original SPDMLR involves an SPD matrix parameter for each class, which causes heavy Riemannian computations. For a fair comparison, we also implement a similar trivialization as Sec. 3.1, denoted as SPDMLR-Trivlz. We implement SPDMLR-Trivlz under LEM, LCM, and AIM, respectively. Tab. 5 presents the 5-fold average results on all three datasets. CorMLR performs better than SPDMLR-Trivlz on HDM05 and FPHA. Although CorMLR performs worse on Radar, we emphasize that these comparisons are conducted on a single MLR layer, which fails to fully uncover the potential of correlation matrices. Besides, SPDMLR under AIM is much slower than others, especially on HDM05, due to its complex computations. In contrast, CorMLR, especially under ECM and PHCM, offers competitive or superior efficiency.

**Covariance vs. Correlation.** Tabs. 2 and 5 show that correlation achieves relatively larger gains than SPD covariance on HDM05. As detailed in Apps. H.5.1 and H.5.2, covariance matrices on HDM05 exhibit large coefficients of variation for diagonal variances, and their diagonal magnitudes are much larger than the off-diagonal entries. In such cases, covariance matrices can introduce nuisance noise and make it harder for the model to exploit informative off-diagonal correlations. In contrast, correlation rebalances diagonal and off-diagonal contributions and encourages the network to focus on vibrant pairwise correlations. This behavior is

*Table 3.* Ablations on mixed geometries. Each row shows the metric used for Convolution (Conv), and each column is the metric for MLR. The  diagonal  entries indicate configurations where both layers use the same metric. The best result in each row is highlighted in **red**.

| Dataset | HDM05 | | | | | FPHA | | | | |
|---|---|---|---|---|---|---|---|---|---|---|
| MLR / Conv | ECM | LECM | OLM | LSM | PHCM | ECM | LECM | OLM | LSM | PHCM |
| ECM | **81.35 ± 1.27** | 73.38 ± 0.34 | 80.11 ± 0.77 | 78.54 ± 0.43 | 80.80 ± 0.54 | **92.17 ± 0.49** | 91.50 ± 0.21 | 91.67 ± 0.28 | 87.37 ± 1.14 | 91.97 ± 0.24 |
| LECM | 66.49 ± 1.13 | 78.05 ± 1.14 | **79.21 ± 1.23** | 73.61 ± 0.99 | 58.37 ± 2.24 | 87.90 ± 0.57 | **91.17 ± 0.32** | 90.25 ± 0.25 | 89.63 ± 0.31 | 86.09 ± 0.98 |
| OLM | 77.82 ± 0.48 | 76.56 ± 0.89 | **81.46 ± 0.61** | 80.77 ± 0.81 | 77.39 ± 1.29 | 92.17 ± 0.58 | **92.27 ± 0.78** | 91.63 ± 0.12 | 89.90 ± 0.67 | 91.83 ± 0.15 |
| LSM | 68.83 ± 1.19 | 70.41 ± 1.57 | 67.56 ± 1.52 | **74.89 ± 1.07** | 72.69 ± 3.56 | 78.97 ± 2.80 | 75.10 ± 1.15 | 82.25 ± 3.38 | **83.43 ± 0.65** | 78.97 ± 4.97 |
| PHCM | 81.16 ± 0.40 | 80.05 ± 0.45 | 81.96 ± 0.51 | 78.28 ± 0.64 | **82.26 ± 0.92** | 88.30 ± 0.81 | 79.80 ± 0.69 | 87.37 ± 0.72 | 86.63 ± 0.27 | **90.03 ± 0.63** |

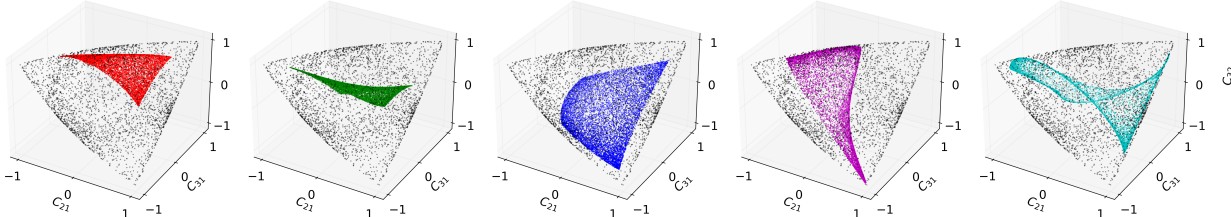

ECM Hyperplane    LECM Hyperplane    OLM Hyperplane    LSM Hyperplane    PHCM Hyperplane

*Figure 5.* Illustration of the decision hyperplanes in the correlation MLRs under five different geometries. The $3 \times 3$ correlation manifold can be embedded as an open elliptope in $\mathbb{R}^3$, by visualizing the strictly lower triangular part of each $C \in \mathrm{Cor}^+(3)$. The black dots denote the boundary. The PHCM hyperplane is defined by the one in the $\beta$-concatenated Poincaré space.

*Table 5.* Comparison of SPDMLR-Trivlz on raw covariances against CorMLR on raw correlations on all three datasets. The input matrix dimensions are $93 \times 93$, $63 \times 63$, and $20 \times 20$, respectively.

| Dataset | Measurement | SPDMLR-Trivlz | | | CorMLR | | | | |
|---|---|---|---|---|---|---|---|---|---|
| | | LEM | LCM | AIM | ECM | LECM | OLM | LSM | PHCM |
| Radar | Acc | 95.47 ± 0.66 | **95.55 ± 0.35** | 94.87 ± 0.87 | 89.47 ± 0.93 | 87.41 ± 0.23 | 85.79 ± 0.83 | 91.63 ± 0.32 | 83.33 ± 1.29 |
| | Fit Time (s/epoch) | 0.65 | 0.63 | 0.99 | **0.56** | 0.62 | 0.78 | 0.68 | 0.74 |
| HDM05 | Acc | 54.31 ± 1.65 | 45.12 ± 1.05 | 52.46 ± 2.44 | **65.57 ± 0.62** | 64.44 ± 0.63 | 62.86 ± 0.65 | 64.01 ± 0.92 | 62.78 ± 0.85 |
| | Fit Time (s/epoch) | 3.24 | 5.38 | 260.67 | 3.18 | 3.87 | 3.39 | 3.57 | **2.73** |
| FPHA | Acc | 84.13 ± 1.14 | 76.62 ± 0.43 | 83.25 ± 0.59 | **85.37 ± 0.16** | 85.24 ± 0.22 | 84.67 ± 0.27 | 80.17 ± 0.15 | 73.67 ± 0.32 |
| | Fit Time (s/epoch) | 0.51 | 0.52 | 18.96 | 0.51 | 0.64 | 0.8 | 0.81 | **0.45** |

consistent with the strong gains of correlation embeddings on HDM05 and suggests that correlation modeling is particularly beneficial when covariance matrices are dominated by large and highly variable diagonal components.

**Normalized Covariance vs. Correlation.** As correlation matrices can be viewed as normalized covariance matrices, a natural idea is to normalize covariance by a scalar, such as its largest eigenvalue. As discussed in App. H.6, feeding SPD networks with covariance matrices scaled by their largest eigenvalue leads to only marginal changes and could degrade performance. These results indicate that simple scalar scaling does not reproduce the benefits of correlation normalization. This can be explained from a statistical perspective. Since dividing a covariance matrix by a scalar is equivalent to uniformly rescaling raw samples before covariance computation, the normalized inputs remain covariance matrices. In contrast, correlation normalization rescales each pair of variables by their own standard deviations and produces standardized correlation coefficients, which is statistically distinct from scalar normalization.

**Activations.** Following HNN++ (Shimizu et al., 2021) and GyroSPD++ (Nguyen et al., 2024), CorNet omits explicit activations because the correlation manifold already intro-

duces nonlinearity. In App. H.7, we further study the effects of the activation function. Following Ganea et al. (2018, Sec. 3.2), we implement the activation via the tangent space. The results indicate that activation offers no benefit and can even degrade performance.

**Scalability.** We evaluate the efficiency of CorNet under different metrics across dimensions from $30 \times 30$ to $1000 \times 1000$. As shown in App. H.8, ECM is consistently the most efficient. At high dimensions, PHCM becomes the second most efficient due to its relatively simple diffeomorphism, whereas LECM is the slowest due to its costly $\log \circ \Theta$ mapping.

## 6. Conclusion

This paper systematically extends the FC, convolutional, and MLR layers to the correlation manifold under five newly developed Riemannian geometries. By preserving intrinsic correlation structures and enabling flexible variation of latent geometry within a unified network architecture, our framework highlights the distinct advantages of correlation manifolds beyond SPD and Grassmannian alternatives. In addition, we propose accurate backpropagation schemes for OLM and LSM. Extensive experiments demonstrate

the effectiveness and efficiency of our approach. These foundational layers open the door to constructing richer architectures on the correlation manifold, including RNNs, transformers, and residual networks.

## Impact Statement

This paper presents work whose goal is to advance the field of Machine Learning. There are many potential societal consequences of our work, none which we feel must be specifically highlighted here.

## Acknowledgments

This work was supported by the FIS project GUIDANCE (No. FIS2023-03251), the EU Horizon project ELLIOT (No. 101214398), and a DAAD Research Grant in Germany (57811724). We acknowledge CINECA for awarding high-performance computing resources under the ISCRA initiative, and the EuroHPC Joint Undertaking for granting access to Leonardo at CINECA, Italy.

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

# Appendix Contents

*Table 6.* Summary of notation. All numbers and operators are assumed to be real.

| Notation | Explanation |
|---|---|
| $(\mathcal{M}, g)$ or $\mathcal{M}$ | Riemannian manifold |
| $E$ or $E_{\mathcal{M}}$ | Origin of the manifold $\mathcal{M}$ |
| $T_P\mathcal{M}$ | Tangent space at $P \in \mathcal{M}$ |
| $\{O_k\}_{k=1}^m$ | Orthonormal basis over the $m$-dimensional $T_E\mathcal{M}$ |
| $g_p(\cdot, \cdot)$ or $\langle \cdot, \cdot \rangle_P$ | Riemannian metric at $P \in \mathcal{M}$ |
| $\|\cdot\|_P$ | Norm induced by $\langle \cdot, \cdot \rangle_P$ on $T_P\mathcal{M}$ |
| $\mathrm{Log}_P$ | Riemannian logarithmic map at $P$ |
| $\mathrm{Exp}_P$ | Riemannian exponential map at $P$ |
| $\gamma(t; P, Q)$ | Geodesic connecting $P, Q \in \mathcal{M}$ |
| $\Gamma_{P \to Q}$ | Riemannian parallel transportation along the geodesic connecting $P$ and $Q$ |
| $H_{a,p}$ or $H_{A,P}$ | Margin hyperplane |
| $f_{*,P}$ | Differential map of the smooth map $f$ at $P \in \mathcal{M}$ |
| $v_k(X)$ | $v_k(X)$ in Riemannian MLR (Eq. (2)) for $X \in \mathcal{M}$ |
| $\mathcal{F}: \mathcal{N} \to \mathcal{M}$ | Riemannian FC layer from $\mathcal{N}$ to $\mathcal{M}$, defined by Eq. (9) |
| $\mathbb{R}, \mathbb{R}^n \& \mathbb{R}^{n \times n}$ | Euclidean spaces of real scalars, $n$-dimensional real vectors, and $n \times n$ matrices |
| $\mathrm{Diag}(n)$ | Euclidean space of $n \times n$ diagonal matrices |
| $\mathrm{Diag}^+(n)$ | Manifold of $n \times n$ positive diagonal matrices |
| $\mathcal{S}^n$ | Euclidean space of $n \times n$ symmetric matrices |
| $\mathrm{Hol}(n)$ | Euclidean space of $n \times n$ symmetric matrices with null diagonals |
| $\mathrm{Row}_0(n)$ | Euclidean space of $n \times n$ symmetric matrices with null row sum |
| $\mathcal{S}^n_{++}$ | Manifold of $n \times n$ SPD matrices |
| $\mathrm{Row}_1^+(n)$ | Manifold of $n \times n$ SPD matrices with unit row sum. |
| $\mathrm{Cor}^+(n)$ | Manifold of $n \times n$ full rank correlation matrices |
| $\mathrm{LT}(n)$ | Euclidean space of $n \times n$ lower triangular matrices |
| $\mathrm{LT}^1(n)$ | Euclidean space of $n \times n$ lower triangular matrices with unit diagonals |
| $\mathrm{LT}^0(n)$ | Euclidean space of $n \times n$ lower triangular matrices with null diagonals |
| $\mathrm{LT}^+(n)$ | Cholesky manifold of $n \times n$ lower triangular matrices with positive diagonals |
| $\mathcal{L}^n$ | Manifold of $n \times n$ lower triangular matrices with positive diagonals and unit row norm |
| $\mathbb{PHS}^{n-1}$ | Product space of $n-1$ open hemispheres |
| $\mathbb{PP}^{n-1}$ | Product space of $n-1$ Poincaré balls |
| $\langle \cdot, \cdot \rangle \& \|\cdot\|$ | Canonical Euclidean inner product and norm |
| $\langle \cdot, \cdot \rangle^{(\alpha,\beta,\gamma)} \& \langle \cdot, \cdot \rangle^{(\alpha,\delta,\zeta)}$ | Permutation-invariant inner product over $\mathrm{Hol}(n)$ & $\mathrm{Row}_0(n)$ |
| $\log, \exp, \& \mathrm{Chol}$ | Matrix logarithm, exponentiation, Cholesky decomposition |
| $\lfloor \cdot \rfloor$ | Returns the strictly lower triangular matrix of a square matrix |
| $\phi^{\mathrm{EC}}(\cdot)$ | $\phi^{\mathrm{EC}}(C) = \lfloor \Theta(C) \rfloor$ the isometry w.r.t. ECM |
| $\mathbb{D}(\cdot)$ | Returns a diagonal matrix with diagonals from a square matrix |
| $\mathrm{diag}(\cdot)$ | Returns a diagonal matrix from an input vector |
| $\mathrm{Dv}(\cdot)$ | Returns a vector of diagonal elements from a square matrix |
| $(\cdot)_{\frac{1}{2}}$ | $(S)_{\frac{1}{2}} = \lfloor S \rfloor + \frac{1}{2}\mathbb{D}(S)$ for any square matrix $S$ |
| $\odot$ | Hadamard product |
| $\mathrm{Cor}$ | $\mathrm{Cor} : \Sigma \in \mathcal{S}^n_{++} \longmapsto \mathbb{D}(\Sigma)^{-1/2}\Sigma\mathbb{D}(\Sigma)^{-1/2} \in \mathrm{Cor}^+(n)$ |
| $\Theta$ | $\Theta : C \in \mathrm{Cor}^+(n) \longmapsto \mathbb{D}(\mathrm{Chol}(C))^{-1}\mathrm{Chol}(C) \in \mathrm{LT}^1(n)$ |
| off | Returns a matrix in $\mathrm{Hol}(n)$ consisting of off-diagonal elements |
| $\mathrm{Log}^\circ \& \mathrm{Exp}^\circ$ | Off-log and its inverse |
| $\mathrm{Log}^\star \& \mathrm{Exp}^\star$ | Log-scaled and its inverse |
| $I$ or $I_n \& \mathbf{0}$ | Identity matrix & Zero matrix or vector |
| $\mathbf{1}$ | Vector with all 1 entities |
| $\mathbb{P}^n_K$ | General Poincaré ball, $\mathbb{P}^n_K = \left\{x \in \mathbb{R}^n \mid \|x\|^2 < -\frac{1}{K}\right\} (K < 0)$ |
| $\mathbb{P}^n$ | (Canonical) Poincaré ball, $\mathbb{P}^n = \mathbb{P}^n_{-1}$ |
| $\mathrm{H}\mathbb{S}^n$ | Open hemisphere, $\mathrm{H}\mathbb{S}^n = \left\{x \in \mathbb{R}^{n+1} \mid \|x\| = 1 \text{ and } x_{n+1} > 0\right\}$ |
| $\mathbb{H}^n$ | Hyperboloid, $\mathbb{H}^n = \left\{x \in \mathbb{R}^{n+1} \mid \|x\|_{\mathcal{L}}^2 = -1\right\}$ with $\|x\|_{\mathcal{L}}^2 = \sum_{i=1}^n x_i^2 - x_{n+1}^2$ |
| $\psi_{\mathrm{H}\mathbb{S}^n \to \mathbb{P}^n}$ $\psi_{\mathbb{P}^n \to \mathrm{H}\mathbb{S}^n}$ | Isometries between $\mathrm{H}\mathbb{S}^n$ and $\mathbb{P}^n$ |

## List of Acronyms

ECM      Euclidean–Cholesky Metric 1
LECM      Log-Euclidean–Cholesky Metric 1

## A. Notation

Tab. 6 summarizes all the notation used in this paper for better clarity.

## B. Full-Rank Correlation Geometries

This section follows all the notation in Tab. 6. As ECM, LECM, OLM, and LSM are pullback metrics from Euclidean spaces by diffeomorphisms, they are collectively called Log-Euclidean metrics (Thanwerdas, 2024). As all five metrics are pullback metrics, the Riemannian operators can be directly derived by the properties of Riemannian isometries (Chen et al., 2026, App. C.2), without computing Christoffel symbols or solving ODEs.

### B.1. Pullback Metrics

As all five involved Riemannian metrics on the correlation manifold are pullback metrics, we first review pullback metrics. The idea of pullback is ubiquitous in differential geometry and can be considered as a natural extension of the bijection in the set theory.

**Definition B.1** (Pullback Metrics (Lee, 2018)). Suppose $\mathcal{M}_1, \mathcal{M}_2$ are smooth manifolds, $g$ is a Riemannian metric on $\mathcal{M}_2$, and $f : \mathcal{M}_1 \to \mathcal{M}_2$ is a diffeomorphism. Then the pullback of $g$ by $f$ is defined pointwise,

$$(f^*g)_p(V, W) = g_{f(p)}(f_{*,p}(V), f_{*,p}(W)), \tag{15}$$

where $f_{*,p}(\cdot)$ is the differential map of $f$ at $p \in \mathcal{M}_1$, and $V, W \in T_p\mathcal{M}_1$. $f^*g$ is a Riemannian metric on $\mathcal{M}_1$, called the pullback metric of $g$ by $f$. Here, $f$ is also called a Riemannian isometry.

Although pullback metrics can also be defined by smooth maps (Lee, 2018), this paper focuses on diffeomorphisms.

### B.2. Symmetric Matrix Functions

This subsection reviews the eigenvalue function over symmetric matrices. For more in-depth discussions, please refer to Bhatia (2009, Ch. 2.7.13) or Bhatia (2013, Ch. V.3).

We denote $\mathcal{S}^n$ as the Euclidean space of $n \times n$ real symmetric matrices, and $\mathcal{S}^n_{++}$ as the SPD manifold of $n \times n$ SPD matrices. Let $\mathring{I}$ be an open interval of $\mathbb{R}$ and $f : \mathring{I} \to \mathbb{R}$ be a smooth function. The smooth map induced by $f$ for any symmetric matrix $S$ with all eigenvalues in $\mathring{I}$ is defined as

$$f : S \longmapsto Uf(\Sigma)U^\top \in \mathcal{S}^n, \text{ with } S = U\Sigma U^\top \text{ as the eigendecomposition.} \tag{16}$$

Its differential is known as the Daleckii-Krein formula:

$$f_{*,S}(V) = U\left(L \odot \left(U^\top V U\right)\right)U^\top, \quad \forall V \in \mathcal{S}^n, \tag{17}$$

$$L_{i,j} = \begin{cases} \frac{f(\sigma_i) - f(\sigma_j)}{\sigma_i - \sigma_j}, & \text{if } \sigma_i \neq \sigma_j \\ f'(\delta_i), & \text{otherwise} \end{cases} \tag{18}$$

where $L$ is called the Loewner matrix with the $(i, j)$-th element defined as Eq. (18), and $\odot$ denotes the Hadamard product. Two special cases are the matrix logarithm: $\log : \mathcal{S}^n_{++} \to \mathcal{S}^n$ and its inverse, the matrix exponentiation $\exp : \mathcal{S}^n \to \mathcal{S}^n_{++}$.

*Table 8.* Riemannian metrics on the correlation manifold with the associated isometric prototype spaces and diffeomorphisms.

| Metric | Prototype space | Diffeomorphisms | Properties |
|---|---|---|---|
| ECM (Thanwerdas & Pennec, 2022b) | $\mathrm{LT}^1(n) = \mathrm{LT}^0(n) + I_n$ | $\Theta : C \in \mathrm{Cor}^+(n) \longmapsto \mathbb{D}(\mathrm{Chol}(C))^{-1}\,\mathrm{Chol}(C) \in \mathrm{LT}^1(n)$ 
 $\Theta^{-1} = \mathrm{Cor} \circ \mathrm{Chol}^{-1} : \mathrm{LT}^1(n) \longrightarrow \mathrm{Cor}^+(n)$ | Null curvature |
| LECM (Thanwerdas & Pennec, 2022b) | $\mathrm{LT}^0(n)$ | $\log \circ \Theta : \mathrm{Cor}^+(n) \longrightarrow \mathrm{LT}^0(n)$ 
 $(\log \circ \Theta)^{-1} = \mathrm{Cor} \circ \mathrm{Chol}^{-1} \circ \exp : \mathrm{LT}^0(n) \longrightarrow \mathrm{Cor}^+(n)$ | Null curvature |
| OLM (Thanwerdas, 2024) | $\mathrm{Hol}(n)$ | $\mathrm{Log}^\circ : C \in \mathrm{Cor}^+(n) \longmapsto \mathrm{off} \circ \log(C) \in \mathrm{Hol}(n)$ 
 $(\mathrm{Log}^\circ)^{-1} = \mathrm{Exp}^\circ : H \in \mathrm{Hol}(n) \longmapsto \exp(\mathcal{D}(H) + H) \in \mathrm{Cor}^+(n)$ | Permutation-invariance 
 Null curvature |
| LSM (Thanwerdas, 2024) | $\mathrm{Row}_0(n)$ | $\mathrm{Log}^\star : C \in \mathrm{Cor}^+(n) \longmapsto \log(\mathcal{D}^\star(C) C \mathcal{D}^\star(C)) \in \mathrm{Row}_0(n)$ 
 $(\mathrm{Log}^\star)^{-1} = \mathrm{Exp}^\star : R \in \mathrm{Row}_0(n) \longmapsto \mathrm{Cor}(\exp(R)) \in \mathrm{Cor}^+(n)$ | Permutation-invariance 
 Inverse-consistency 
 Null curvature |
| PHCM (Thanwerdas & Pennec, 2022b) | $\mathbb{PHS}^{n-1}$ | $\mathrm{Chol} : \mathrm{Cor}^+(n) \longrightarrow \mathcal{L}^n \cong \mathbb{PHS}^{n-1}$ 
 $\mathrm{Chol}^{-1} : \mathcal{L}^n \cong \mathbb{PHS}^{n-1} \longrightarrow \mathrm{Cor}^+(n)$ | Nonpositive 
 sectional curvature |

## B.3. Geometries of the Correlation Manifold

Following the notation in Tab. 6, this subsection is a more detailed discussion of Sec. 2 in the main paper. The involved five geometries on the correlation matrices can be classified into two classes: (1) non-permutation-invariant metrics, including ECM, LECM, and PHCM; and (2) permutation-invariant metrics, including OLM and LSM. Tab. 8 summarizes the diffeomorphisms and prototype spaces discussed in Sec. 2.

### B.3.1. NON-PERMUTATION-INVARIANT METRICS

The non-permutation-invariant metrics (Thanwerdas & Pennec, 2022b), namely ECM, LECM, and PHCM, are defined by pullback:

$$\text{ECM: } \mathrm{Cor}^+(n) \xrightleftharpoons[\Theta^{-1}=\mathrm{Cor}\,\circ\,\mathrm{Chol}^{-1}]{\Theta=\mathbb{D}^{-1}(\mathrm{Chol}(\cdot))\,\mathrm{Chol}(\cdot)} \mathrm{LT}^1(n) = I_n + \mathrm{LT}^0(n), \tag{19}$$

$$\text{LECM: } \mathrm{Cor}^+(n) \xrightleftharpoons[(\log\,\circ\,\Theta)^{-1}=\mathrm{Cor}\,\circ\,\mathrm{Chol}^{-1}\,\circ\,\exp]{\log\,\circ\,\Theta} \mathrm{LT}^0(n), \tag{20}$$

$$\text{PHCM: } \mathrm{Cor}^+(n) \xrightleftharpoons[\mathrm{Chol}^{-1}]{\mathrm{Chol}} \mathcal{L}^n \cong \mathbb{PHS}^{n-1} := \prod_{i=1}^{n-1}\{\mathrm{HS}^i, \alpha_i g^{\mathrm{HS}^i}\}, \tag{21}$$

where each $\alpha_i > 0$ is the positive weight. In the following, we first review the associated maps in ECM and LECM, followed by a discussion of PHCM.

**ECM and LECM.** For any $C \in \mathrm{Cor}^+(n)$, $V \in T_C \mathrm{Cor}^+(n) \cong \mathrm{Hol}(n)$, $K \in \mathrm{LT}^1(n)$ and $X, \xi \in \mathrm{LT}^0(n)$, the involved maps and their differentials in ECM and LECM are

$$\Theta(C) = \mathbb{D}(L)^{-1} L, \tag{22}$$

$$\Theta^{-1}(K) = \mathbb{D}\left(KK^\top\right)^{-\frac{1}{2}} KK^\top \mathbb{D}\left(KK^\top\right)^{-\frac{1}{2}}, \tag{23}$$

$$\log(K) = \sum_{k=1}^{n-1} \frac{(-1)^{k-1}}{k}\left(K - I_n\right)^k, \tag{24}$$

$$\exp(\xi) = \sum_{k=0}^{n-1} \frac{1}{k!}\xi^k, \tag{25}$$

$$\Theta_{*,C}(V) = \Theta(C)\left(L^{-1}VL^{-\top}\right)_{\frac{1}{2}} - \frac{1}{2}\mathbb{D}\left(L^{-1}VL^{-\top}\right)\Theta(C), \tag{26}$$

$$(\Theta_{*,C})^{-1}(\xi) = \left(L\xi^\top - C\mathbb{D}\left(L\xi^\top\right)\right)\mathbb{D}(L) + \mathbb{D}(L)\left(\xi L^\top - \mathbb{D}\left(L\xi^\top\right)C\right), \tag{27}$$

$$\log_{*,K}(\xi) = \sum_{k=1}^{n-1} \frac{(-1)^{k-1}}{k}\left[\left(K - I_n\right)^{k-1}\xi + \cdots + \xi\left(K - I_n\right)^{k-1}\right], \tag{28}$$

*Table 9.* Riemannian operators under the non-permutation-invariant log-Euclidean Metrics. Here, $C, C' \in \mathrm{Cor}^+(n)$ are correlation matrices and $V, W \in T_C\mathrm{Cor}^+(n) \cong \mathrm{Hol}(n)$ are tangent vectors. Although the inner product $\langle \cdot, \cdot \rangle$ could be any Euclidean inner product, this paper focuses on the canonical one.

| Operation | ECM | LECM |
|---|---|---|
| $g_C(V, W)$ | $\langle \Theta_{*,C}(V), \Theta_{*,C}(W) \rangle$ | $\langle (\log \circ \Theta)_{*,C}(V), (\log \circ \Theta)_{*,C}(W) \rangle$ |
| $\mathrm{Exp}_C(V)$ | $\Theta^{-1}\left(\Theta\left(C\right) + \Theta_{*,C}\left(V\right)\right)$ | $(\log \circ \Theta)^{-1}\left(\log \circ \Theta\left(C\right) + (\log \circ \Theta)_{*,C}\left(V\right)\right)$ |
| $\mathrm{Log}_C(C')$ | $\Theta_{*,\Theta(C)}^{-1}\left(\Theta\left(C'\right) - \Theta\left(C\right)\right)$ | $(\log \circ \Theta)_{*,\log \circ \Theta(C)}^{-1}\left(\log \circ \Theta\left(C'\right) - \log \circ \Theta\left(C\right)\right)$ |
| $\gamma(t; C, C')$ | $\Theta^{-1}\left((1 - t)\Theta\left(C\right) + t\Theta\left(C'\right)\right)$ | $(\log \circ \Theta)^{-1}\left((1 - t)\log \circ \Theta\left(C\right) + t\log \circ \Theta\left(C'\right)\right)$ |
| $\mathrm{d}(C, C')$ | $\left\|\Theta\left(C\right) - \Theta\left(C'\right)\right\|$ | $\left\|\log \circ \Theta\left(C\right) - \log \circ \Theta\left(C'\right)\right\|$ |
| Fréchet mean | $\Theta^{-1}\left(\frac{1}{k}\sum_{i=1}^k \Theta\left(C_i\right)\right)$ | $(\log \circ \Theta)^{-1}\left(\frac{1}{k}\sum_{i=1}^k \log \circ \Theta\left(C_i\right)\right)$ |
| Curvature | $0$ | $0$ |
| $\Gamma_{C \to C'}(V)$ | $(\Theta_{*,C'})^{-1}\left(\Theta_{*,C}\left(V\right)\right)$ | $((\log \circ \Theta)_{*,C'})^{-1}\left((\log \circ \Theta)_{*,C}\left(V\right)\right)$ |

$$\exp_{*,X}(\xi) = \sum_{k=1}^{n-1} \frac{1}{k!}\left(X^{k-1}\xi + X^{k-2}\xi X + \cdots + \xi X^{k-1}\right), \tag{29}$$

$$(\log \circ \Theta)_{*,C}(V) = \log_{*,\Theta(C)}\left(\Theta_{*,C}(V)\right), \tag{30}$$

$$\mathrm{Chol}_{*,C}(V) = L\left(L^{-1}VL^{-\top}\right)_{\frac{1}{2}}, \tag{31}$$

$$(\mathrm{Chol}_{*,C})^{-1}(Z) = LZ^\top + ZL^\top, \quad \forall Z \in T_L\mathrm{LT}^+(n) \cong \mathrm{LT}(n). \tag{32}$$

where $L$ is the Cholesky factor of $C$ and $I_n$ is the $n \times n$ identity matrix. Due to the nilpotency of $\mathrm{LT}^0(n)$, the matrix logarithm over $\mathrm{LT}^1(n)$ and exponentiation over $\mathrm{LT}^0(n)$ are free from eigendecomposition. With the above equations, Tab. 9 summarizes the Riemannian operators under ECM and LECM.

**PHCM.** It is the pullback metric by the Cholesky decomposition from the product space $\prod_{i=1}^{n-1}\{\mathrm{H}\mathbb{S}^i, \alpha_i g^{\mathrm{H}\mathbb{S}^i}\}$, where each $\alpha_i$ denotes positive weights and $g^{\mathrm{H}\mathbb{S}^i}$ denotes the metric tensor over $\mathrm{H}\mathbb{S}^i$. Particularly, the PHCM with all weights equal to 1 is called the canonical PHCM. Without loss of generality, we focus on the canonical PHCM. The closed-form expressions of the Riemannian operations under PHCM are a bit heavy as they are obtained by the product metric.

Given $C \in \mathrm{Cor}^+(n)$ and $L = \mathrm{Chol}(C) \in \mathcal{L}^n$, we denote $\Psi = \psi^1 \times \cdots \times \psi^{n-1} : \mathcal{L}^n \to \prod_{i=1}^{n-1} \mathrm{H}\mathbb{S}^i$ with each $\psi^i$ as

$$\psi^i(L) = (L_{i+1,1}, \ldots, L_{i+1,i+1}) \in \mathrm{H}\mathbb{S}^i, \tag{33}$$

where $L_{i+1} = (L_{i+1,1}, \ldots, L_{i+1,i+1}, 0, \ldots, 0)$ is the $(i+1)$-th row of $L$. The Riemannian operators under PHCM can be obtained using the product geometry and $\mathrm{H}\mathbb{S}^i$ geometry. Following the notation in Tab. 9, the Riemannian metrics, logarithm, exponentiation, and geodesic (distance) under the canonical PHCM are

$$g_C(V, V) = \left\|\mathbb{D}(L)^{-1}L\left(L^{-1}VL^{-\top}\right)_{\frac{1}{2}}\right\|^2, \tag{34}$$

$$\mathrm{Exp}_C(V) = (\mathrm{Chol})^{-1}\left(\psi^{-1}\left(\mathrm{Exp}_{\psi^1(L)}^{\mathrm{H}\mathbb{S}^1}(\psi^1(\mathrm{Chol}_{*,C}(V))), \cdots, \mathrm{Exp}_{\psi^{n-1}(L)}^{\mathrm{H}\mathbb{S}^{n-1}}(\psi^{n-1}(\mathrm{Chol}_{*,C}(V)))\right)\right), \tag{35}$$

$$\mathrm{Log}_C(C') = (\mathrm{Chol})_{*,C'}^{-1}\left(\psi^{-1}\left(\mathrm{Log}_{\psi^1(L)}^{\mathrm{H}\mathbb{S}^1}(\psi^1(L')), \cdots, \mathrm{Log}_{\psi^{n-1}(L)}^{\mathrm{H}\mathbb{S}^{n-1}}(\psi^{n-1}(L'))\right)\right), \tag{36}$$

$$\gamma(t; C, C') = (\mathrm{Chol})^{-1}\left(\psi^{-1}\left(\gamma^{\mathrm{H}\mathbb{S}^1}(t; \psi^1(C), \psi^1(C')), \cdots, \gamma^{\mathrm{H}\mathbb{S}^{n-1}}(t; \psi^{n-1}(C), \psi^{n-1}(C'))\right)\right), \tag{37}$$

$$d\left(C, C'\right)^2 = \sum_{i=1}^{n-1} \mathrm{arccosh}\left(-\left\langle \psi^i(L), \psi^i(L')\right\rangle_{\mathcal{L}}\right)^2, \tag{38}$$

where $L = \mathrm{Chol}(C) \in \mathcal{L}^n$, $L' = \mathrm{Chol}(C') \in \mathcal{L}^n$, and $\mathrm{Log}^{\mathrm{H}\mathbb{S}^i}$, $\mathrm{Exp}^{\mathrm{H}\mathbb{S}^i}$ and $\gamma^{\mathrm{H}\mathbb{S}^i}$ are the counterparts over $\mathrm{H}\mathbb{S}^i$, which have closed-form expressions (Thanwerdas & Pennec, 2022b, Thm. 4.2). Here, $\|\cdot\|_{\mathcal{L}}$ is the norm induced by Lorentz inner product:

$$\|x\|_{\mathcal{L}}^2 = \sum_{i=1}^n x_i^2 - x_{n+1}^2, \quad \forall x \in \mathbb{R}^{n+1}. \tag{39}$$

*Remark* B.2. The Riemannian structure of $\mathrm{H}\mathbb{S}^n$ is defined by the pullback metric from the hyperboloid model. All the Riemannian operators over $\mathrm{H}\mathbb{S}^n$ have closed-form expressions (Thanwerdas & Pennec, 2022b, Thm. 4.2).

### B.3.2. PERMUTATION-INVARIANT METRICS

Let $\mathfrak{S}^n$ be the group of permutation matrices $P_\sigma = \left[\delta_{i,\sigma(j)}\right]_{1 \leqslant i,j \leqslant n}$ by permutation $\sigma$, and $\mathcal{D}^\pm(n) = \{\mathrm{diag}\left((\varepsilon_1, \ldots, \varepsilon_n)\right), \varepsilon \in \{-1, 1\}^n\}$ be the group of diagonal matrices with coefficients in $\{-1, 1\}$. Thanwerdas (2024, Thm. 1.1) showed that the largest congruence action on full-rank correlation matrices is the action of signed permutation matrices:

$$\star : (A, C) \in \mathfrak{S}^\pm(n) \times \mathrm{Cor}^+(n) \longmapsto ACA^\top \in \mathrm{Cor}^+(n), \tag{40}$$

with $\mathfrak{S}^\pm(n) = \mathcal{D}^\pm(n)\mathfrak{S}^n$. Based on this finding, Thanwerdas (2024) proposed two permutation-invariant metrics, namely OLM and LSM, by pulling back permutation-invariant inner products via the following permutation-equivariant diffeomorphisms:

$$\mathrm{Cor}^+(n) \xrightleftharpoons[\mathrm{Exp}^\circ]{\mathrm{Log}^\circ = \mathrm{off} \circ \log_*} \mathrm{Hol}(n), \tag{41}$$

$$\mathrm{Cor}^+(n) \xrightleftharpoons[\mathrm{Exp}^\star = \mathrm{Cor} \circ \exp]{\mathrm{Log}^\star} \mathrm{Row}_0(n), \tag{42}$$

$$\mathrm{Exp}^\circ : \mathrm{Hol}(n) \ni H \longmapsto \exp(\mathcal{D}(H) + H), \tag{43}$$

$$\mathrm{Log}^\star : \mathrm{Cor}^+(n) \ni C \longmapsto \log(\mathcal{D}^\star(C)C\mathcal{D}^\star(C)) \in \mathrm{Row}_0(n). \tag{44}$$

where $\log(\cdot)$ and $\exp(\cdot)$ are symmetric matrix logarithm and exponentiation. The involved $\mathcal{D}$ and $\mathcal{D}^\star$ can be formally expressed as $\mathcal{D} : \mathrm{Hol}(n) \to \mathrm{Diag}(n)$ and $\mathcal{D}^\star : \mathrm{Cor}^+(n) \to \mathrm{Diag}^+(n)$, where $\mathrm{Diag}(n)$ denotes the Euclidean space of $n \times n$ diagonal matrices, and $\mathrm{Diag}^+(n)$ is a submanifold of $\mathrm{Diag}(n)$, consisting of positive diagonal matrices.

The differentials of $\mathrm{Log}^\circ$ and $\mathrm{Log}^\star$ and their inverses can be calculated by the differential of symmetric matrix logarithm and exponentiation (Thanwerdas, 2024, Thms. 2.4 and 4.1). Given $C \in \mathrm{Cor}^+(n)$, tangent vector $V \in T_C\mathrm{Cor}^+(n) \cong \mathrm{Hol}(n)$, $H, W \in \mathrm{Hol}(n)$, and $S = H + \mathcal{D}(H) = U\Delta U^\top$, the differential of $\mathrm{Log}^\circ$ and its inverse $\mathrm{Exp}^\circ$ are

$$\mathrm{Log}^\circ_{*,C}(V) = \mathrm{off}\left(\log_{*,C}(V)\right), \tag{45}$$

$$\mathrm{Exp}^\circ_{*,H}(W) = \exp_{*,S}\left(W + \mathcal{D}_{*,H}(W)\right), \tag{46}$$

$$\mathcal{D}_{*,H}(W) = -\mathrm{diag}\left(\left(H^0\right)^{-1}\mathbb{D}\left(\exp_{*,S}(W)\right)\mathbf{1}\right), \tag{47}$$

$$\mathcal{S}^n_{++} \ni H^0_{il} = \sum_{j,k} P_{ij}P_{ik}P_{lj}P_{lk}L_{j,k}, \tag{48}$$

where $L$ is the Loewner matrix of $\exp_{*,S}$, and $\mathbf{1}$ is the vector of all 1 entities. Here, $\log_*$ and $\exp_*$ can be calculated using the Daleckii-Krein formula of the symmetric matrix, while $\mathrm{diag}(\cdot) : \mathbb{R}^n \to \mathrm{Diag}(n)$ returns a diagonal matrix from an input vector. Further denoting $X, Y \in \mathrm{Row}_0(n)$ and $\Sigma = \mathcal{D}^\star(C)C\mathcal{D}^\star(C)$, the differentials of $\mathrm{Log}^\star$ and its inverse $\mathrm{Exp}^\star$ are

$$\mathrm{Log}^\star_{*,C}(V) = \log_{*,\Sigma}\left(\Delta V \Delta + \frac{1}{2}\left(V^0\Sigma + \Sigma V^0\right)\right), \tag{49}$$

$$\mathrm{Exp}^\star_{*,X}(Y) = \Delta^{-1}\left[\exp_{*,X}(Y) - \frac{1}{2}\left(\Delta^{-2}\mathbb{D}\left(\exp_{*,X}(Y)\right)\Sigma + \Sigma\mathbb{D}\left(\exp_{*,X}(Y)\right)\Delta^{-2}\right)\right]\Delta^{-1} \tag{50}$$

with $\Delta = \mathbb{D}(\Sigma)^{1/2}$ and $V^0 = -2\,\mathrm{diag}\left((I_n + \Sigma)^{-1}\Delta V \Delta \mathbf{1}\right)$.

As both $\mathrm{Log}^\star_*$ and $\mathrm{Log}^\circ_*$ are permutation-equivariant (Thanwerdas, 2024), permutation-invariant metrics over the correlation manifold can be induced by permutation-invariant inner products over $\mathrm{Hol}(n)$ and $\mathrm{Row}_0(n)$, respectively. The following two theorems review such inner products.

**Theorem B.3** (Permutation-invariant inner products on $\mathrm{Hol}(n)$ (Thanwerdas, 2022)). *Supposing $n \geq 4$, permutation-invariant inner products on $\mathrm{Hol}(n)$ are:*

$$\langle X_1, X_2\rangle^{(\alpha,\beta,\gamma)} = \alpha\,\mathrm{tr}(X_1 X_2) + \beta\,\mathrm{Sum}\left(X_1 X_2\right) + \gamma\,\mathrm{Sum}(X_1)\,\mathrm{Sum}(X_2), \quad \forall X_1, X_2 \in \mathrm{Hol}(n), \tag{51}$$

*Table 10.* Riemannian geometries under the permutation-invariant log-Euclidean Metrics.

| Operation | OLM | LSM |
|---|---|---|
| $g_C(V, W)$ | $\langle \mathrm{Log}^{\circ}_{*,C}(V), \mathrm{Log}^{\circ}_{*,C}(W) \rangle^{(\alpha,\beta,\gamma)}$ | $\langle \mathrm{Log}^{\star}_{*,C}(V), \mathrm{Log}^{\star}_{*,C}(W) \rangle^{(\alpha,\delta,\zeta)}$ |
| $\mathrm{Exp}_C(V)$ | $\mathrm{Exp}^{\circ}\left(\mathrm{Log}^{\circ}(C) + \mathrm{Log}^{\circ}_{*,C}(V)\right)$ | $\mathrm{Exp}^{\star}\left(\mathrm{Log}^{\star}(C) + \mathrm{Log}^{\star}_{*,C}(V)\right)$ |
| $\mathrm{Log}_C(C')$ | $\mathrm{Exp}^{\circ}_{*,\mathrm{Log}^{\circ}(C)}\left(\mathrm{Log}^{\circ}(C') - \mathrm{Log}^{\circ}(C)\right)$ | $\mathrm{Exp}^{\star}_{*,\mathrm{Log}^{\star}(C)}\left(\mathrm{Log}^{\star}(C') - \mathrm{Log}^{\star}(C)\right)$ |
| $\gamma(t; C, C')$ | $\mathrm{Exp}^{\circ}\left((1-t)\mathrm{Log}^{\circ}(C) + t\mathrm{Log}^{\circ}(C')\right)$ | $\mathrm{Exp}^{\star}\left((1-t)\mathrm{Log}^{\star}(C) + t\mathrm{Log}^{\star}(C')\right)$ |
| $\mathrm{d}(C, C')$ | $\|\mathrm{Log}^{\circ}(C) - \mathrm{Log}^{\circ}(C')\|^{(\alpha,\beta,\gamma)}$ | $\|\mathrm{Log}^{\star}(C) - \mathrm{Log}^{\star}(C')\|^{(\alpha,\delta,\zeta)}$ |
| Fréchet mean | $\mathrm{Exp}^{\circ}\left(\frac{1}{k}\sum_{i=1}^{k}\mathrm{Log}^{\circ}(C_i)\right)$ | $\mathrm{Exp}^{\star}\left(\frac{1}{k}\sum_{i=1}^{k}\mathrm{Log}^{\star}(C_i)\right)$ |
| Curvature | 0 | 0 |
| $\Gamma_{C\to C'}(V)$ | $\left(\mathrm{Log}^{\circ}_{*,C'}\right)^{-1}\left(\mathrm{Log}^{\circ}_{*,C}(V)\right)$ | $\left(\mathrm{Log}^{\star}_{*,C'}\right)^{-1}\left(\mathrm{Log}^{\star}_{*,C}(V)\right)$ |
| Properties | Permutation-invariance
Singed-permutation-invariance $(\beta = \gamma = 0)$
Inverse-consistency | Permutation-invariance |

with $\alpha > 0$, $2\alpha + (n-2)\beta > 0$, and $\alpha + (n-1)(\beta + n\gamma) > 0$. *For $n = 3$, permutation-invariant inner products have the same form with $\alpha = 0$:*

$$\langle X_1, X_2 \rangle^{(\alpha,\beta,\gamma)} = \beta \, \mathrm{Sum}(X_1 X_2) + \gamma \, \mathrm{Sum}(X_1) \, \mathrm{Sum}(X_2), \quad \text{with } \beta > 0 \text{ and } \beta + 3\gamma > 0. \tag{52}$$

*For $n = 2$, they have the same form with $\alpha = \beta = 0$:*

$$\langle X_1, X_2 \rangle^{(\alpha,\beta,\gamma)} = \gamma \, \mathrm{Sum}(X_1) \, \mathrm{Sum}(X_2), \quad \text{with } \gamma > 0. \tag{53}$$

**Theorem B.4** (Permutation-invariant inner products on $\mathrm{Row}_0(n)$ (Thanwerdas, 2024)). *For $n \geq 4$, permutation-invariant inner products on $\mathrm{Row}_0(n)$ are*

$$\langle Y_1, Y_2 \rangle^{(\alpha,\delta,\zeta)} = \alpha \, \mathrm{tr}(Y_1 Y_2) + \delta \, \mathrm{tr}(\mathbb{D}(Y_1)\mathbb{D}(Y_2)) + \zeta \, \mathrm{tr}(Y_1) \, \mathrm{tr}(Y_2), \quad \forall Y_1, Y_2 \in \mathrm{Row}_0(n), \tag{54}$$

with $\alpha > 0$, $n\alpha + (n-2)\delta > 0$, and $n\alpha + (n-1)(\delta + n\zeta) > 0$. *For $n = 3$, the permutation-invariant inner products have the same form with $\alpha = 0$. For $n = 2$, they have the same form with $\alpha = \delta = 0$.*

As shown by Thanwerdas (2022), OLM is further invariant to signed-permutation under $\beta = \gamma = 0$, where the associated $\langle \cdot, \cdot \rangle^{(\alpha,0,0)}$ is reduced to the scaled canonical Euclidean inner product:

$$\langle V, W \rangle^{(\alpha,0,0)} = \alpha \langle V, W \rangle, \quad \forall V, W \in \mathrm{Hol}(n). \tag{55}$$

In the main paper, we assume that $\langle \cdot, \cdot \rangle^{(\alpha,\beta,\gamma)}$ and $\langle \cdot, \cdot \rangle^{(\alpha,\delta,\zeta)}$ are the canonical Euclidean inner product.

Lastly, we briefly review inverse-consistency, a property exclusive to LSM. The cor-inversion is defined as $\mathcal{I} : \mathrm{Cor}^+(n) \ni C \mapsto \mathrm{Cor}\left(C^{-1}\right) \in \mathrm{Cor}^+(n)$ (Thanwerdas, 2024, Def. 1.4). It corresponds to the matrix inversion $\mathrm{inv} : \mathcal{S}^n_{++} \ni \Sigma \mapsto \Sigma^{-1} \in \mathcal{S}^n_{++}$, as represented on the following commuting diagram:

$$\begin{array}{ccc} \mathcal{S}^n_{++} & \xrightarrow{\mathrm{inv}} & \mathcal{S}^n_{++} \\ \mathrm{Cor}\downarrow & & \downarrow\mathrm{Cor} \\ \mathrm{Cor}^+(n) & \xrightarrow{\mathcal{I}} & \mathrm{Cor}^+(n) \end{array} \tag{56}$$

As shown by Thanwerdas (2024, Thm. 1.7), LSM enjoys inverse-consistency:

$$\mathrm{Log}^{\star}(\mathcal{I}(C)) = -\mathrm{Log}^{\star}(C), \quad \forall C \in \mathrm{Cor}^+(n). \tag{57}$$

Tab. 10 summarizes the Riemannian structures of OLM and LSM.

*Remark* B.5. We make the following remarks w.r.t. OLM and LSM.

1. **Invariance and dimension:** Thms. B.3 and B.4 implies that when $n \leq 3$, the canonical inner products over $\mathrm{Hol}(n)$ and $\mathrm{Row}_0(n)$, as well as the induced OLM and LSM, are no longer invariant metrics. However, our main paper focuses on cases where $n > 3$.

2. **$\mathcal{D}$ and $\mathcal{D}^\star$:** $\mathcal{D}$ is also well-defined over $\mathcal{S}^n$, a surjective map $\mathcal{D} : \mathcal{S}^n \to \mathrm{Diag}(n)$. In this way, $\mathrm{Exp}^\circ : \mathcal{S}^n \to \mathrm{Cor}^+(n)$ is no longer bijective (Thanwerdas, 2024, Thm. 2.1). Similarly, $\mathcal{D}^\star$ is well defined over $\mathcal{S}_{++}^n$, a surjective map $\mathcal{D}^\star : \mathcal{S}_{++}^n \to \mathrm{Diag}^+(n)$. Consequently, $\mathrm{Log}^\star : \mathcal{S}_{++}^n \to \mathrm{Row}_0(n)$ is no longer bijective (Thanwerdas, 2024, Thm. 3.5).

## C. Revisiting Previous Layers

### C.1. Revisiting Multinomial Logistic Regression

We briefly review the Euclidean Multinomial Logistic Regression (MLR) and its Riemannian extensions (Lebanon & Lafferty, 2004; Ganea et al., 2018; Nguyen & Yang, 2023; Nguyen et al., 2024; Bdeir et al., 2024; Chen et al., 2024a;c). Given $C$ classes, the Euclidean MLR computes the multinomial probability of each class $k \in \{1, \dots, C\}$ for the input feature vector $x \in \mathbb{R}^n$:

$$p(y = k \mid x) \propto \exp\left(v_k(x)\right), \tag{58}$$

with $v_k(x) = \langle a_k, x \rangle - b_k$ and $b_k \in \mathbb{R}, a_k \in \mathbb{R}^n$. Lebanon & Lafferty (2004, Sec. 5) first reformulated $v_k(x)$ by the margin distance to the hyperplane:

$$v_k(x) = \mathrm{sign}(\langle a_k, x - p_k \rangle) \|a_k\| d(x, H_{a_k, p_k}), \tag{59}$$

$$H_{a_k, p_k} = \{x \in \mathbb{R}^n \mid \langle a_k, x - p_k \rangle = 0\}, \tag{60}$$

where $\langle a_k, p_k \rangle = b_k$, and $H_{a_k, p_k}$ is a margin hyperplane. Based on the above reformulation, Ganea et al. (2018); Nguyen & Yang (2023); Bdeir et al. (2024); Chen et al. (2024a;c) generalized the MLR to different manifolds. Given a manifold-valued input $X \in \mathcal{M}$, the MLR (Chen et al., 2024c) over $\mathcal{M}$ is defined as

$$p(y = k \mid X) \propto \exp\left(v_k(X)\right), \tag{61}$$

$$v_k(X) = \mathrm{sign}(\langle A_k, \mathrm{Log}_{P_k}(S) \rangle_{P_k}) \|A_k\|_{P_k} \, d(S, H_{A_k, P_k}), \tag{62}$$

$$H_{A_k, P_k} = \{S \in \mathcal{M} \mid \langle \mathrm{Log}_{P_k}(S), A_k \rangle_{P_k} = 0\}, \tag{63}$$

$$d(S, H_{A_k, P_k}) = \inf_{Q \in H_{A_k, P_k}} d(S, Q), \tag{64}$$

with $P_k \in \mathcal{M}$ and $A_k \in T_{P_k}\mathcal{M}$. Shimizu et al. (2021, Sec. 3.1) demonstrates that $P_k$ and $A_k$ in the hyperbolic Poincaré MLR can be optimized using a Euclidean vector at the tangent space at the zero vector along with a biasing scalar. Inspired by this, this paper sets $P_k = \mathrm{Exp}_E(\gamma_k[Z_k])$ and $A_k = \Gamma_{E \to P_k}(Z_k)$. Here, $E$ is the origin of the $m$-dimensional manifold $\mathcal{M}$, while $\gamma_k \in \mathbb{R}$ and $Z_k \in T_E\mathcal{M} \cong \mathbb{R}^m$ are the MLR parameters.

Following the nomenclature by Chen et al. (2024c), Eq. (63) and Eq. (64) are called the Riemannian hyperplane and Riemannian margin distance to the hyperplane, respectively. Obviously, solving the optimization problem in Eq. (64) is the most challenging part. To circumvent this problem, Chen et al. (2024c, Sec. 3.2) relaxed it via Riemannian trigonometry and approximately solved this problem. Unlike their method, this paper precisely solves Eq. (64) under different metrics in the correlation manifold.

### C.2. Revisiting Poincaré Layers

Let $\mathbb{P}_K^n = \left\{x \in \mathbb{R}^n \mid \|x\|^2 < -1/K\right\}$ be the Poincaré ball ($K < 0$). The Poincaré MLR (Ganea et al., 2018; Shimizu et al., 2021) and FC layers on Poincaré spaces (Shimizu et al., 2021) follow the same logic as Sec. 3.1 and Def. 3.4, respectively.

The Poincaré MLR was first proposed by Ganea et al. (2018), then simplified by Shimizu et al. (2021, Eq. 6). Given $x \in \mathbb{P}_K^n$, the Poincaré MLR is

$$p(y = k \mid x) \propto \exp\left(v_k(x)\right),$$

$$v_k(x) = \frac{2\|z_k\|}{\sqrt{|K|}} \mathrm{asinh}\left(\lambda_x^K \langle \sqrt{|K|}x, [z_k] \rangle \cosh(2\sqrt{|K|}\gamma_k) - \left(\lambda_x^K - 1\right) \sinh(2\sqrt{|K|}\gamma_k)\right), \tag{65}$$

where $\lambda_x^K = 2(1 - |K|\|x\|^2)^{-1}$ is the conformal factor, and $[z_k] = \frac{z_k}{\|z_k\|}$. Here, $z_k \in \mathbb{R}^n$ and $\gamma_k \in \mathbb{R}$ are parameters. Note that $\lim_{K \to 0} v_k(x) = 4(\langle a_k, x \rangle - b_k)$.

Based on the Poincaré MLR, Shimizu et al. (2021, Eq. 7) proposed the Poincaré FC layer $\mathcal{F}(\cdot) : \mathbb{P}_K^n \to \mathbb{P}_K^m$, which is

$$y = \frac{w}{1 + \sqrt{1 + |K|\|w\|^2}}, \qquad w_k = |K|^{-1/2} \sinh\left(\sqrt{|K|} v_k(x)\right), \tag{66}$$

where $\boldsymbol{z} = \{z_k \in \mathbb{R}^n\}_{k=1}^m$ and $\boldsymbol{\gamma} = \{\gamma_k \in \mathbb{R}\}_{k=1}^m$ are the FC parameters.

The Poincaré convolutional layer is defined by the Poincaré $\beta$-concatenation and FC layer. Poincaré $\beta$-concatenation is defined as the scaled concatenation via the tangent space, which generalizes the Euclidean concatenation. Given inputs $\{x_i \in \mathbb{P}_K^{n_i}\}_{i=1}^N$, it is defined as

$$\mathrm{Exp}_{\mathbf{0}}\left(\beta_n \left(\beta_{n_1}^{-1} v_1^\top, \cdots, \beta_{n_N}^{-1} v_N^\top\right)\right)^\top \in \mathbb{P}_K^n, \tag{67}$$

where $v_i = \mathrm{Log}_{\mathbf{0}}(x_i)$ and $n = \sum_{i=1}^N n_i$. Here, $\beta_{n_i}$ and $\beta_n$ are defined by the beta function, *i.e.*, $\beta_\alpha = \mathrm{B}(\alpha/2, 1/2)$. The inverse is called the Poincaré $\beta$-split. The Poincaré convolution is then defined as: (1) $\beta$-concatenating the multi-channel feature in a given receptive field; and (2) performing the Poincaré FC layer.

In the main paper, we focus on the unit Poincaré ball $\mathbb{P}^n$ with $K = -1$.

## D. Discussion on Correlation FC Layer

### D.1. Connections among FC Layers: Correlation, SPD, Poincaré, and Euclidean

We clarify the correspondence between our FC formulation in Eq. (9) and previous FC layers.

#### D.1.1. SPD Manifold

Nguyen et al. (2024, Props. 3.4–3.6) introduced three SPD FC layers based on the gyrovector spaces under Log-Euclidean Metric (LEM), Log-Cholesky Metric (LCM), and Affine-Invariant Metric (AIM), respectively. These gyro SPD FC layers share the same definition as Eq. (9), except that their signed distance and $v_k$ are defined by gyrovector spaces.

#### D.1.2. Poincaré Ball

We show that the Poincaré FC layer $\mathcal{F}(\cdot) : \mathbb{P}_K^n \to \mathbb{P}_K^m$ in Eq. (66) is also defined as our correlation FC layer in Def. 3.4.

We define the zero vector $\mathbf{0} \in \mathbb{P}_K^n$ as the Poincaré origin, as it is the identity element of the Poincaré gyrovector space (Ganea et al., 2018). Obviously, $\{e_k\}_{i=1}^m$ is the orthogonal basis over $T_{\mathbf{0}}\mathbb{P}_K^m$, where $e_k = (\delta_{ik})_{i=1}^m$. Corresponding to Eq. (9), we have

$$\mathrm{sign}(\langle \mathrm{Log}_{\mathbf{0}}(y), e_k \rangle)\, \mathrm{d}(y, H_{e_k, \mathbf{0}}) = v_k(x). \tag{68}$$

Compared with Eq. (56) by Shimizu et al. (2021), we only need to show the LHS. The sign can be calculated as

$$\begin{aligned} \mathrm{sign}(\langle \mathrm{Log}_{\mathbf{0}}(y), e_k \rangle_{\mathbf{0}}) &\overset{(1)}{=} \mathrm{sign}\left(4\left\langle \tanh^{-1}\left(\sqrt{-K}\|y\|\right) \frac{y}{\sqrt{-K}\|y\|}, e_k \right\rangle\right) \\ &\overset{(2)}{=} \mathrm{sign}\left(\langle y, e_k \rangle\right) \\ &= \mathrm{sign}\left(y_k\right). \end{aligned} \tag{69}$$

The above follows from the following.

(1) $\lambda_{\mathbf{0}}^K = \frac{2}{1 + K\|\mathbf{0}\|^2} = 2$ and $\mathrm{Log}_{\mathbf{0}}(y) = \tanh^{-1}\left(\sqrt{-K}\|y\|\right) \frac{y}{\sqrt{-K}\|y\|}$.

(2) $\tanh^{-1}(a) > 0, \iff a > 0$.

Therefore, the LHS of Eq. (68) is simplified as

$$\text{sign}(\langle \text{Log}_{\mathbf{0}}(y), e_k \rangle)\, \text{d}(y, H_{e_k, \mathbf{0}}) = \text{sign}\,(y_k)\, \text{d}(y, H_{e_k, \mathbf{0}})$$

$$\overset{(1)}{=} \text{sign}\,(y_k)\, \frac{1}{\sqrt{-K}}\, \sinh^{-1}\left( \frac{2\sqrt{-K}|y_k|}{1 + K\|y\|^2} \right) \tag{70}$$

$$\overset{(2)}{=} \frac{1}{\sqrt{-K}}\, \sinh^{-1}\left( \frac{2\sqrt{-K}\, y_k}{1 + K\|y\|^2} \right).$$

The above comes from the following.

(1) Thm. 5 by Ganea et al. (2018).

(2) $1 + K\|y\|^2 > 0$ by definition and $\text{sign}(a)\sinh^{-1}(|a|) = \sinh^{-1}(a), \forall a \in \mathbb{R}$.

The last equation in Eq. (70) is the LHS of Eq. (56) in Shimizu et al. (2021), indicating the equality.

### D.1.3. EUCLIDEAN SPACE

We show that the Euclidean FC layer $\mathcal{F}(\cdot) : \mathbb{R}^n \ni x \to y = Ax + b \in \mathbb{R}^m$ can also be defined as our correlation FC layer in Def. 3.4.

In Euclidean space $\mathbb{R}^n$, the zero vector $\mathbf{0} \in \mathbb{R}^n$ is the origin, and $\{e_k\}_{i=1}^m$ is the orthogonal basis over $T_0\mathbb{R}^m \cong \mathbb{R}^m$. Then, the RHS of Eq. (9) becomes

$$v_k(x) = \langle x - p_k, a_k \rangle$$

$$\overset{(1)}{=} \langle x, z_k \rangle - \gamma_k \|z_k\|. \tag{71}$$

where (1) comes from $\text{Exp}_{\mathbf{0}}(\gamma_k[z_k]) = \gamma_k[z_k]$ and $\Gamma_{\mathbf{0} \to p_k}(z_k) = z_k$. The above takes the form of $\langle x, a_k \rangle + b_k$.

On the other hand, the LHS of Eq. (9) becomes

$$\text{sign}(\langle \text{Log}_{\mathbf{0}}(y), e_k \rangle)\, \text{d}(y, H_{e_k, \mathbf{0}}) \overset{(1)}{=} \text{sign}(y_k)\, \text{d}(y, H_{e_k, \mathbf{0}})$$

$$\overset{(2)}{=} y_k. \tag{72}$$

The above comes from the following.

(1) $\text{Log}_0(y) = y$ and $\langle y, e_k \rangle = y_k$.

(2) $\text{d}(y, H_{e_k, \mathbf{0}}) = \frac{|\langle y, e_k \rangle|}{\|e_k\|} = |y_k|$.

### D.2. Log-Euclidean Layers under Product Geometry

We first review some basic facts of the product geometry, and then discuss the Log-Euclidean correlation MLR and FC layer under the product geometry.

**Product of correlation.** Given a manifold $(\mathcal{M}, g)$, the $n$-fold product is $(\mathcal{M}^n, g) = \prod_{i=1}^n (\mathcal{M}, g)$. Each point and tangent vector over $\mathcal{M}^n$ are

$$\mathcal{M}^n \ni P = (P_1 \in \mathcal{M}, \cdots, P_n \in \mathcal{M}), \tag{73}$$

$$T_P\mathcal{M}^n \ni V = (V_1 \in T_{P_1}\mathcal{M}, \cdots, V_n \in T_{P_n}\mathcal{M}). \tag{74}$$

The product metric is

$$\langle V, W \rangle_P = \sum_{i=1}^n \langle V_i, W_i \rangle_{P_i}, \quad \forall V, W \in T_P\mathcal{M}^n. \tag{75}$$

**Correlation MLR.** Following Thm. 3.1, the MLR layer for the input $\boldsymbol{X} = \{X_j \in \text{Cor}^+(n)\}_{j=1}^c \in (\text{Cor}^+(n))^c$ is

$$
\begin{aligned}
v(\boldsymbol{X}) &\overset{(1)}{=} \langle \phi(X), \phi_{*,\boldsymbol{I}}(Z_k) \rangle - \gamma_k \|\phi_{*,\boldsymbol{I}}(Z_k)\| \\
&\overset{(2)}{=} \sum_{j=1}^c v_j(X; Z_{kj}, \gamma_{kj}),
\end{aligned}
\tag{76}
$$

where $\boldsymbol{I} = \{I, \cdots, I\}$. Here, we use a separate $\gamma_{kj}$ for the $k$-th component space. The above comes from the following.

(1) Thm. 3.1.

(2) $Z_k = (Z_{k1} \in T_{P_1}\text{Cor}^+(n), \cdots, Z_{kn} \in T_{P_n}\text{Cor}^+(n))$, $[Z_k] = \left\{ \frac{Z_{k1}}{\|Z_{k1}\|_I}, \cdots, \frac{Z_{kc}}{\|Z_{kc}\|_I} \right\}$.

**Correlation FC layer.** Following Lem. I.2 and Eq. (76), the FC layer $\mathcal{F}(\cdot) : (\text{Cor}^+(n))^c \to \text{Cor}^+(m)$ for the input $\boldsymbol{X}$ is

$$
Y = \phi^{-1}\left( \sum_{i=1}^{d_m} \sum_{j=1}^c v_{ij}(X_j; Z_{ij}, \gamma_{ij})e_i \right),
\tag{77}
$$

where $Z_{ij} \in \text{Hol}(n)$ and $\gamma_{ij} \in \mathbb{R}$.

Eq. (77) implies that $\mathcal{F}(\cdot) : (\text{Cor}^+(n))^c \to \text{Cor}^+(m)$ differs from $\mathcal{F}(\cdot) : \text{Cor}^+(n) \to \text{Cor}^+(m)$ only in $v_{ij}$, where the former is a summation. For example, considering the FC layer $\mathcal{F}(\cdot) : (\text{Cor}^+(n))^c \to \text{Cor}^+(m)$ under ECM, its $v_{ij}$ for the input $\boldsymbol{C} = \{C_j \in \text{Cor}^+(n)\}_{k=1}^c$ is

$$
v_{ij}(\boldsymbol{C}) = \sum_{k=1}^c v_{ijk}^{\text{EC}}(C_k, Z_{ijk}, \gamma_{ijk}),
\tag{78}
$$

where $Z_{ijk} \in \text{Hol}(n)$ and $\gamma_{ijk} \in \mathbb{R}$, for $i, j = 1, \cdots, m$ with $i > j$, and $1 \leq k \leq c$.

## E. Backpropagation over Correlation Geometries

Except for $\mathcal{D}$ and $\mathcal{D}^\star$, all computations involved in the five metrics can be backpropagated using existing techniques or PyTorch's auto-differentiation. Three kinds of matrix computations need to be discussed: 1) matrix logarithm and exponentiation; 2) Cholesky decomposition; 3) $\mathcal{D}$ and $\mathcal{D}^\star$.

**Matrix Logarithm and Exponentiation:** The symmetric matrix exp and log, *i.e.*, $\log : \mathcal{S}_{++}^n \to \mathcal{S}^n$ and $\exp : \mathcal{S}^n \to \mathcal{S}_{++}^n$, can be backpropagated using the Daleckii-Krein formula (Brooks et al., 2019, Eq. 13).

**Cholesky Decomposition:** The backpropagation of the Cholesky decomposition has been well studied by Murray (2016). In addition, as shown by Chen et al. (2024b, App. F), the one in Murray (2016) yields a similar gradient to the one generated by the autograd of `torch.linalg.cholesky`.

$\mathcal{D}$ **and** $\mathcal{D}^\star$**:** Their gradients can be backpropagated either approximately by the ones of their iterative algorithms or accurately by our following two propositions.

**Proposition E.1** (Gradients w.r.t. $\mathcal{D}$). [↓] *Let $l(\cdot)$ be the loss function and $Y = \mathcal{D}(H) + H : \text{Hol}(n) \to \mathcal{S}^n$ for any symmetric hollow matrix $H$, where $\mathcal{S}^n$ is the Euclidean space of $n \times n$ symmetric matrices. Let $Y = U\Delta U^\top$ be the eigendecomposition with $(\delta_1, \cdots, \delta_n)$ as eigenvalues. Given the succeeding gradient $\frac{\partial l}{\partial Y}$, the output gradient $\frac{\partial l}{\partial H}$ is*

$$
\frac{\partial l}{\partial H} = \text{off}\left( \frac{\partial l}{\partial Y} - \exp_{*,Y}\left( \mathbb{D}\left( (H^0)^{-1}\text{Dv}\left( \frac{\partial l}{\partial Y} \right) \mathbf{1}^T \right) \right) \right),
\tag{79}
$$

*with $\mathcal{S}_{++}^n \ni H_{il}^0 = \sum_{j,k} U_{ij}U_{ik}U_{lj}U_{lk}L_{j,k}$ and*

$$
L_{j,k} = \begin{cases} \frac{\exp(\delta_j) - \exp(\delta_k)}{\delta_j - \delta_k}, & \text{if } \delta_j \neq \delta_k \\ \exp'(\delta_j), & \text{otherwise} \end{cases}
\tag{80}
$$

*Table 11.* Summary of $v_k(C)$ in $c$-class correlation MLR layers, where $k$ denotes the $k$-th class and $C \in \text{Cor}^+(n)$ is the input correlation matrix.

| Metric | Expression | Parameters | Refs |
|--------|-----------|-----------|------|
| ECM | $\langle \lfloor \Theta(C) \rfloor, \lfloor Z_k \rfloor \rangle - \gamma_k \lVert \lfloor Z_k \rfloor \rVert$ | $\{Z_k \in \text{Hol}(n), \gamma_k \in \mathbb{R}\}_{k=1}^c$ | Thm. 3.3 |
| LECM | $\langle \log \circ \Theta(C), \lfloor Z_k \rfloor \rangle - \gamma_k \lVert \lfloor Z_k \rfloor \rVert$ | $\{Z_k \in \text{Hol}(n), \gamma_k \in \mathbb{R}\}_{k=1}^c$ | Thm. 3.3 |
| OLM | $\langle \text{Log}^\circ(C), Z_k \rangle - \gamma_k \lVert Z_k \rVert$ | $\{Z_k \in \text{Hol}(n), \gamma_k \in \mathbb{R}\}_{k=1}^c$ | Thm. 3.3 |
| LSM | $\langle \text{Log}^\star(C), \text{Log}^\star_{*,I}(Z_k) \rangle - \gamma_k \lVert \text{Log}^\star_{*,I}(Z_k) \rVert$ | $\{Z_k \in \text{Hol}(n), \gamma_k \in \mathbb{R}\}_{k=1}^c$ | Thm. 3.3 |
| PHCM | $\Psi \circ \text{Chol} \to \beta\text{-concat} \to \text{Poincaré } v_k(x)$ | $\left\{ z_k \in \mathbb{R}^{\frac{n(n-1)}{2}}, \gamma_k \in \mathbb{R} \right\}_{k=1}^c$ | Eqs. (14), (65) and (67) |

*Here,* $\mathbb{D}(\cdot) : \mathbb{R}^{n \times n} \to \text{Diag}(n)$ *extracts the diagonal matrix, while* $\text{Dv}(\cdot) : \mathbb{R}^{n \times n} \to \mathbb{R}^n$ *returns a vector of diagonal elements. Besides,* $\text{off}(\cdot)$ *subtracts the diagonal matrix from a matrix, and* $\exp_{*,Y}$ *is the differential of the symmetric matrix exponential:*

$$\exp_{*,Y}(V) = U \left( L \odot \left( U^\top V U \right) \right) U^\top, \tag{81}$$

*where* $\odot$ *denotes the Hadamard product and* $L$ *is called the Loewner matrix, with the* $(j, k)$-*th element defined as Eq.* (80).

**Proposition E.2** (Gradients w.r.t. $\mathcal{D}^\star$). [↓] *Following the notation in Prop.* E.1, *we further denote* $\Sigma = \mathcal{D}^\star(C) C \mathcal{D}^\star(C) :$ $\text{Cor}^+(n) \to \text{Row}_1^+(n)$, *where* $\text{Row}_1^+(n)$ *is the manifold of* $n \times n$ *SPD matrices with unit row sum. Given the succeeding gradient* $\frac{\partial l}{\partial \Sigma}$, *the output gradient* $\frac{\partial l}{\partial C}$ *is*

$$\frac{\partial l}{\partial C} = \Delta \left( \frac{\partial l}{\partial \Sigma} - \left( (I + \Sigma)^{-1} \widetilde{v} \mathbf{1}^\top \right)_{\text{sym}} \right) \Delta, \tag{82}$$

*where* $\Delta = \mathbb{D}(\Sigma)^{1/2}$, $\widetilde{v} = \text{Dv} \left( \Sigma \frac{\partial l}{\partial \Sigma} + \frac{\partial l}{\partial \Sigma} \Sigma \right)$, $I$ *is the identity matrix, and* $\mathbf{1} \in \mathbb{R}^n$ *is the vector with all entities as 1. Here,* $(A)_{\text{sym}} = \frac{A + A^\top}{2}$.

## F. Order-Invariance of Beta Operations

**Theorem F.1** (Order-invariance). [↓] *Given multichannel data* $x_{i_1, \ldots, i_n} \in \mathbb{P}^{n_{i_n}}$ *with* $i_j \in \{1, \ldots, N_j\}$, *applying the* $\beta$-*concatenation sequentially* $n$ *times in the order* $i_n \to \cdots \to i_1$ *is equivalent to a single* $\beta$-*concatenation along all indices simultaneously. Similarly,* $\beta$-*splitting* $x \in \mathbb{P}^N$ *into multichannel data* $x_{i_1, \ldots, i_n} \in \mathbb{P}^{n_{i_n}}$ *with* $i_j \in \{1, \ldots, N_j\}$ *and* $n_{i_n} \prod_{j=1}^n N_j = N$ *under the sequential order* $i_1 \to \cdots \to i_n$ *is identical to the one under a single* $\beta$-*split to generate all indices simultaneously.*

## G. Summary of Correlation FC and MLR Layers

Tab. 11 summarizes the $c$-class correlation MLR layers, while Algs. 1 and 2 provide the detailed algorithm. Tab. 12 summarizes our correlation FC layers, while Algs. 3 and 4 provide the detailed algorithm.

## H. Additional Details and Experiments

### H.1. Basic Layers in SPDNet

SPDNet (Huang & Van Gool, 2017) is the most classic SPD neural network. SPDNet mimics the conventional densely connected feedforward network, consisting of three basic building blocks:

$$\text{BiMap layer: } S^k = W^k S^{k-1} W^{k\top}, \text{ with } W^k \text{ semi-orthogonal}, \tag{83}$$

$$\text{ReEig layer: } S^k = U^{k-1} \max(\Sigma^{k-1}, \epsilon I_n) U^{k-1\top}, \text{ with } S^{k-1} \overset{\text{SVD}}{=} U^{k-1} \Sigma^{k-1} U^{k-1\top}, \tag{84}$$

$$\text{LogEig layer: } S^k = \log(S^{k-1}). \tag{85}$$

*Table 12.* Summary of correlation FC layers, $\mathcal{F} : \mathrm{Cor}^+(n) \ni C \mapsto Y \in \mathrm{Cor}^+(m)$. Each $v_{ij}^g$ with $g \in \{\mathrm{EC}, \mathrm{LEC}, \mathrm{OL}, \mathrm{LS}\}$ is defined by Tab. 11.

| Metric | Expression | Parameters | Refs |
|---|---|---|---|
| ECM | $Y = \mathrm{Cor} \circ \mathrm{Chol}^{-1}\left(V^{\mathrm{EC}} + I_m\right)$ 
 $V_{ij}^{\mathrm{EC}} = \begin{cases} v_{ij}^{\mathrm{EC}}(C), & i > j, \\ 0, & \text{otherwise} \end{cases}$ | $\{Z_{ij} \in \mathrm{Hol}(n), \gamma_{ij} \in \mathbb{R}\}_{1 \le j < i \le m}$ | Thm. 3.5 |
| LECM | $Y = \mathrm{Cor} \circ \mathrm{Chol}^{-1} \circ \exp\left(V^{\mathrm{LEC}}\right)$ 
 $V_{ij}^{\mathrm{LEC}} = \begin{cases} v_{ij}^{\mathrm{LEC}}(C), & i > j, \\ 0, & \text{otherwise} \end{cases}$ | $\{Z_{ij} \in \mathrm{Hol}(n), \gamma_{ij} \in \mathbb{R}\}_{1 \le j < i \le m}$ | Thm. 3.5 |
| OLM | $Y = \mathrm{Exp}^{\circ}\left(V^{\mathrm{OL}}\right)$ 
 $V_{ij}^{\mathrm{OL}} = \begin{cases} v_{ij}^{\mathrm{OL}}(C)/\sqrt{2}, & i > j, \\ V_{ji}^{\mathrm{OL}}, & i < j, \\ 0, & \text{otherwise} \end{cases}$ | $\{Z_{ij} \in \mathrm{Hol}(n), \gamma_{ij} \in \mathbb{R}\}_{1 \le j < i \le m}$ | Thm. 3.5 |
| LSM | $Y = \mathrm{Cor} \circ \exp\left(V^{\mathrm{LS}}\right)$ 
 $V_{ij}^{\mathrm{LS}} = \begin{cases} v_{ij}^{\mathrm{LS}}(C)/\sqrt{6}, & m > i > j \ge 1, \\ v_{ii}^{\mathrm{LS}}(C)/\sqrt{3}, & m > i \ge 1, \\ V_{ji}^{\mathrm{LS}}, & i < j, \\ -\sum_{k=1}^{m-1} V_{kj}^{\mathrm{LS}}, & i = m, 1 \le j < m, \\ \sum_{k=1}^{m-1}\sum_{l=1}^{m-1} V_{lk}^{\mathrm{LS}}, & i = j = m \end{cases}$ | $\{Z_{ij} \in \mathrm{Hol}(n), \gamma_{ij} \in \mathbb{R}\}_{1 \le j \le i \le m-1}$ | Thm. 3.5 |
| PHCM | $\Psi \circ \mathrm{Chol} \to \beta\text{-concat} \to \text{Poincaré FC}$ 
 $\beta\text{-split} \to (\Psi \circ \mathrm{Chol})^{-1}$ | $\left\{ z_k \in \mathbb{R}^{\frac{n(n-1)}{2}}, \gamma_k \in \mathbb{R} \right\}_{k=1}^{m(m-1)/2}$ | Eqs. (14), (66) and (67) |

where $\max()$ is element-wise maximization and $\log$ is the matrix logarithm. BiMap and ReEig mimic transformation and non-linear activation, where the input and output are both SPD matrices. LogEig maps SPD matrices into the tangent space at the identity matrix for classification.

*Remark* H.1. All three basic layers in SPDNet are designed for the SPD manifold. Although correlation is still SPD, it has its own geometries. Therefore, applying SPD networks, such as SPDNet, to correlation inputs might bring suboptimal performance, which motivates us to develop correlation networks based on correlation geometries.

## H.2. Datasets

The following introduces the details of each dataset.

**Radar** (Brooks et al., 2019).[3] It consists of 3,000 synthetic radar signals equally distributed in 3 classes.

**HDM05** (Müller et al., 2007).[4] It consists of 2,343 skeleton-based motion capture sequences executed by different actors. Each frame consists of 3D coordinates of 31 joints. We remove the under-represented clips, trimming the dataset down to 2,326 instances scattered throughout 122 classes. We randomly select 50% of the samples from each category for training and the remaining 50% for testing.

**FPHA** (Garcia-Hernando et al., 2018).[5] It includes 1,175 skeleton-based first-person hand gesture videos of 45 different categories with 600 clips for training and 575 for testing. Each frame contains the 3D coordinates of 21 hand joints.

**NTU120**[6] **(Liu et al., 2019).** This data set contains 114,480 sequences in 120 action classes. We use mutual actions and adopt the cross-setup protocol (Liu et al., 2019).

For the HDM05 and FPHA datasets, we preprocess each sequence using the code[7] provided by Vemulapalli et al. (2014)

---

[3] https://www.dropbox.com/s/dfnlx2bnyh3kjwy/data.zip?dl=0
[4] https://resources.mpi-inf.mpg.de/HDM05/
[5] https://github.com/guiggh/hand_pose_action
[6] https://github.com/shahroudy/NTURGB-D
[7] https://ravitejav.weebly.com/kbac.html

---

**Algorithm 1** Log-Euclidean correlation MLR

---

**Input**           : correlation matrix $C \in \mathrm{Cor}^+(n)$, number of classes $c$, and Log-Euclidean metric $g \in \{\mathrm{EC}, \mathrm{LEC}, \mathrm{OL}, \mathrm{LS}\}$.
**Parameters :** class weights $\{Z_k \in \mathrm{Hol}(n)\}_{k=1}^c$ and class biases $\{\gamma_k \in \mathbb{R}\}_{k=1}^c$.
**Output**        : class probabilities $p \in \mathbb{R}^c$.

```
// In practice, the following is efficiently implemented as tensor operations in
   PyTorch rather than a for-loop.
```
**for** $k \leftarrow 1$ **to** $c$ **do**
    **switch** $g$ **do**
        **case** EC **do**
          $v_k(C) \leftarrow \langle \lfloor \Theta(C) \rfloor, \lfloor Z_k \rfloor \rangle - \gamma_k \|\lfloor Z_k \rfloor\|$;          `// Thm. 3.5`
        **end**
        **case** LEC **do**
          $v_k(C) \leftarrow \langle \log \circ \Theta(C), \lfloor Z_k \rfloor \rangle - \gamma_k \|\lfloor Z_k \rfloor\|$;       `// Thm. 3.5`
        **end**
        **case** OL **do**
          $v_k(C) \leftarrow \langle \mathrm{Log}^\circ(C), Z_k \rangle - \gamma_k \|Z_k\|$;              `// Thm. 3.5`
        **end**
        **case** LS **do**
          $v_k(C) \leftarrow \langle \mathrm{Log}^\star(C), \mathrm{Log}^\star_{*,I}(Z_k) \rangle - \gamma_k \|\mathrm{Log}^\star_{*,I}(Z_k)\|$;  `// Thm. 3.5`
        **end**
    **end**
**end**
$p = \mathrm{softmax}(v_1(C), \ldots, v_c(C))$

---

**Algorithm 2** PHCM correlation MLR

---

**Input**           : correlation matrix $C \in \mathrm{Cor}^+(n)$ and number of classes $c$.
**Parameters :** Poincaré MLR weights $\{z_k \in \mathbb{R}^N\}_{k=1}^c$ and biases $\{\gamma_k \in \mathbb{R}\}_{k=1}^c$, where $N = n(n-1)/2$.
**Output**        : class probabilities $p \in \mathbb{R}^c$.

$\{p_j\}_{j=1}^{n-1} \leftarrow \Psi \circ \mathrm{Chol}(C)$ ;               `// map to `$\mathbb{PP}^{n-1}$` by Eq. (14)`
$x \leftarrow \beta\text{-concat}\left(\{p_j\}_{j=1}^{n-1}\right)$ ;           `// Poincaré `$\beta$`-concat in Eq. (67)`
$p \leftarrow \mathrm{Poincaré\ MLR}\left(x; \{z_k, \gamma_k\}_{k=1}^c\right)$ ;        `// Poincaré MLR in Eq. (65)`

---

to normalize body part lengths and ensure invariance to scale and view. For NTU120, we follow Chen et al. (2021) to preprocess the data.

### H.3. Input Data

#### H.3.1. SPD INPUT IN SPD NETWORKS

For GyroLE, GyroAI, GyroLC, and GyroSPD++, inputs are similar to our CorNets, except that inputs are the SPD covariance matrices. For other SPD baselines, such as SPDNet, SPDNetBN, LieBN, MLR, and RResNet, each sequence is represented by a global covariance matrix as their original papers (Huang & Van Gool, 2017; Brooks et al., 2019; Chen et al., 2024b;a; Katsman et al., 2024). The sizes of the covariance matrices are $20 \times 20$, $93 \times 93$, $63 \times 63$, and $150 \times 150$ on the Radar, HDM05, FPHA, and NTU120 datasets, respectively.

#### H.3.2. GRASSMANNIAN INPUT IN GRASSMANNIAN NETWORKS

For GrNet, GyroGr, and GyroGr-Scaling baselines, each sequence is represented by an 8-channel Grassmannian tensor as their original papers (Huang et al., 2018; Nguyen & Yang, 2023). The sizes of the Grassmannian matrices are $8 \times 20 \times 8$, $8 \times 93 \times 10$, $8 \times 63 \times 10$ and $8 \times 150 \times 10$ on the Radar, HDM05, FPHA, and NTU120 datasets, respectively.

---

**Algorithm 3** Log-Euclidean correlation FC layer

---

**Input**         : input correlation $C \in \mathrm{Cor}^+(n)$, output size $m$, and Log-Euclidean metric $g \in \{\mathrm{EC}, \mathrm{LEC}, \mathrm{OL}, \mathrm{LS}\}$.
**Parameters :** FC parameters $\{Z_{ij} \in \mathrm{Hol}(n), \gamma_{ij} \in \mathbb{R}\}$ with valid index pairs $(i, j)$ as in Tab. 12.
**Output**        : output correlation $Y \in \mathrm{Cor}^+(m)$.

**switch** $g$ **do**

    **case** EC **do**
        $\mid$ $Y \leftarrow \mathrm{Cor} \circ \mathrm{Chol}^{-1}\left(V^{\mathrm{EC}} + I_m\right)$ ;                       `// VEC:  Eqs. (8) and (10)`
    **end**
    **case** LEC **do**
        $\mid$ $Y \leftarrow \mathrm{Cor} \circ \mathrm{Chol}^{-1} \circ \exp\left(V^{\mathrm{LEC}}\right)$ ;                   `// VLEC: Eqs. (8) and (11)`
    **end**
    **case** OL **do**
        $\mid$ $Y \leftarrow \mathrm{Exp}^{\circ}\left(V^{\mathrm{OL}}\right)$ ;                             `// VOL:  Eqs. (8) and (12)`
    **end**
    **case** LS **do**
        $\mid$ $Y \leftarrow \mathrm{Cor} \circ \exp\left(V^{\mathrm{LS}}\right)$ ;                          `// VLS:  Eqs. (8) and (13)`
    **end**

**end**

---

**Algorithm 4** PHCM correlation FC layer

---

**Input**         : input correlation $C \in \mathrm{Cor}^+(n)$ and output size $m$.
**Parameters :** Poincaré FC weights $\{z_k \in \mathbb{R}^N\}_{k=1}^d$ and biases $\{\gamma_k \in \mathbb{R}\}_{k=1}^d$, where $N = n(n-1)/2$ and $d = m(m-1)/2$.
**Output**        : output correlation $Y \in \mathrm{Cor}^+(m)$.

$\{p_j\}_{j=1}^{n-1} \leftarrow \Psi \circ \mathrm{Chol}(C)$ ;                            `// map to PPⁿ⁻¹ by Eq. (14)`
$x \leftarrow \beta\text{-concat}\left(\{p_j\}_{j=1}^{n-1}\right)$ ;                     `// Poincaré β-concat in Eq. (67)`
$y \leftarrow \text{Poincaré FC}\left(x; \{z_k, \gamma_k\}_{k=1}^d\right)$ ;                         `// Eq. (66)`
$\{q_j\}_{j=1}^{m-1} \leftarrow \beta\text{-split}(y)$ ;           `// Poincaré β-split, inverse of Eq. (67)`
$Y \leftarrow \left(\Psi \circ \mathrm{Chol}\right)^{-1}\left(\{q_j\}_{j=1}^{m-1}\right)$ ;                  `// the inverse of Eq. (14)`

---

### H.3.3. CORRELATION INPUT IN CORNETS

For the input of our CorNets, we first follow Wang et al. (2024); Nguyen et al. (2024) to model each sample into a multi-channel SPD tensor. Then, each SPD matrix is transformed to their correlation matrix by

$$\mathrm{Cor} : \mathcal{S}_{++}^n \ni \Sigma \longmapsto C = \mathbb{D}(\Sigma)^{-\frac{1}{2}} \Sigma \mathbb{D}(\Sigma)^{-\frac{1}{2}} \in \mathrm{Cor}^+(n). \tag{86}$$

The following introduces the SPD modeling.

For the HDM05 and FPHA datasets, we follow Nguyen & Yang (2023) to model each skeleton sequence into a multi-channel covariance tensor $[c, n, n]$. Specifically, we first identify the closest left (right) neighbor of every joint based on their distance to the hip (wrist) joint, and then combine the 3D coordinates of each joint and those of its left (right) neighbor to create a feature vector for the joint. For a given frame $t$, we compute its Gaussian embedding (Lovrić et al., 2000):

$$Y_t = \left(\det \Sigma_t\right)^{-\frac{1}{n+1}} \begin{bmatrix} \Sigma_t + \mu_t \left(\mu_t\right)^T & \mu_t \\ \left(\mu_t\right)^T & 1 \end{bmatrix}, \tag{87}$$

where $\mu_t$ and $\Sigma_t$ are the mean vector and covariance matrix computed from the set of feature vectors within the frame. The lower part of matrix $\log\left(Y_t\right)$ is flattened to obtain a vector $\tilde{v}_t$. All vectors $\tilde{v}_t$ within a time window $[t, t + c - 1]$, where $c$ is determined from a temporal pyramid representation of the sequence (the number of temporal pyramids is set to 2 in our experiments), are used to compute a covariance matrix as

$$\widetilde{\Sigma}_t = \frac{1}{c} \sum_{i=t}^{t+c-1} \left(\tilde{v}_i - \overline{v}_t\right)\left(\tilde{v}_i - \overline{v}_t\right)^T, \tag{88}$$

where $\overline{v}_t = \frac{1}{c} \sum_{i=t}^{t+c-1} \tilde{v}_i$. The resulting $\{\widetilde{\Sigma}_t\}$ are the covariance matrices that we need. On the FPHA dataset, we generate the covariance based on three sets of neighbors: left, right, and vertical (bottom) neighbors.

For the Radar dataset, we follow Wang et al. (2024) to use the temporal convolution followed by a covariance pooling layer to obtain a multi-channel covariance tensor of shape $[c, 20, 20]$.

After preprocessing, the input correlation tensor shapes are $[7, 20, 20]$, $[3, 28, 28]$, $[9, 28, 28]$ and $[6, 28, 28]$ on the Radar, HDM05, FPHA, and NTU120 datasets, respectively.

## H.4. Implementation Details

*Table 13.* Hyer-parameters in CorNets

| Dataset | Model | Optimizer | lr | wd | Matrix Power | Converged Epoch |
|---------|-------|-----------|-----|-----|--------------|-----------------|
| Radar | CorNet-ECM | ADAM | $1e^{-2}$ | N/A | 1.5 | 50 |
| | CorNet-LECM | ADAM | $1e^{-2}$ | N/A | -0.25 | 50 |
| | CorNet-OLM | ADAM | $1e^{-2}$ | N/A | -0.25 | 50 |
| | CorNet-LSM | ADAM | $1e^{-2}$ | N/A | 0.75 | 50 |
| | CorNet-PHCM | ADAM | $1e^{-2}$ | N/A | 0.75 | 50 |
| HDM05 | CorNet-ECM | ADAM | $1e^{-3}$ | $1e^{-3}$ | 0.125 | 100 |
| | CorNet-LECM | ADAM | $1e^{-4}$ | $1e^{-3}$ | 0.5 | 150 |
| | CorNet-OLM | SGD | $5e^{-2}$ | $1e^{-3}$ | 0.25 | 200 |
| | CorNet-LSM | ADAM | $1e^{-3}$ | N/A | -0.75 | 50 |
| | CorNet-PHCM | ADAM | $1e^{-2}$ | N/A | -0.25 | 50 |
| FPHA | CorNet-ECM | ADAM | $5e^{-3}$ | N/A | -0.25 | 150 |
| | CorNet-LECM | ADAM | $5e^{-4}$ | $1e^{-4}$ | -0.5 | 150 |
| | CorNet-OLM | ADAM | $1e^{-4}$ | N/A | -1 | 50 |
| | CorNet-LSM | ADAM | $1e^{-3}$ | N/A | -1 | 50 |
| | CorNet-PHCM | ADAM | $1e^{-3}$ | $1e^{-4}$ | -0.5 | 150 |
| NTU120 | CorNet-ECM | SGD | $1e^{-2}$ | N/A | 0.25 | 50 |
| | CorNet-LECM | SGD | $1e^{-2}$ | N/A | 0.25 | 50 |
| | CorNet-OLM | SGD | $5e^{-3}$ | N/A | 0.25 | 50 |
| | CorNet-LSM | SGD | $1e^{-3}$ | N/A | 0.25 | 50 |
| | CorNet-PHCM | ADAM | $1e^{-3}$ | N/A | 0.25 | 50 |

**SPD baselines.** We follow the official Pytorch code of SPDNetBN[8] to implement SPDNet and SPDNetBN. For LieBN[9], we focus on the instantiation under Log-Cholesky Metric (LCM) (Lin, 2019), while for RResNet[10], we implement the ones induced by Affine-Invariant Metric (AIM) (Pennec et al., 2006) and Log-Euclidean Metric (LEM) (Arsigny et al., 2005). For SPD MLR[11], we implement the one based on LCM. Due to the lack of official code, Gyro-based models are carefully reimplemented from their original papers. Following Nguyen et al. (2024), GyroSPD++ combines an AIM-based convolution with an LEM-based MLR.

**Grassmannian baselines.** Since GrNet is officially implemented by Matlab, we carefully re-implemented it using PyTorch. Additionally, as both GryoGr and GryoGr-Scaling do not release official code, we re-implemented them based on the original paper (Nguyen & Yang, 2023). For all Grassmannian comparative methods, we use SGD (Robbins & Monro, 1951) with a learning rate of $5e^{-2}$.

**CorNets.** On all three datasets, we employ a single convolutional kernel for global convolution, *i.e.*, applying a global

---

[8]https://proceedings.neurips.cc/paper_files/paper/2019/file/6e69ebbfad976d4637bb4b39de261bf7-Supplemen
zip
[9]https://github.com/GitZH-Chen/LieBN
[10]https://github.com/CUAI/Riemannian-Residual-Neural-Networks
[11]https://github.com/GitZH-Chen/SPDMLR

receptive field across the channel dimension. The output dimensions of the correlation convolutional layer are $8 \times 8$, $26 \times 26$, $26 \times 26$, and $11 \times 11$ for the Radar, HDM05, FPHA, and NTU120 datasets, respectively.

We primarily use the Adam (Kingma, 2015) and SGD (Robbins & Monro, 1951) optimizers. Inspired by the deformation effect on the latent SPD geometries by the matrix power over the SPD manifold (Chen et al., 2024c, Fig. 1), we apply the matrix power before correlation modeling ($\mathrm{Cor}(\cdot)$) as activation. In particular, when the data are centered at zero and power is $-1$, $\mathrm{Cor}(\Sigma^{-1})$ corresponds to the partial correlation matrix of the covariance matrix $\Sigma$ (Thanwerdas, 2024, Lem. 1.6). The batch size is set to 30, and training is capped at 200 epochs, although most cases converge in fewer than 150 epochs. Due to the different correlation geometries, the hyperparameters vary for CorNets under different geometries. Tab. 13 summarize all the hyperparameters.

**Extra computational details for OLM and LSM layers.** For the MLR, FC, and convolutional layers induced by OLM and LSM, the key computations involve $\mathrm{Exp}^{\circ}$ and $\mathrm{Log}^{\star}$, which depend on the calculations of $\mathcal{D}$ and $\mathcal{D}^{\star}$. In our experiments, we empirically observe that iterating until convergence is more effective for $\mathcal{D}$, whereas a single step of Newton's method generally performs best for $\mathcal{D}^{\star}$. Accordingly, we set $\mathcal{D}$ to iterate until convergence, leveraging Prop. E.1 for accurate backpropagation. For $\mathcal{D}^{\star}$, we adopt a single iteration in Newton's method and use automatic differentiation (autograd) through this single step for backpropagation.

## H.5. Analysis of Covariance versus Correlation

In this section, we analyze when and why correlation matrices provide stronger representations than covariance matrices. App. H.5.1 quantifies the variability of diagonal variances via per-sample coefficients of variation, and App. H.5.2 compares the magnitudes of diagonal and off-diagonal entries via their ratios. These analyses lead to two insights: (1) large variability and magnitude of diagonal elements can act as nuisance noise for SPD networks by overshadowing informative off-diagonal correlations; (2) under such cases, correlation representations that normalize variances and emphasize pairwise correlations tend to be more effective, which is especially evident on HDM05.

### H.5.1. COEFFICIENT OF VARIATION OF DIAGONAL VARIANCES

This section investigates why CorNets yield substantially larger gains over SPD networks on HDM05 compared to FPHA.

**Setup.** For each covariance matrix $\Sigma \in \mathcal{S}_{++}^n$ we extract the diagonal vector

$$v = (\Sigma_{11}, \ldots, \Sigma_{NN}). \tag{89}$$

We compute the coefficient of variation of $v$ as

$$\mathrm{CV} = \frac{\mathrm{std}(v)}{\mathrm{mean}(v) + \varepsilon}, \tag{90}$$

where $\varepsilon = 10^{-8}$ ensures numerical stability. As shown in App. H.3.3, each sequence is modeled as a $c$-channel tensor of covariance matrices. The above procedure yields one coefficient of variation per channel for each sample. We visualize their empirical distributions per channel.

**Analysis.** Figs. 6 and 7 show that the coefficients of variation w.r.t. diagonal variance are large on both datasets. On FPHA, most values fall between $0.8$ and $2.0$. On HDM05, they are even larger, typically between $1.0$ and $3.0$. Such large fluctuations indicate that diagonal variances change substantially and could bring nuisance noise for SPD networks. In contrast, correlation matrices allow CorNets to focus on pairwise relationships. This explains the consistent improvements over SPD networks and the larger gains on HDM05.

### H.5.2. RATIO OF DIAGONAL TO OFF-DIAGONAL ENTRIES IN COVARIANCE FEATURES

This section further examines why CorNets achieve larger gains over SPD networks on HDM05 than on FPHA. We analyze the ratio of diagonal to off-diagonal entries in covariance matrices on FPHA and HDM05, to quantify how strongly variance terms overshadow pairwise correlations.

**Setup.** For each covariance matrix $\Sigma \in \mathcal{S}_{++}^n$ we compute the mean magnitude of diagonal entries

$$D = \frac{1}{N} \sum_{i=1}^{N} |\Sigma_{ii}|, \tag{91}$$

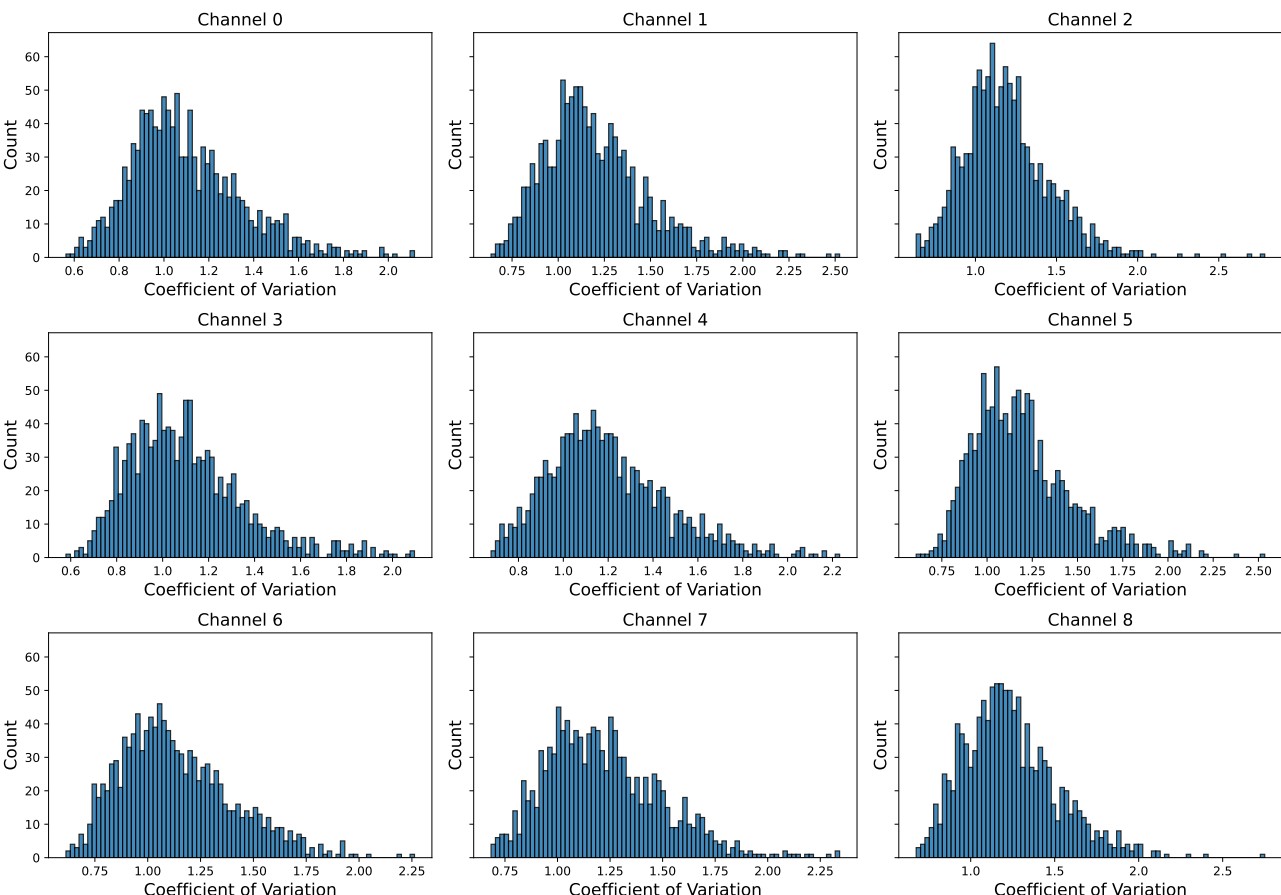

*Figure 6.* Distribution of per-sample coefficients of variation of diagonal variances on FPHA. Higher values indicate stronger diagonal variability, which could cause nuisance noise.

and the mean magnitude of off-diagonal entries

$$O = \frac{1}{N(N-1)} \sum_{i \neq j} |\Sigma_{ij}|. \tag{92}$$

We then form the sample-wise ratio

$$R = \frac{D}{O}, \tag{93}$$

which measures how much larger the diagonal amplitudes are compared to the off-diagonal correlations. Each sample yields one ratio per channel, and we visualize the empirical distributions of these ratios on FPHA and HDM05.

**Analysis.** Figs. 8 and 9 show that both datasets have ratios well above one. On FPHA, most ratios lie between 1.7 and 3.0, indicating that diagonal amplitudes are noticeably larger than off-diagonal correlations. HDM05 exhibits even larger ratios, typically between 2.0 and 6.0, with many above 3.0. These statistics indicate that covariance representations on both datasets are strongly dominated by diagonal entries, with more pronounced dominance on HDM05. When diagonal terms dominate, SPD networks trained on covariance inputs tend to overemphasize variances and underexploit informative pairwise correlations. Correlation matrices normalize variances and highlight off-diagonal interactions, which explains why CorNets outperform SPD baselines on both datasets and why the improvement is substantially larger on HDM05.

### H.6. Normalized Covariance vs. Correlation

**Setup.** We evaluate SPD-based baselines by covariance inputs normalized by their largest eigenvalue. Given a covariance matrix $\Sigma$, we get the normalized SPD input $\widehat{\Sigma} = \Sigma/\lambda_{\max}(\Sigma)$ and feed it into existing SPD networks. This variant is

*Figure 7.* Distribution of per-sample coefficients of variation of diagonal variances on HDM05. Higher values indicate stronger diagonal variability, which could cause nuisance noise.

denoted by "-EigN". We report results on the Radar, HDM05, and FPHA datasets for representative SPD models: SPDNet, SPDNetBN, SPDResNet, SPDNetLieBN, SPDNetMLR, GyroAI, and GyroSPD++. Here, SPDResNet is implemented under the LEM, while SPDNetLieBN follows the LCM.

**Results.** Tab. 14 summarizes the results. On HDM05, eigenvalue normalization has only a marginal effect and the normalized variants achieve accuracy comparable to their unnormalized counterparts. On FPHA and, in particular, on Radar, normalization usually reduces accuracy. The behavior of GyroSPD++ is especially informative. GyroSPD++ and CorNet share similar architecture, consisting of one convolution followed by a MLR layer. However, GyroSPD++-EigN performs worse than GyroSPD++ on all three datasets, while CorNet with correlation inputs achieves clear improvements over GyroSPD++. These phenomena can be explained by two factors.

1. *Redundancy.* The raw samples on HDM05 and FPHA have already undergone centering, scaling, and normalization before covariance modeling. Dividing by $\lambda_{\max}(\Sigma)$ therefore introduces little additional control over scale, which explains the marginal effect on HDM05.

2. *Scaled covariance versus correlation.* Since EigN is equivalent to uniformly rescaling the raw samples before covariance computation, the normalized covariance matrices remain covariances and do not encode new statistical information. Moreover, forcing the largest eigenvalue to 1 can remove potentially informative differences in overall energy across samples, which aligns with the degradation observed for EigN variants, especially GyroSPD++-EigN. In contrast, correlation normalization uses a different scaling factor for each pair of variables,

$$\text{Cor}_{ij} = \frac{\Sigma_{ij}}{\sqrt{\Sigma_{ii}\Sigma_{jj}}}, \tag{94}$$

producing standardized correlation coefficients. Therefore, global eigenvalue scaling is statistically distinct from correlation normalization and fails to capture the benefits of explicit correlation modeling.

### H.7. Ablations on Activations

In the main experiments, we follow HNN++ (Shimizu et al., 2021) and GyroSPD++ (Nguyen et al., 2024), and do not use explicit activations, as the manifold itself introduces nonlinearity. We further conduct an ablation on activations. Following Ganea et al. (2018, Sec. 3.2), we define activations in the tangent space at the identity, *i.e.*, $\text{Exp}_I \circ \delta \circ \text{Log}_I$ for four Log-Euclidean metrics, and $\text{Exp}_0 \circ \delta \circ \text{Log}_0$ for PHCM in the $\beta$-concatenated Poincaré vector, where $\delta$ is ReLU (Glorot et al., 2011). Specifically, we insert a ReLU after the correlation convolution. As shown in Tab. 15, adding activations generally yields no benefits and can even degrade performance. The variant without activation consistently achieves higher or comparable accuracy, except CorNet-OLM for HDM05. Moreover, CorNet-LSM with activation fails to converge on HDM05 and FPHA. These results suggest that CorNet already provides sufficient nonlinearity, rendering additional activations redundant.

### H.8. Scalability of Correlation Metrics

We evaluate the computational efficiency of correlation metrics across increasing input dimensions using CorNet with one correlation FC layer followed by one correlation MLR layer. Each input correlation matrix of size $[n, n]$ is mapped

*Table 14.* SPD networks with or without normalized SPD inputs

| Manifold | Method | Radar | HDM05 | FPHA |
|---|---|---|---|---|
| $\mathcal{S}^n_{++}$ | SPDNet | 93.25 ± 1.10 | 64.57 ± 0.61 | 85.59 ± 0.72 |
| | SPDNet-EigN | 86.91 ± 0.57 | 66.62 ± 0.73 | 84.90 ± 0.62 |
| | SPDNetBN | 94.85 ± 0.99 | 71.28 ± 0.79 | 89.33 ± 0.49 |
| | SPDNetBN-EigN | 89.25 ± 1.19 | 71.59 ± 0.68 | 88.47 ± 0.39 |
| | SPDResNet | 95.89 ± 0.86 | 70.12 ± 2.45 | 85.07 ± 0.99 |
| | SPDResNet-EigN | 92.61 ± 0.96 | 71.02 ± 0.91 | 84.53 ± 0.46 |
| | SPDNetLieBN | 94.80 ± 0.71 | 71.78 ± 0.44 | 86.33 ± 0.43 |
| | SPDNetLieBN-EigN | 88.91 ± 1.21 | 70.61 ± 1.04 | 83.73 ± 0.65 |
| | SPDNetMLR | 95.64 ± 0.83 | 65.90 ± 0.93 | 85.67 ± 0.69 |
| | SPDNetMLR-EigN | 89.41 ± 0.58 | 66.89 ± 0.63 | 83.63 ± 1.09 |
| | GyroAI | 96.29 ± 0.48 | 72.34 ± 1.06 | 89.60 ± 0.37 |
| | GyroAI-EigN | 91.36 ± 0.80 | 72.64 ± 0.70 | 89.90 ± 0.31 |
| | GyroSPD++ | 95.20 ± 0.88 | 69.82 ± 1.79 | 89.50 ± 0.37 |
| | GyroSPD++-EigN | 90.83 ± 1.09 | 66.92 ± 0.28 | 84.29 ± 0.14 |
| $\mathrm{Cor}^+(n)$ | CorNet-ECM | 97.71 ± 0.61 | 81.35 ± 1.27 | **92.17 ± 0.49** |
| | CorNet-LECM | **98.40 ± 0.70** | 78.05 ± 1.14 | 91.17 ± 0.32 |
| | CorNet-OLM | 97.57 ± 0.76 | 81.46 ± 0.61 | 91.63 ± 0.12 |
| | CorNet-LSM | 96.24 ± 1.48 | 74.89 ± 1.07 | 83.43 ± 0.65 |
| | CorNet-PHCM | 96.56 ± 0.86 | **82.26 ± 0.92** | 90.03 ± 0.63 |

*Table 15.* Comparison of CorNet with or without activations.

| Metric | Activation | Radar | HDM05 | FPHA |
|---|---|---|---|---|
| ECM | ReLU | 97.41 ± 0.25 | 81.23 ± 0.46 | 89.80 ± 0.58 |
| | None | **97.71 ± 0.61** | **81.35 ± 1.27** | **92.17 ± 0.49** |
| LECM | ReLU | 97.23 ± 0.67 | 77.51 ± 1.02 | 91.00 ± 0.15 |
| | None | **98.40 ± 0.70** | **78.05 ± 1.14** | **91.17 ± 0.32** |
| OLM | ReLU | 97.52 ± 0.47 | **81.86 ± 0.65** | 91.47 ± 0.19 |
| | None | **97.57 ± 0.76** | 81.46 ± 0.61 | **91.63 ± 0.12** |
| LSM | ReLU | 95.60 ± 0.97 | N/A | N/A |
| | None | **96.24 ± 1.48** | **74.89 ± 1.07** | **83.43 ± 0.65** |
| PHCM | ReLU | 96.40 ± 0.25 | 77.32 ± 1.56 | 88.63 ± 0.22 |
| | None | **96.56 ± 0.86** | **82.26 ± 0.92** | **90.03 ± 0.63** |

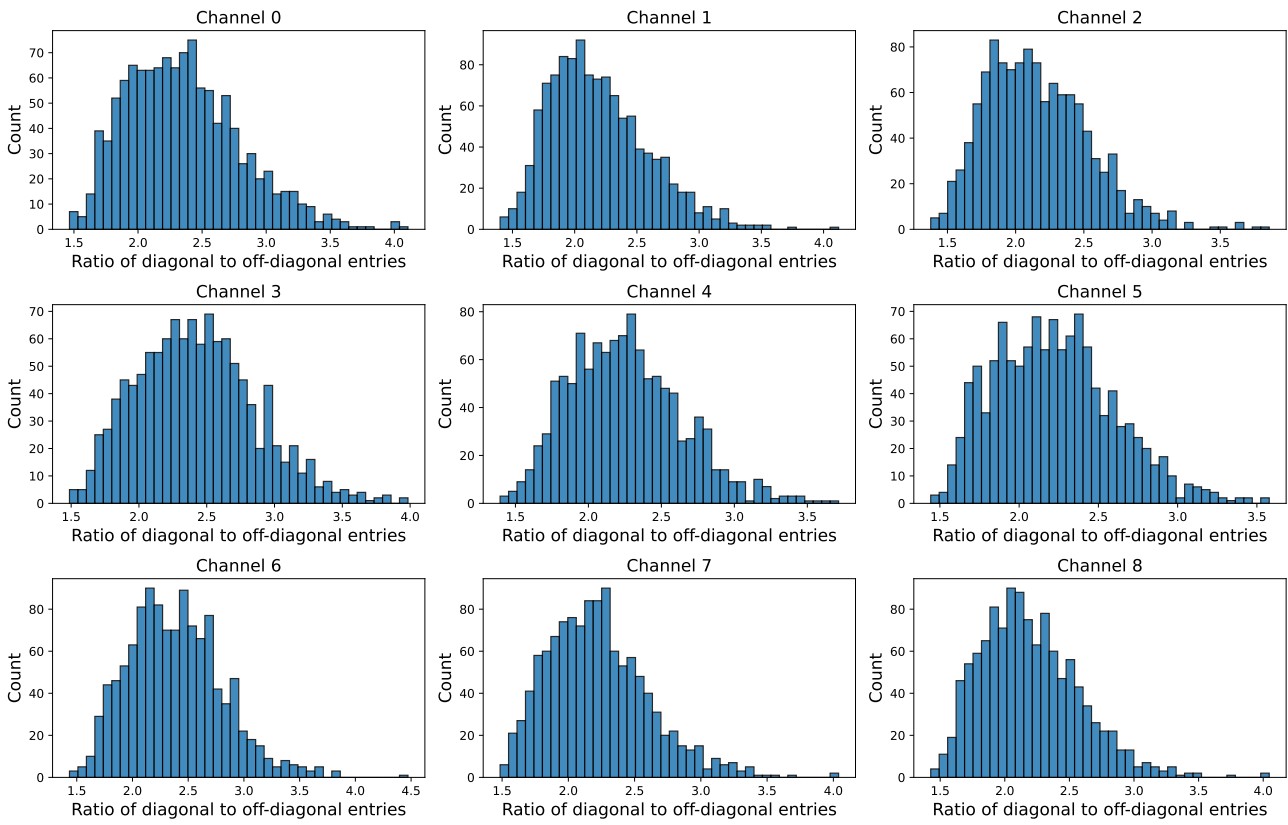

*Figure 8.* Distribution of ratios of diagonal to off-diagonal entries on FPHA.

to $[20, 20]$ by the FC layer and then classified into 10 classes by the MLR layer. For each $30 \leq n \leq 1000$, we randomly generate 30 correlation matrices and record the average runtime of a single forward pass. As implied by Tab. 8, the runtime is governed by two factors: the co-domain computation (Euclidean or hyperbolic) and the complexity of the diffeomorphism. The results are summarized in Tab. 16. We have the following findings.

- ECM is consistently the most efficient metric, benefiting from both a Euclidean co-domain and the simplest diffeomorphism.

- At low dimensions ($n \leq 400$), the ordering is

$$\text{ECM} < \text{LECM} < \text{OLM} \approx \text{LSM} < \text{PHCM}.$$

  Here, co-domain operations dominate, and PHCM is slowest due to costly hyperbolic computations.

- At high dimensions ($n \geq 700$), the ordering changes to

$$\text{ECM} < \text{PHCM} < \text{OLM} \approx \text{LSM} < \text{LECM}.$$

  Here, diffeomorphisms dominate: ECM and PHCM scale better thanks to relatively lightweight Cholesky decomposition, while OLM and LSM slow down due to matrix logarithm/exponentiation. LECM is the slowest, as its $\log \circ \Theta$ requires two nested matrix functions.

### H.9. Additional Details on Visualization

We provide additional interpretations on Figs. 2 and 5 by first describing how SPD and correlation matrices are visualized, then explaining the construction of Fig. 2, and finally clarifying how the decision hyperplanes in Fig. 5 are obtained.

*Figure 9.* Distribution of ratios of diagonal to off-diagonal entries on HDM05.

*Table 16.* Average runtime (s) of a single forward pass in CorNet under different metrics and input dimensions. The top two efficient metrics in each row are highlighted in **red** and **blue**, respectively.

| Dim | ECM | LECM | OLM | LSM | PHCM |
|-----|-----|------|-----|-----|------|
| 30 | **0.0004** | 0.0018 | **0.0012** | 0.0019 | 0.0131 |
| 50 | **0.0004** | **0.0027** | 0.0318 | 0.0334 | 0.0211 |
| 100 | **0.0008** | **0.0054** | 0.0764 | 0.0781 | 0.0413 |
| 150 | **0.0015** | **0.0100** | 0.1247 | 0.1267 | 0.2284 |
| 200 | **0.0025** | **0.0197** | 0.1906 | 0.1938 | 0.3320 |
| 250 | **0.0037** | **0.0345** | 0.2352 | 0.2379 | 0.4414 |
| 300 | **0.0053** | **0.0733** | 0.3434 | 0.3454 | 0.5732 |
| 400 | **0.0092** | **0.1796** | 0.5163 | 0.5261 | 0.4807 |
| 500 | **0.0143** | **0.3076** | 0.6907 | 0.6961 | 0.5693 |
| 600 | **0.0206** | **0.5983** | 0.9331 | 0.9484 | 0.7923 |
| 700 | **0.0289** | 1.0961 | 1.2432 | 1.2575 | **1.0417** |
| 800 | **0.039** | 1.8689 | 1.6658 | 1.6815 | **1.3387** |
| 900 | **0.0535** | 2.9886 | 2.2156 | 2.2303 | **1.7324** |
| 1000 | **0.0706** | 3.7259 | 2.539 | 2.5783 | **1.229** |

### H.9.1. VISUALIZATION OF LOW-DIMENSIONAL SPD AND CORRELATION MATRICES

Any $2 \times 2$ covariance matrix in $\mathcal{S}_{++}^2$ can be written as

$$\Sigma = \begin{pmatrix} a & b \\ b & d \end{pmatrix}, \qquad a > 0, \ d > 0, \ ad - b^2 > 0. \tag{95}$$

Embedding $\Sigma$ into $\mathbb{R}^3$ via the map $\Sigma \mapsto (a, b, d)$ identifies $\mathcal{S}_{++}^2$ with the interior of the quadratic cone

$$\left\{ (a, b, d) \in \mathbb{R}^3 \mid a > 0, \ d > 0, \ ad - b^2 > 0 \right\}, \tag{96}$$

which is an open cone in $\mathbb{R}^3$.

For $2 \times 2$ correlation matrices, any $C \in \mathrm{Cor}^+(2)$ has the form

$$C = \begin{pmatrix} 1 & r \\ r & 1 \end{pmatrix}, \qquad r \in (-1, 1). \tag{97}$$

Thus, $\mathrm{Cor}^+(2)$ is one dimensional. Embedding $C$ into $\mathbb{R}^3$ as $(1, r, 1)$ yields a line segment inside the cone corresponding to $\mathcal{S}_{++}^2$.

For $3 \times 3$ correlation matrices, any $C \in \mathrm{Cor}^+(3)$ is parameterized by its off-diagonal entries $(r_{12}, r_{13}, r_{23})$:

$$C = \begin{pmatrix} 1 & r_{12} & r_{13} \\ r_{12} & 1 & r_{23} \\ r_{13} & r_{23} & 1 \end{pmatrix}. \tag{98}$$

Embedding $C$ into $\mathbb{R}^3$ via $C \mapsto (r_{12}, r_{13}, r_{23})$ produces an open elliptope in $\mathbb{R}^3$. This is the representation of $\mathrm{Cor}^+(3)$ used in Fig. 5, where each point in the elliptope corresponds to one $3 \times 3$ correlation matrix.

### H.9.2. CONSTRUCTION OF FIG. 2

Given a covariance matrix $\Sigma \in \mathcal{S}_{++}^n$, its correlation matrix is defined in Sec. 2 as

$$C = \mathrm{Cor}(\Sigma) = \mathbb{D}(\Sigma)^{-1/2} \Sigma \mathbb{D}(\Sigma)^{-1/2}. \tag{99}$$

This map normalizes the diagonal entries and thus many covariance matrices share the same correlation. To see this explicitly in the $2 \times 2$ case, fix a correlation matrix

$$C = \begin{pmatrix} 1 & r \\ r & 1 \end{pmatrix}, \qquad r \in (-1, 1), \tag{100}$$

and consider any positive diagonal matrix

$$D = \mathrm{diag}(\lambda_1, \lambda_2), \qquad \lambda_1 > 0, \ \lambda_2 > 0. \tag{101}$$

The corresponding covariance matrix

$$\Sigma = DCD = \begin{pmatrix} \lambda_1^2 & \lambda_1 \lambda_2 r \\ \lambda_1 \lambda_2 r & \lambda_2^2 \end{pmatrix} \tag{102}$$

satisfies $\mathrm{Cor}(\Sigma) = C$. Since $\lambda_1$ and $\lambda_2$ can take any positive values, there are infinitely many $\Sigma \in \mathcal{S}_{++}^2$ that map to the same correlation $C$. When embedded into $\mathbb{R}^3$ as $(a, b, d)$, these $\Sigma$ form a two-dimensional surface lying inside the cone corresponding to $\mathcal{S}_{++}^2$. Fig. 2 visualizes this one-to-many relationship by plotting several such surfaces for different correlations, together with their corresponding points on the correlation manifold.

### H.9.3. CONSTRUCTION OF FIG. 5

For ECM, LECM, OLM, and LSM, the decision hyperplane in the correlation MLR is the Riemannian hyperplane in Eq. (63) specialized to $\mathcal{M} = \mathrm{Cor}^+(n)$:

$$H_{A,P} = \left\{ X \in \mathrm{Cor}^+(n) \mid \langle \mathrm{Log}_P(X), A \rangle_P = 0 \right\}, \qquad P \in \mathrm{Cor}^+(n), \ A \in T_P \mathrm{Cor}^+(n). \tag{103}$$

In Fig. 5, we focus on $\mathrm{Cor}^+(3)$ and visualize it as the open elliptope in $\mathbb{R}^3$ via the embedding

$$C \in \mathrm{Cor}^+(3) \longmapsto (C_{21}, C_{31}, C_{32}) \in \mathbb{R}^3. \tag{104}$$

Given a Log-Euclidean metric and parameters $(A, P)$, each correlation matrix $C$ is first mapped to the tangent space at $P$ by $\mathrm{Log}_P(C)$, and we evaluate the linear form $\langle \mathrm{Log}_P(C), A \rangle_P$. The set of points in the elliptope where this scalar equals zero corresponds to the decision hyperplane $H_{A,P}$ and is plotted as the separating surface.

For PHCM, the margin hyperplane is defined in the $\beta$-concatenated Poincaré embedding. Let $\Psi \circ \mathrm{Chol}$ be the diffeomorphism in Eq. (14) that maps $C \in \mathrm{Cor}^+(n)$ to the poly-Poincaré space $\mathbb{PP}^{n-1}$, and let $\beta$-concatenation be the Poincaré operation in Eq. (67). We define

$$\tilde{x}(X) = \beta\text{-concat}\big(\Psi \circ \mathrm{Chol}(X)\big) \in \mathbb{P}^N, \qquad N = \frac{n(n-1)}{2}, \tag{105}$$

and the PHCM hyperplane

$$H_{a,p} = \left\{ X \in \mathrm{Cor}^+(n) \mid \big\langle \mathrm{Log}_p(\tilde{x}(X)), a \big\rangle_p = 0 \right\}, \qquad p \in \mathbb{P}^N, \ a \in T_p \mathbb{P}^N. \tag{106}$$

Here $\mathbb{P}^N = \left\{ x \in \mathbb{R}^N \mid \|x\|^2 < 1 \right\}$ is the $N$-dimensional Poincaré ball. In Fig. 5, we first map each correlation matrix $C \in \mathrm{Cor}^+(3)$ to $\tilde{x}(C) \in \mathbb{P}^N$, apply the Poincaré logarithm $\mathrm{Log}_p$ at a reference point $p$, and then visualize the zero level set of the linear form $\big\langle \mathrm{Log}_p(\tilde{x}(C)), a \big\rangle_p$ as the PHCM decision hyperplane.

### H.10. Hardware

On the HDM05 and FPHA datasets, SPDNet, RResNet, SPDNetBN, SPDNetLieBN, and MLR require SVD operations on relatively large matrices, which are more efficiently executed on a CPU. As a result, these methods are implemented on a CPU, whereas all other cases are executed on a single A6000 GPU.

# I. Proofs

### I.1. Proof of Thm. 3.1

We first prove a lemma for MLRs on general isometric manifolds, of which this theorem is a specific case. Notably, the result and proof can be readily extended to the case where $\mathbb{R}^m$ is endowed with an arbitrary inner product.

**Lemma I.1** (Isometric Riemannian MLRs). *Given $m$-dimensional Riemannian manifolds $\left(\widetilde{\mathcal{M}}, g^{\widetilde{\mathcal{M}}}\right)$ and $(\mathcal{M}, g^{\mathcal{M}})$ with a Riemannian isometry $\phi : \widetilde{\mathcal{M}} \to \mathcal{M}$, their origins are $E \in \widetilde{\mathcal{M}}$ and $\phi(E) \in \mathcal{M}$. The Riemannian MLR over $\widetilde{\mathcal{M}}$ for the input $X \in \mathcal{M}$ of each class $k = 1, \cdots, C$ can be calculated by the one over $\mathcal{M}$:*

$$v_k^{\widetilde{\mathcal{M}}}(X; Z_k, \gamma_k) = v_k^{\mathcal{M}}(\phi(X); \phi_{*,E}(Z_k), \gamma_k), \tag{107}$$

*with $\gamma_k \in \mathbb{R}$, $Z_k \in T_E\widetilde{\mathcal{M}} \cong \mathbb{R}^m$, and $\phi_{*,E} : T_E\widetilde{\mathcal{M}} \to T_{\phi(E)}\mathcal{M}$ as the differential map. Here, $v_k^{\widetilde{\mathcal{M}}}$ and $v_k^{\mathcal{M}}$ are the specific realizations of Eq. (2) over $\widetilde{\mathcal{M}}$ and $\mathcal{M}$, respectively.*

*Proof.* We omit the subscript $k$ in $A_k$ and $P_k$ for simplicity. We denote $\widetilde{\Gamma}$, $\widetilde{\mathrm{Log}}$, $\langle \cdot, \cdot \rangle_P$, $\|\cdot\|_P$, $\widetilde{\mathrm{d}}(X, \widetilde{H}_{A,P})$, $\widetilde{H}_{A,P}$ as the parallel transport along the geodesic, Riemannian logarithm, Riemannian metric, the induced norm, margin distance and hyperplane over $\widetilde{\mathcal{M}}$, while the counterparts over $\mathcal{M}$ are denoted as $\Gamma$, $\mathrm{Log}$, $\langle \cdot, \cdot \rangle_{\phi(P)}$, $\|\cdot\|_{\phi(P)}$, $\mathrm{d}$, and $H$, respectively.

From the isometry, we have

$$\|A\|_P = \|\phi_{*,P}(A)\|_{\phi(P)}, \tag{108}$$

$$\left\langle \widetilde{\mathrm{Log}}_P(X), A \right\rangle_P = \left\langle \mathrm{Log}_{\phi(P)}(\phi(X)), \phi_{*,P}(A) \right\rangle_{\phi(P)}. \tag{109}$$

The above equations imply

$$\phi\left(\widetilde{H}_{A,P}\right) = H_{\phi_{*,P}(A),\phi(P)}. \tag{110}$$

Denoting $\mathcal{H} = H_{\phi_{*,P}(A),\phi(P)}$, we have the following for the margin distance

$$\begin{aligned}
\widetilde{\mathrm{d}}(X, \widetilde{H}_{A,P}) &= \inf_{Q \in \widetilde{H}_{A,P}} \widetilde{\mathrm{d}}(X, Q) \\
&\overset{(1)}{=} \inf_{Q \in \widetilde{H}_{A,P}} \mathrm{d}(\phi(X), \phi(Q)) \\
&\overset{(2)}{=} \inf_{R \in \mathcal{H}} \mathrm{d}(\phi(X), R) \\
&\overset{(3)}{=} \mathrm{d}(\phi(X), \mathcal{H}).
\end{aligned} \tag{111}$$

The above comes from the following.

(1) Isometry.

(2) Eq. (110).

(3) Definition of margin distance.

Combining the above, we have

$$\begin{aligned}
&v^{\widetilde{\mathcal{M}}}(X; P, A) \\
&= \mathrm{sign}(\langle A, \widetilde{\mathrm{Log}}_P(X)\rangle_P)\|A\|_P \widetilde{\mathrm{d}}(X, \widetilde{H}_{A,P}) \\
&= \mathrm{sign}\left(\left\langle \mathrm{Log}_{\phi(P)}(\phi(X)), \phi_{*,P}(A) \right\rangle_{\phi(P)}\right) \|\phi_{*,P}(A)\|_{\phi(P)} \, \mathrm{d}(X, H_{\phi_{*,P}(A),\phi(P)}) \\
&= v^{\mathcal{M}}(\phi(X); \phi(P), \phi_{*,P}(A)).
\end{aligned} \tag{112}$$

Finally, let us further consider trivialization. By isometry, we have the following:

$$\begin{aligned} A &= \widetilde{\Gamma}_{E \to P}(Z) \\ &= \phi_{*,P}^{-1}\left(\Gamma_{\phi(E) \to \phi(P)}(\phi_{*,E}(Z))\right), \end{aligned} \tag{113}$$

$$\begin{aligned} P &= \widetilde{\mathrm{Exp}}_E(\gamma[Z]) \\ &= \phi^{-1}\left(\mathrm{Exp}_{\phi(E)}(\gamma[\phi_{*,E}(Z)])\right). \end{aligned} \tag{114}$$

Then, we have

$$\phi_{*,P}(A) = \Gamma_{\phi(E) \to \phi(P)}(\phi_{*,E}(Z)), \tag{115}$$

$$\phi(P) = \mathrm{Exp}_{\phi(E)}(\gamma[\phi_{*,E}(Z)]). \tag{116}$$

Putting the above two equations into Eq. (112), we have

$$\begin{aligned} v^{\widetilde{\mathcal{M}}}(X; Z, \gamma) &= v^{\widetilde{\mathcal{M}}}(X; P, A) \\ &= v^{\mathcal{M}}\left(\phi(X); \phi(P), \phi_{*,P}(A)\right) \\ &= v^{\mathcal{M}}\left(\phi(X); \mathrm{Exp}_{\phi(E)}(\gamma[\phi_{*,E}(Z)]), \Gamma_{\phi(E) \to \phi(P)}(\phi_{*,E}(Z))\right) \\ &= v_k^{\mathcal{M}}(\phi(X); \phi_{*,E}(Z_k), \gamma_k). \end{aligned} \tag{117}$$

$\square$

Thm. 3.1 is a special case of Lem. I.1 and can be readily proven accordingly.

*Proof of Thm. 3.1.* **MLR:** In Euclidean space $\mathbb{R}^m$, simple computations show that Eq. (2) becomes Eq. (1), where the latter is equal to $\langle a_k, x - p_k \rangle$. Based on Lem. I.1, we have

$$\begin{aligned} v_k(X; Z_k, \gamma_k) &= v_k^{\mathbb{R}^m}(\phi(X); \phi_{*,E}(Z_k), \gamma_k), \\ &= \langle \phi(X) - \gamma_k[\phi_{*,E}(Z_k)], \phi_{*,E}(Z_k) \rangle \\ &= \langle \phi(X), \phi_{*,E}(Z_k) \rangle - \gamma_k \|\phi_{*,E}(Z_k)\|, \end{aligned} \tag{118}$$

**Margin hyperplane:** In Euclidean space $\mathbb{R}^m$, the Riemannian margin hyperplane becomes the Euclidean one, which is parameterized by $\langle a_k, x - p_k \rangle = 0$. Together with Eq. (118), the results can be easily obtained. $\square$

## I.2. Proof of Prop. 3.2

*Proof.* First, we have the following:

$$\Theta(I) = I, \tag{119}$$

$$\mathrm{Chol}(I) = I, \tag{120}$$

$$\log_{*,I}(V) = V, \quad \forall V \in \mathrm{Hol}(n), \tag{121}$$

$$\log_{*,I}(V) = V, \quad \forall V \in \mathrm{LT}^0(n), \tag{122}$$

$$\mathcal{D}^{\star}(I) = I. \tag{123}$$

Putting the above into Eqs. (26), (30) and (45), one can directly get the result w.r.t. ECM, LECM, and OLM. For LSM, based Eq. (49), we have

$$\begin{aligned} \mathrm{Log}_{*,I}^{\star}(V) &= \log_{*,\Sigma}\left(\Delta V \Delta + \frac{1}{2}\left(V^0 \Sigma + \Sigma V^0\right)\right) \\ &\overset{(1)}{=} V + \frac{1}{2}\left(V^0 + V^0\right) \\ &\overset{(2)}{=} V - \mathrm{diag}\left(V\mathbf{1}\right). \end{aligned} \tag{124}$$

The above comes from the following.

(1) $\Sigma = \Delta = I$

(2)

$$V^0 = -2\operatorname{diag}\left((I_n + \Sigma)^{-1}\Delta V \Delta \mathbf{1}\right)$$
$$= -\operatorname{diag}(V\mathbf{1})$$

(125)

$\square$

### I.3. Proof of Thm. 3.5

Denote $\mathbf{0}_n$ and $\mathbf{0}_m$ as the $n \times n$ and $m \times m$ zero matrices. Let $d_n = \frac{n(n-1)}{2}$ and $d_m = \frac{m(m-1)}{2}$ be the manifold dimensions of $\operatorname{Cor}^+(n)$ and $\operatorname{Cor}^+(m)$, respectively. We have the following general results.

**Lemma I.2.** *Let* $\left(\operatorname{Cor}^+(n), g^n\right)$ *be isometric to* $\mathbb{R}^{d_n}$ *by the diffeomorphism* $\phi : \operatorname{Cor}^+(n) \to \mathbb{R}^{d_n}$, *and* $\left(\operatorname{Cor}^+(m), g^m\right)$ *be isometric to* $\mathbb{R}^{d_m}$ *by the diffeomorphism* $\phi : \operatorname{Cor}^+(m) \to \mathbb{R}^{d_m}$. *The diffeomorphism satisfies* $I_n = (\phi)^{-1}(\mathbf{0}_n)$ *and* $I_m = (\phi)^{-1}(\mathbf{0}_m)$. *The correlation FC layer* $\mathcal{F} : \operatorname{Cor}^+(n) \to \operatorname{Cor}^+(m)$ *for the input* $X \in \operatorname{Cor}^+(n)$ *is*

$$Y = (\phi)^{-1}\left(\sum_{i=1}^{d_m} v_i(X)e_i\right),$$

(126)

*where* $\{e_i\}_{i=1}^{d_m}$ *is the canonical orthonormal basis over* $\mathbb{R}^{d_m}$ *with* $e_i = (\delta_{ik})_{k=1}^{d_m}$ *for each* $i$. *Here,* $\{v_i(X)\}_{i=1}^{d_m}$ *is given by Thm. 3.1:* $v_i(X) = \langle \phi(X), \phi_{*, I_n}(Z_i) \rangle - \gamma_i \|\phi_{*, I_n}(Z_i)\|$, *with* $Z_i \in \mathbb{R}^{d_n}$ *and* $\gamma_i \in \mathbb{R}$ *as the FC parameters.*

*Proof of Lem. I.2.* For simplicity, we use $I$ and $\mathbf{0}$ for the identity and zero matrices. Let $\{O_k = \phi_{*,I}^{-1}(e_k)\}_{i=1}^{d_m}$. Then $\{O_k\}_{i=1}^{d_m}$ is an orthonormal basis over $T_I\operatorname{Cor}^+(m)$.

The LHS of Eq. (9) is

$$\operatorname{sign}\left(\langle \operatorname{Log}_I(Y), O_k \rangle_I\right) \operatorname{d}(Y, H_{O_k, I}) \operatorname{d}(Y, H_{O_k, I})$$
$$\overset{(1)}{=} \operatorname{sign}\left(\langle (\phi_{*,I})^{-1}\phi(Y), O_k \rangle_I\right) \operatorname{d}(Y, H_{O_k, I})$$
$$\overset{(2)}{=} \operatorname{sign}\left(\langle \phi(Y), e_k \rangle\right) \operatorname{d}(Y, H_{O_k, I})$$
$$\overset{(3)}{=} \operatorname{sign}\left(\langle \phi(Y), e_k \rangle\right) \operatorname{d}(\phi(Y), H_{e_k, \mathbf{0}})$$
$$= (\phi(Y))_k,$$

(127)

where (1-2) come from the isometry, and (3) comes from Eq. (111).

The RHS of Eq. (9) can be implied by Thm. 3.1. $\square$

Lem. I.2 can be naturally extended to the cases where the inner products of $\mathbb{R}^{d_n}$ and $\mathbb{R}^{d_m}$ are not canonical.

**Lemma I.3.** *Following all the notation in Lem. I.2, we further assume that the inner products* $Q^n(\cdot, \cdot)$ *over* $\mathbb{R}^{d_n}$ *and* $Q^m(\cdot, \cdot)$ *over* $\mathbb{R}^{d_m}$ *are not necessarily canonical. In addition,* $f : (\mathbb{R}^{d_m}, Q^m(\cdot, \cdot)) \to (\mathbb{R}^{d_m}, \langle \cdot, \cdot \rangle)$ *is a linear isometry to the canonical inner product. Then, we have*

$$Y = \phi^{-1} \circ f^{-1}\left(\sum_{i=1}^{d_m} v_i(X)f^{-1}(e_i)\right),$$

(128)

$$v_i(X) = Q^n\left(\phi(X), \phi_{*, I_n}(Z_i)\right) - \gamma_i \|\phi_{*, I_n}(Z_i)\|^{Q^n},$$

(129)

*where* $\|\cdot\|^{Q^n}$ *is the norm induced by* $Q^n$.

*Proof of Lem. I.3.* First, we denote

$$\psi^m = f \circ \phi : \left(\operatorname{Cor}^+(m), g^m\right) \to (\mathbb{R}^{d_m}, \langle \cdot, \cdot \rangle).$$

(130)

Note that the differential of any linear map between vector spaces is itself. The rest of the proof is identical to that of Lem. I.2. $\square$

$$
\begin{pmatrix}
0 & 0 & 0 & \cdots & 0 \\
l_{21} & 0 & 0 & \cdots & 0 \\
l_{31} & l_{32} & 0 & \cdots & 0 \\
\vdots & \vdots & \vdots & \ddots & \vdots \\
l_{m1} & l_{m2} & l_{m3} & \cdots & 0
\end{pmatrix}
\quad
\begin{pmatrix}
0 & \star & \star & \cdots & \star \\
h_{21} & 0 & \star & \cdots & \star \\
h_{31} & h_{32} & 0 & \cdots & \star \\
\vdots & \vdots & \vdots & \ddots & \star \\
h_{m1} & h_{m2} & h_{m3} & \cdots & 0
\end{pmatrix}
\quad
\begin{pmatrix}
r_{11} & \star & \cdots & \star \\
r_{21} & r_{22} & \cdots & \star \\
\vdots & \vdots & \ddots & \star \\
r_{m-1,1} & r_{m-1,2} & \cdots & \star \\
-\sum_{i=1}^{m-1} r_{i1} & -\sum_{i=1}^{m-1} r_{i2} & \cdots & \sum_{j=1}^{m-1}\sum_{i=1}^{m-1} r_{ij}
\end{pmatrix}
$$

$$L \in \mathrm{LT}^0(m) \qquad\qquad H \in \mathrm{Hol}(m) \qquad\qquad R \in \mathrm{Row}_0(m)$$

*Figure 10.* Illustration of the Euclidean spaces $\mathrm{LT}^0(m)$, $\mathrm{Hol}(m)$ and $\mathrm{Row}_0(m)$, where $\star$ can be obtained by symmetry.

Now, we present the proof of Thm. 3.5.

*Proof of Thm. 3.5.* As ECM, LECM, OLM, and LSM are pullback metrics from Euclidean spaces, we resort to Lem. I.2 and its extension Lem. I.3. Denoting the zero matrix as $\mathbf{0}$, we have the following:

$$\phi^{\mathrm{EC}}(I_n) = \log \circ \Theta(I_n) = \mathbf{0} \in \mathrm{LT}^0(n), \tag{131}$$

$$\mathrm{Log}^\circ(I_n) = \mathbf{0} \in \mathrm{Hol}(n), \tag{132}$$

$$\mathrm{Log}^\star(I_n) = \mathbf{0} \in \mathrm{Row}_0(n). \tag{133}$$

Therefore, the identity matrix is indeed the origin defined in Lem. I.2.

Recalling Lem. I.2, the prototype space is the vector space with the standard vector inner product. Obviously, $\mathrm{LT}^0(m)$, $\mathrm{Hol}(m)$, and $\mathrm{Row}_0(m)$ are linearly isomorphic to $\mathbb{R}^{m(m-1)/2}$. As shown in Fig. 10, each $L \in \mathrm{LT}^0(m)$ can be identified with a vector of its lower triangular part. Besides, $\mathrm{LT}^0(m)$ with the canonical matrix inner product is identified with $\mathbb{R}^{\frac{m(m-1)}{2}}$ with standard vector inner product. Therefore, the basis over $\mathrm{LT}^0(n)$ corresponding to the canonical orthonormal basis over $\mathbb{R}^{m(m-1)/2}$ is

$$(\mathrm{LT}^0(m), \langle \cdot, \cdot \rangle) : U_{ij}^{\mathrm{LT}^0(m)} = E_{ij}, \quad 1 \le j < i \le m, \tag{134}$$

where $E_{ij} \in \mathbb{R}^{m \times m}$ is the standard basis matrix, with the $(k, l)$-th element defined as

$$(E_{ij})_{kl} = \begin{cases} 1 & \text{if } k = i \text{ and } l = j, \\ 0 & \text{otherwise.} \end{cases} \tag{135}$$

Without loss of generality, we identify $(\mathrm{LT}^0(m), \langle \cdot, \cdot \rangle)$ with $(\mathbb{R}^{\frac{m(m-1)}{2}}, \langle \cdot, \cdot \rangle)$, and refer to $\{E_{ij}\}_{1 \le j < i \le m}$ as the canonical orthonormal basis.

However, $\{E_{ij}\}$ is neither a canonical orthonormal basis nor even orthonormal for $\mathrm{Hol}(m)$ and $\mathrm{Row}_0(m)$ under the standard matrix inner product. According to Lem. I.3, we only need to find the linear isometry that maps these two spaces into $(\mathrm{LT}^0(m), \langle \cdot, \cdot \rangle)$. By Fig. 10, we have the following linear isometries to pull back these two inner products to the standard ones over $\mathrm{LT}^0(m)$:

$$
\begin{aligned}
f_{\mathrm{Hol}(m) \to \mathrm{LT}^0(m)} &: (\mathrm{Hol}(m), \langle \cdot, \cdot \rangle) \to (\mathrm{LT}^0(m), \langle \cdot, \cdot \rangle), \\
\mathrm{Hol}(m) &\ni H \longmapsto \sqrt{2} \lfloor H \rfloor \in \mathrm{LT}^0(m), \\
f_{\mathrm{Row}_0(m) \to \mathrm{LT}^0(m)} &: (\mathrm{Row}_0(m), \langle \cdot, \cdot \rangle) \to (\mathrm{LT}^0(m), \langle \cdot, \cdot \rangle), \\
\mathrm{Row}_0(m) &\ni R \longmapsto \sqrt{6} \left\lfloor \widetilde{R} \right\rfloor + \sqrt{3} \mathbb{D}(\widetilde{R}) \in \mathrm{LT}^0(m),
\end{aligned}
\tag{136}
$$

where $\widetilde{R} \in \mathcal{S}^{m-1}$ is the leading principal submatrix of order $m - 1$ of $R$. The bases $f_{\mathrm{Hol}(m) \to \mathrm{LT}^0(m)}^{-1}(\{E_{ij}\})$ and $f_{\mathrm{Row}_0(m) \to \mathrm{LT}^0(m)}^{-1}(\{E_{ij}\})$ are as follows:

$$(\mathrm{Hol}(m), \langle \cdot, \cdot \rangle) : U_{ij}^{\mathrm{Hol}(m)} = \frac{E_{ij} + E_{ji}}{\sqrt{2}}, \quad 1 \le j < i \le m \tag{137}$$

$$(\mathrm{Row}_0(m), \langle \cdot, \cdot \rangle) : U_{ij}^{\mathrm{Row}_0(m)} = \begin{cases} \frac{E_{ii} - E_{in} - E_{ni}}{\sqrt{3}}, & \text{if } 1 \le i < m \\ \frac{E_{ij} + E_{ji} - E_{ni} - E_{in} - E_{nj} - E_{jn}}{\sqrt{6}}, & \text{if } 1 \le j < i < m \end{cases} \tag{138}$$

Putting the required diffeomorphisms and $v_{ij}^g$ in Thm. 3.3 into Lem. I.3 for ECM, LECM, OLM, and LSM, the corresponding FC layers can be readily obtained.

$\square$

### I.4. Proof of Prop. 4.1

*Proof.* First, we review the isometries between the open hemisphere and hyperboloid (Thanwerdas & Pennec, 2022b, Eqs. (4.1-4.2)), and the one between Poincaré ball and hyperboloid (Skopek et al., 2020, Sec. 2.1):

$$\psi_{\mathbb{HS}^n \to \mathbb{H}^n} : (x_1, \ldots, x_{n+1})^\top \in \mathbb{HS}^n \longmapsto \frac{1}{x_{n+1}} (x_1, \ldots, x_n, 1)^\top \in \mathbb{H}^n, \tag{139}$$

$$\psi_{\mathbb{H}^n \to \mathbb{HS}^n} : (y_1, \ldots, y_{n+1})^\top \in \mathbb{H}^n \longmapsto \frac{1}{y_{n+1}} (y_1, \ldots, y_n, 1)^\top \in \mathbb{HS}^n, \tag{140}$$

$$\psi_{\mathbb{H}^n \to \mathbb{P}^n} : (x^T, x_{n+1})^\top \in \mathbb{H}^n \longmapsto \frac{x}{1 + x_{n+1}} \in \mathbb{P}^n, \tag{141}$$

$$\psi_{\mathbb{P}^n \to \mathbb{H}^n} : y \in \mathbb{P}^n \longmapsto \left( \frac{2y^T}{1 - \|y\|^2}, \frac{1 + \|y\|^2}{1 - \|y\|^2} \right)^T = \frac{1}{1 - \|y\|^2} \begin{pmatrix} 2y \\ 1 + \|y\|^2 \end{pmatrix} \in \mathbb{H}^n. \tag{142}$$

For any $(x^\top, x_{n+1})^\top \in \mathbb{HS}^n$ and $y \in \mathbb{P}^n$, we have

$$
\begin{aligned}
\psi_{\mathbb{HS}^n \to \mathbb{P}^n} \left( \begin{pmatrix} x \\ x_{n+1} \end{pmatrix} \right) &= \psi_{\mathbb{H}^n \to \mathbb{P}^n} \circ \psi_{\mathbb{HS}^n \to \mathbb{H}^n} \left( \begin{pmatrix} x \\ x_{n+1} \end{pmatrix} \right) \\
&= \psi_{\mathbb{H}^n \to \mathbb{P}^n} \left( \frac{1}{x_{n+1}} \begin{pmatrix} x \\ 1 \end{pmatrix} \right) \\
&= \frac{x}{x_{n+1}} \frac{1}{1 + \frac{1}{x_{n+1}}} \\
&= \frac{x}{1 + x_{n+1}}
\end{aligned}
\tag{143}
$$

$$
\begin{aligned}
\psi_{\mathbb{P}^n \to \mathbb{HS}^n} (y) &= \psi_{\mathbb{H}^n \to \mathbb{HS}^n} \circ \psi_{\mathbb{P}^n \to \mathbb{H}^n} (y) \\
&= \psi_{\mathbb{P}^n \to \mathbb{H}^n} \left( \frac{1}{1 - \|y\|^2} \begin{pmatrix} 2y \\ 1 + \|y\|^2 \end{pmatrix} \right) \\
&= \frac{1}{1 + \|y\|^2} \begin{pmatrix} 2y \\ 1 - \|y\|^2 \end{pmatrix}
\end{aligned}
\tag{144}
$$

$\square$

### I.5. Proof of Prop. E.1

*Proof.* We denote $D = \mathcal{D}(H)$. By Eq. (47), we have

$$
\begin{aligned}
dY &= dD + dH \\
dD &= -\operatorname{diag}\left( \left(H^0\right)^{-1} \mathbb{D} \left( \exp_{*,Y}(dH) \right) \mathbf{1} \right).
\end{aligned}
\tag{145}
$$

Following Ionescu et al. (2015), we denote the inner product $\langle \cdot, \cdot \rangle$ as $\cdot : \cdot$ for simplicity. By the invariance of differential and

properties of trace (Ionescu et al., 2015, Eqs. 67-72), we have the following:

$$
\begin{aligned}
\frac{\partial l}{\partial Y} : dY &= \frac{\partial l}{\partial Y} : dD + \frac{\partial l}{\partial Y} : dH \\
&= \frac{\partial l}{\partial Y} : -\operatorname{diag}\left((H^0)^{-1} \mathbb{D}\left(\exp_{*,Y}(dH)\right) \mathbf{1}\right) + \frac{\partial l}{\partial Y} : dH \\
&\overset{(1)}{=} \operatorname{tr}\left(-\operatorname{Dv}\left(\frac{\partial l}{\partial Y}\right)^T (H^0)^{-1} \mathbb{D}\left(\exp_{*,Y}(dH)\right) \mathbf{1}\right) + \frac{\partial l}{\partial Y} : dH \\
&\overset{(2)}{=} \operatorname{tr}\left(-\left[\mathbf{1}\operatorname{Dv}\left(\frac{\partial l}{\partial Y}\right)^T (H^0)^{-1}\right] \mathbb{D}\left(\exp_{*,Y}(dH)\right)\right) + \frac{\partial l}{\partial Y} : dH \\
&= -(H^0)^{-1}\operatorname{Dv}\left(\frac{\partial l}{\partial Y}\right)\mathbf{1}^T : \mathbb{D}\left(\exp_{*,Y}(dH)\right) + \frac{\partial l}{\partial Y} : dH \\
&= -\mathbb{D}\left((H^0)^{-1}\operatorname{Dv}\left(\frac{\partial l}{\partial Y}\right)\mathbf{1}^T\right) : \exp_{*,Y}(dH) + \frac{\partial l}{\partial Y} : dH \\
&\overset{(3)}{=} \left[\frac{\partial l}{\partial Y} - \exp_{*,Y}\left(\mathbb{D}\left((H^0)^{-1}\operatorname{Dv}\left(\frac{\partial l}{\partial Y}\right)\mathbf{1}^T\right)\right)\right] : dH \\
&\overset{(4)}{=} \operatorname{off}\left[\frac{\partial l}{\partial Y} - \exp_{*,Y}\left(\mathbb{D}\left((H^0)^{-1}\operatorname{Dv}\left(\frac{\partial l}{\partial Y}\right)\mathbf{1}^T\right)\right)\right] : dH
\end{aligned}
\tag{146}
$$

The above comes from the following.

(1)

$$
A : \operatorname{diag}(b) = \operatorname{Dv}(A) : b, \quad \forall A \in \mathbb{R}^{n\times n}, b \in \mathbb{R}^n, \tag{147}
$$
$$
a : b = a^\top b = \operatorname{tr}(a^\top b), \quad \forall a, b \in \mathbb{R}^n. \tag{148}
$$

(2) Cyclic property of the trace for matrices $A$, $B$, and $C$ of compatible dimensions: $\operatorname{tr}(ABC) = \operatorname{tr}(CAB)$.

(3) For any $A \in R^{n\times n}$ and $S \in \mathcal{S}^n$, by the properties of trace, we have

$$
\begin{aligned}
A : \exp_{*,Y}(S) &= A : U\left(L \odot \left(U^\top S U\right)\right) U^\top \\
&= U\left(L \odot U^\top A U\right) U^\top : S \\
&= \exp_{*,Y}(A) : S.
\end{aligned}
\tag{149}
$$

(4) $H$ has zero diagonal elements.

The invariance of the first-order differential gives

$$
\frac{\partial l}{\partial Y} : dY = \frac{\partial l}{\partial H} : dH. \tag{150}
$$

By the last equation in Eq. (146), we can differentiate $\frac{\partial l}{\partial C}$.

$\square$

## I.6. Proof of Prop. E.2

*Proof.* Denoting $\Sigma = f(C) = \mathcal{D}^\star(C)C\mathcal{D}^\star(C) : \operatorname{Cor}^+(n) \to \operatorname{Row}_1^+(n)$, we have

$$
\operatorname{Log}_{*,C}^\star = \log_{*,\Sigma} \circ f_{*,C}. \tag{151}
$$

Combining with the differential of $\operatorname{Log}^\star$ shown in Eq. (49), we have the following differential equation:

$$
d\Sigma = \Delta dC\Delta - \left(V^0\Sigma + \Sigma V^0\right), \tag{152}
$$

with $V^0 = \operatorname{diag}\left((I_n + \Sigma)^{-1} \Delta dC \Delta \mathbf{1}\right)$. Similar with Prop. E.1, we have the following:

$$
\begin{aligned}
\frac{\partial l}{\partial \Sigma} : d\Sigma &= \frac{\partial l}{\partial \Sigma} : \left(\Delta dC \Delta - \left(V^0 \Sigma + \Sigma V^0\right)\right) \\
&= \left(\Delta \frac{\partial l}{\partial \Sigma} \Delta\right) : dC - \frac{\partial l}{\partial \Sigma} : \left(V^0 \Sigma + \Sigma V^0\right) \\
&= \left(\Delta \frac{\partial l}{\partial \Sigma} \Delta\right) : dC - \left(\frac{\partial l}{\partial \Sigma} \Sigma + \Sigma \frac{\partial l}{\partial \Sigma}\right) : \operatorname{diag}\left((I_n + \Sigma)^{-1} \Delta dC \Delta \mathbf{1}\right) \\
&= \left(\Delta \frac{\partial l}{\partial \Sigma} \Delta\right) : dC - \operatorname{Dv}\left(\frac{\partial l}{\partial \Sigma} \Sigma + \Sigma \frac{\partial l}{\partial \Sigma}\right) : \left((I_n + \Sigma)^{-1} \Delta dC \Delta \mathbf{1}\right) \\
&= \left(\Delta \frac{\partial l}{\partial \Sigma} \Delta\right) : dC - \operatorname{tr}\left(\widetilde{v}^{\top} (I_n + \Sigma)^{-1} \Delta dC \Delta \mathbf{1}\right) \\
&= \left(\Delta \frac{\partial l}{\partial \Sigma} \Delta\right) : dC - \operatorname{tr}\left(\Delta \mathbf{1} \widetilde{v}^{\top} (I_n + \Sigma)^{-1} \Delta dC\right) \\
&= \left(\Delta \frac{\partial l}{\partial \Sigma} \Delta\right) : dC - \left(\Delta (I_n + \Sigma)^{-1} \widetilde{v} \mathbf{1}^{\top} \Delta\right) : dC \\
&= \left(\Delta \frac{\partial l}{\partial \Sigma} \Delta - \Delta (I_n + \Sigma)^{-1} \widetilde{v} \mathbf{1}^{\top} \Delta\right) : dC \\
&= \left(\Delta \left(\frac{\partial l}{\partial \Sigma} - (I + \Sigma)^{-1} \widetilde{v} \mathbf{1}^{\top}\right) \Delta\right) : dC.
\end{aligned}
\tag{153}
$$

By imposing symmetrization, we can obtain the results. $\qquad \square$

### I.7. Proof of Thm. F.1

As $\beta$-splitting is the inverse of $\beta$-concatenation (Shimizu et al., 2021), we only need to show the case w.r.t. $\beta$-concatenation. Besides, it suffices to prove the 2D case, which is shown in the following lemma.

**Lemma I.4.** *Given $x_{ij} \in \mathbb{P}^{n_j}$ with $\{i \in 1, \ldots, N_i\}$ and $\{1, \ldots, N_j\}$, applying the $\beta$-concatenation sequentially 2 times in the order $j \to i$ is equivalent to a single $\beta$-concatenation along all indices simultaneously.*

*Proof.* Denoting $d = n_j \times N_j$ and $v_{ij} = \operatorname{Log}_0(x_{ij})$, we have the following

$$
\begin{aligned}
&\operatorname{Exp}_0\left(\operatorname{concat}_{i=1}^{N_i}\left(\beta_{N_i \times d}\beta_d^{-1}\operatorname{concat}_{j=1}^{N_j}\left(\beta_d\beta_{n_j}^{-1}v_{ij}\right)\right)\right) \\
&= \operatorname{Exp}_0\left(\operatorname{concat}_{i=1,j=1}^{i=N_i,j=N_j}\left(\beta_{N_i \times d}\beta_d^{-1}\beta_d\beta_{n_j}^{-1}v_{ij}\right)\right) \\
&= \operatorname{Exp}_0\left(\operatorname{concat}_{i=1,j=1}^{i=N_i,j=N_j}\left(\beta_{N_i \times d}\beta_{n_j}^{-1}v_{ij}\right)\right).
\end{aligned}
\tag{154}
$$

The last line implies the claim. $\qquad \square$

A special case of the above lemma is where all $n_j$ are identical.

**Corollary I.5.** *Given $x_{ij} \in \mathbb{P}^n$ with $\{i \in 1, \ldots, N_i\}$ and $\{1, \ldots, N_j\}$, applying the $\beta$-concatenation sequentially 2 times in the order $j \to i$ is equivalent to a single $\beta$-concatenation along all indices simultaneously.*

Thm. F.1 can be obtained by Lem. I.4 and Cor. I.5.

