# OpenReview forum: "Riemannian Networks over Full-Rank Correlation Matrices"
_ICML.cc/2026/Conference — ICML 2026 regular_

### Official Review · Reviewer_h5f7 · 2026-03-10

**Soundness:** 3
**Presentation:** 3
**Significance:** 2
**Originality:** 3
**Overall Recommendation:** 5
**Confidence:** 3

**Summary:**

The paper proposes to extend different neural network layers (such as Fully Connected (FC), convolutional layers etc) from the standard Euclidean space to the manifold of full-rank correlation matrices, which are Symmetric Positive Definite (SPD) matrices related to the covariance matrix.
Previously, different Riemannian metrics were introduced in the literature to work with the correlation manifold, some of them relying on classic tools such as the Cholesky decomposition.
The paper formalizes differential geometry tools to extend standard Euclidean layers to Riemannian layers corresponding to those respective Riemannian geometries, including backpropagation, allowing for direct end-to-end learning while using the metric of the manifold. The approach is evaluated on 4 datasets for radar signal classification and human action recognition.

**Compliance With Llm Reviewing Policy:**

Affirmed.

**Final Justification:**

The authors addressed my concerns in the rebuttal.

**Key Questions For Authors:**

Due to the lack of clear motivation for the proposed manifold, I cannot give a higher score than weak accept. I would be interested in a clear and concise motivation for using the proposed approach compared to existing approaches.

**Limitations:**

yes

**Strengths And Weaknesses:**

**Strengths**: The first part of the paper is a set of recipes explaining how to extend standard layers to different types of Riemannian manifolds using differential geometry tools. In themselves, the proposed tools do not seem difficult to formulate, but the list is exhaustive. The second part of the paper is an exhaustive set of experiments of four different datasets, 3 of those are relatively small whereas NTU120 is larger with 114,480 sequences of 120 action classes.

**Weakness**: The paper is interesting overall. However, it is not clear why using the proposed manifold of full-rank correlation matrices is supposed to work better than other types of manifolds (e.g. covariance or even SPD matrices in general). For instance, in the case of hyperbolic networks, the motivation of representing tree-like structures is well-motivated. In this paper, it is not clear why someone would want to prioritize correlation matrices instead of any other representation.

**Soundness**: The submission is technically sound, and the claims are well supported. As mentioned above, the main weakness is the lack of motivation to use the proposed manifold. The methods used are appropriate.

**Presentation**: The submission is clearly written and well structured. The overall narrative is easy to follow, and the work properly positions itself.

**Significance**: It is difficult to evaluate the significance of the paper when the motivation to use the proposed approach is not clear.

**Originality**: The work introduces efficient formulations of layers for different types of correlation manifolds.

---

> ### Author Rebuttal · Authors · 2026-03-31
>
> We thank Reviewer $\textcolor{blue}{h5f7}$ for the careful review and the constructive comments.
>
> ## 1. Motivation
>
> - **Why second-order representations.** Correlation and covariance are meaningful only when second-order structure is informative for the task. Particularly, the SPD matrices have shown success in different application domains [a-c]. However, the correlation matrices are less explored in machine learning.
>
> - **Correlation vs. covariance.** We do not argue that correlation should universally replace covariance. Rather, correlation is especially useful when diagonal variances are overly large or unstable, while pairwise dependence is more informative. App. H.5 shows this on HDM05: diagonal variances have large variation, and the diagonal entries are much larger than the off-diagonal ones. In such cases, covariance modeling can be dominated by fluctuating diagonals and underexploit informative pairwise correlations. Correlation suppresses this nuisance and highlights pairwise dependence. This is consistent with Tabs. 2 and 5, where correlation shows stronger results than SPD baselines on HDM05.
>
> - **Correlation vs. normalized covariance.** Correlation is also not equivalent to simply scalar-normalized covariance. App. H.6 shows that SPD networks with normalized covariance inputs lead to only marginal changes and do not reproduce the gains of correlation networks. This is because scalar normalization still keeps the input as covariance, whereas correlation applies pair-specific normalization and produces standardized correlation coefficients.
>
> - **Network necessity.** Although correlation matrices are SPD, treating them as SPD inputs ignores their intrinsic geometry. Tab. 4 shows that simply replacing covariance with correlation in SPDNet gives mixed results, improving HDM05 but degrading Radar and FPHA. By contrast, Tab. 2 shows that CorNets consistently outperform SPD and Grassmannian baselines. Thus, the gain comes from both correlation inputs and correlation-aware networks.
>
> - **Better efficiency.** Among the five correlation geometries, CorNet-ECM is the most efficient. Particularly, on the relatively large NTU120, CorNet-ECM is the fastest among all SPD and Grassmannian baselines, even faster than the vanilla SPDNet. As shown in Tab. 5, CorMLR is also faster than SPDMLR for raw correlation/covariance classification. On HDM05, SPDMLR-Trivlz (AIM) takes $260.67$ s/epoch and SPDMLR-Trivlz (LEM) takes $3.24$ s/epoch, whereas CorMLR (ECM) and CorMLR (PHCM) take only $3.18$ and $2.73$ s/epoch, respectively.
>
> Tab. D: Representative runtime results from Tabs. 2 and 5.
>
> | Setting | Method | Time (s/epoch) |
> | --- | --- | ---: |
> | NTU120 | GrNet | $50.97$ |
> | NTU120 | SPDNet | $12.77$ |
> | NTU120 | GyroSPD++ | $216.46$ |
> | NTU120 | CorNet-ECM | $12.06$ |
> | NTU120 | CorNet-LECM | $12.68$ |
> | HDM05, single-layer classification | SPDMLR-Trivlz (AIM) | $260.67$ |
> | HDM05, single-layer classification | SPDMLR-Trivlz (LEM) | $3.24$ |
> | HDM05, single-layer classification | CorMLR (ECM) | $3.18$ |
> | HDM05, single-layer classification | CorMLR (PHCM) | $2.73$ |
>
> ## References
> > [a] Deep CNNs Meet Global Covariance Pooling: Better Representation and Generalization
> >
> > [b] Riemannian Geometry-Based EEG Approaches: A Literature Review
> >
> > [c] Efficient Degradation-agnostic Image Restoration via Channel-Wise Functional Decomposition and Manifold Regularization

---

> > ### Author Rebuttal · Reviewer_h5f7 · 2026-04-03
> >
> > I thank the authors for their rebuttal, the motivation became more clear: using the correlation manifold is more useful when the variances are very different. Was it measured experimentally that the datasets on which the proposed correlation manifold works better are those with a larger variance of variances?

---

> > > ### Author Response · Authors · 2026-04-03
> > >
> > > We thank the reviewer for the quick response and follow-up question.
> > >
> > > ## App. H.5 provides empirical evidence for why CorNets gain more on HDM05 than on FPHA
> > >
> > > Yes. On the datasets analyzed in App. H.5, this was measured empirically. The purpose of App. H.5 is precisely to explain why the gain of CorNets over SPD baselines is larger on HDM05 than on FPHA (as shown in Tabs. 2 and 5).
> > >
> > > - **What we measured.**
> > >   - **Coefficient of variation**.  In App. H.5.1, for each covariance matrix $\Sigma \in S _{++}^{n}$, we extract the diagonal vector
> > >   $$
> > >   v = \left( \Sigma _{11}, \ldots, \Sigma _{NN} \right),
> > >   $$
> > >   and compute the coefficient of variation
> > >   $$
> > >   CV = \frac{\mathrm{std}\left( v \right)}{\mathrm{mean}\left( v \right) + \varepsilon},
> > >   $$
> > >   with $\varepsilon = 10^{-8}$, as in Eqs. (89) and (90).
> > >   - **Diagonal-to-off-diagonal magnitude ratio**. In App. H.5.2, we compute the mean diagonal magnitude
> > >   $$
> > >   D = \frac{1}{N} \sum _{i=1}^{N} \left| \Sigma _{ii} \right|,
> > >   $$
> > >   the mean off-diagonal magnitude
> > >   $$
> > >   O = \frac{1}{N \left( N - 1 \right)} \sum _{i \neq j} \left| \Sigma _{ij} \right|,
> > >   $$
> > >   and their ratio
> > >   $$
> > >   R = \frac{D}{O}.
> > >   $$
> > >  Here, $CV$ measures the variability of diagonal variances, while $R$ measures how strongly diagonal terms dominate off-diagonal correlations.
> > >
> > > - **What the measurements show.**
> > >   - The empirical distributions in Figs. 6 and 7 show that $CV$ is mostly between $0.8$ and $2.0$ on FPHA, but typically between $1.0$ and $3.0$ on HDM05.
> > >   - Likewise, Figs. 8 and 9 show that $R$ is mostly between $1.7$ and $3.0$ on FPHA, but typically between $2.0$ and $6.0$ on HDM05.
> > >   - Therefore, on HDM05, covariance representations are more strongly dominated by large and highly fluctuating diagonal terms. This can introduce nuisance noise for SPD networks and weaken the informative off-diagonal correlations. In contrast, correlation normalization suppresses this effect and emphasizes pairwise dependence. This is the empirical reason why CorNets achieve a larger improvement over SPD baselines on HDM05 than on FPHA. Please see App. H.5 for the full discussion.

---

### Official Review · Reviewer_RR5h · 2026-03-12

**Soundness:** 3
**Presentation:** 3
**Significance:** 3
**Originality:** 2
**Overall Recommendation:** 4
**Confidence:** 3

**Summary:**

Full-rank correlation matrix manifolds remain relatively underexplored in the literature. This paper proposes a method for constructing Riemannian neural networks on the manifold of correlation matrices. Specifically, it employs five metric, Euclidean–Cholesky Metric, Log-Euclidean–Cholesky Metric, Poly-Hyperbolic–Cholesky Metric, Off-Log Metric, and Log-Scaled Metric, to extend fundamental neural network layers, including Multinomial Logistic Regression, Fully Connected layers, and convolutional layers, enabling them to operate on these geometric structures. Experimental results on full-rank correlation matrix representations demonstrate the effectiveness of the proposed framework compared with existing approaches based on symmetric positive definite and Grassmannian networks.

**Compliance With Llm Reviewing Policy:**

Affirmed.

**Key Questions For Authors:**

1.A correlation matrix can essentially be viewed as an SPD matrix with an additional normalization constraint (unit diagonal). Since a correlation matrix can be obtained from an SPD matrix through normalization, it would be helpful for the authors to clarify the specific motivation for studying correlation matrices separately. In particular, does this formulation provide any advantages, such as improved robustness or more efficient optimization?

2.The validation of the proposed backpropagation formulas appears somewhat insufficient. The paper derives explicit gradient formulas for the OLM and LSM metrics. However, the experimental does not clearly demonstrate how using these exact gradients compares with PyTorch, which computes gradients automatically for the iterative algorithms. In particular, it remains unclear whether the proposed analytical gradients provide practical benefits in terms of convergence speed, final accuracy, or training stability.

3. Limited analysis across metrics: While multiple metrics are considered, the paper provides limited analysis comparing them, for example in terms of computational complexity, convergence speed, or optimization characteristics.

**Limitations:**

Yes

**Strengths And Weaknesses:**

Strengths

1.The paper presents a solid theoretical foundation with detailed derivations and systematically defines fundamental deep learning layers, MLR, FC, and convolution, on the correlation matrix manifold, which helps bridge the gap in applying deep learning methods to this manifold.

2.The paper develops exact backpropagation algorithms for the OLM and LSM metrics, which mitigate the potential inaccuracies of PyTorch’s automatic differentiation when used with iterative procedures.

Weaknesses

Limited novelty: While the paper extends MLR, FC, and convolutional layers to the correlation matrix manifold, which is a relatively unexplored setting, the five metrics (Euclidean–Cholesky Metric[1], Log-Euclidean–Cholesky Metric[1], Poly-Hyperbolic–Cholesky Metric[1], Off-Log Metric[2], and Log-Scaled Metric[2]) used in the framework are all adopted from prior work and mainly applied in this context. The main methodological contribution appears to be the derivation of exact backpropagation algorithms for the OLM and LSM metrics.

[1] Thanwerdas, Y. and Pennec, X. Theoretically and computationally convenient geometries on full-rank correlation matrices. SIAM Journal on Matrix Analysis and Applications, 43(4):1851–1872, 2022b.

[2] Thanwerdas, Y. Permutation-invariant log-Euclidean geometries on full-rank correlation matrices. SIAM Journal on Matrix Analysis and Applications, 45(2):930–953, 2024.

---

> ### Author Rebuttal · Authors · 2026-03-30
>
> We thank Reviewer $\textcolor{green}{RR5h}$ for thoughtful comments.
>
> ## 1. Novelty
>
> Our main novelty is not new metrics, but a systematic derivation of neural layers on $\mathrm{Cor} ^{+}(n)$ from five existing correlation geometries.
>
> - **Log-Euclidean layers.** For ECM, LECM, OLM, and LSM, we start from the point-to-hyperplane formulation of MLR and solve the corresponding margin-distance problem on $\mathrm{Cor} ^{+}(n)$, yielding unified correlation MLR, FC, and convolutional layers.
>
> - **PHCM layers.** For PHCM, we identify $\mathrm{Cor} ^{+}(n)$ with a poly-Poincaré space and derive the corresponding layers from Poincaré layers.
>
> - **Backpropagation.** For OLM and LSM, we derive analytic BP formulas as accurate alternatives to autograd for these iterative procedures.
>
> ## 2. Motivation
>
> - **Representation motivation.** We do not mean correlation should universally replace covariance. However, correlation emphasizes pairwise dependence and is especially useful when diagonal variances are large or unstable, where covariance can be dominated by diagonal terms. More analysis is presented in App. H.5. App. H.6 further shows that normalized covariance yields only marginal changes and fails to reproduce the gains of CorNets.
>
> - **Network necessity.** Although correlation matrices are SPD, treating them as SPD inputs ignores their intrinsic geometry. Tab. 4 shows that simply replacing covariance with correlation in SPDNet gives mixed results, improving HDM05 but degrading Radar and FPHA. By contrast, Tab. 2 shows that CorNets consistently outperform SPD and Grassmannian baselines. Thus, the gain comes from both correlation inputs and correlation-aware networks.
>
> - **Better efficiency.** CorNet-ECM is the fastest on NTU120, and Tab. 5 shows that CorMLR can also be more efficient than SPD MLR.
>
> ## 3. Analytic BP vs. autograd
>
> When the forward iterative solver converges, analytic BP and autograd are numerically equivalent and mainly differ in runtime. When the solver is truncated, the two choices are no longer equivalent and become a design choice.
>
> - **Efficiency advantage.** When the inner iterative solver converges, analytic BP is consistently faster than autograd. We benchmark the loss $\|X - \phi(X)\| _F ^2$, with $\phi=\mathrm{Exp} ^{\circ}$ for OLM and $\phi=\mathrm{Log} ^{\star}$ for LSM, using a batch size of $10$ and $10$ repeats. The table below reports average forward and backward runtime, showing consistent speedups across all dimensions.
>
> |Metric|n=32|n=64|
> |-|-|-|
> |LSM with autograd|0.337 ± 0.007|0.532 ± 0.006|
> |LSM with analytic BP|0.150 ± 0.003 (↓ 55.5%)|0.226 ± 0.013 (↓ 57.5%)|
> |OLM with autograd|1.092 ± 0.049|3.329 ± 0.070|
> |OLM with analytic BP|0.863 ± 0.038 (↓ 21.0%)|2.687 ± 0.011 (↓ 19.3%)|
>
> - **Setting in CorNet.** For OLM, converged forward computation gives better empirical performance, and the reported results use analytic BP for efficiency. For LSM, one forward iteration works best empirically, and the reported results use one-step PyTorch autograd for efficiency.
>
> - **Optimization behavior.** Under converged forward computation, such as OLM, the two backward strategies show no meaningful difference in convergence speed, training stability, or final accuracy. The main difference appears in one-step LSM. Extending Tab. 3 with mixed-geometry LSM variants on HDM05, we find that autograd gives better final accuracy overall in this truncated regime.
>
> |Geometry|autograd|analytic BP|
> |-|-|-|
> |ECM-LSM|**78.54 ± 0.43**|75.60 ± 0.79|
> |LECM-LSM|73.61 ± 0.99|**74.66 ± 1.04**|
> |PHCM-LSM|**78.28 ± 0.64**|77.61 ± 0.51|
>
> - **Potential.** However, since some methods use a finite-step iterative forward pass with an analytic gradient [a], our analytic BP offers an alternative approach.
>
> ## 4. Metrics comparison
>
> - **Properties.** Tab. 8 summarizes the properties, as recapped below.
>
> |Metric|Properties|
> |-|-|
> |ECM|Null curvature|
> |LECM|Null curvature|
> |OLM|Permutation-invariance, Null curvature|
> |LSM|Permutation-invariance, inverse-consistency, null curvature|
> |PHCM|Nonpositive sectional curvature|
>
> - **Complexity.** Tab. 2 shows that CorNet-ECM is the most efficient one. App. H.8 further studies scalability. The table below shows that ECM remains the fastest, PHCM scales better at high dimensions, and LECM is the slowest because $\log \circ \Theta$ requires two nested matrix functions.
>
> |Dim|ECM|LECM|OLM|LSM|PHCM|
> |-|---:|---:|---:|---:|---:|
> |30|0.0004|0.0018|0.0012|0.0019|0.0131|
> |1000|0.0706|3.7259|2.5390|2.5783|1.2290|
>
> - **Convergence curves.** The curves are shown in the [Anonymous PDF](https://anonymous.4open.science/r/CorNet-4B2E/CorNet-Rebuttal.pdf). We do not observe a clear difference in convergence speed, although LSM is slightly less stable.
>
> - **Consistent vs. mixed geometries.** Tab. 3 provides this ablation. Overall, using a consistent metric across convolution and MLR generally gives the best accuracy.
>
> ## References
> > [a] Fast Differentiable Matrix Square Root and Inverse Square Root

---

> > ### Author Rebuttal · Reviewer_RR5h · 2026-04-03
> >
> > Thanks for your response, which addresses my concerns. I will keep my score.

---

> > > ### Author Response · Authors · 2026-04-03
> > >
> > > Thank you very much for your quick and positive response. We appreciate your recognition of our rebuttal and are grateful for your thoughtful feedback, which helped improve the clarity and overall quality of our work.

---

### Official Review · Reviewer_vmVB · 2026-03-13

**Soundness:** 3
**Presentation:** 3
**Significance:** 3
**Originality:** 4
**Overall Recommendation:** 4
**Confidence:** 5

**Summary:**

This paper studies deep learning on full-rank correlation matrices and consider the correlation manifold as a normalized alternative to the SPD manifold rather than simply reusing existing SPD architectures. The main technical contribution is to extend core neural components, including multinomial logistic regression, fully connected layers, and convolutional layers, to five correlation geometries. The paper also develops accurate backpropagation schemes for OLM and LSM, and builds CorNets on top of these layers. Empirically, the method is evaluated against existing SPD and Grassmannian baselines on radar and action-recognition benchmarks, with the paper arguing that respecting the intrinsic geometry of correlation matrices is preferable to naively feeding correlation inputs into standard SPD networks.

**Compliance With Llm Reviewing Policy:**

Affirmed.

**Final Justification:**

As all my concerns are fully addressed during the rebuttal period with constructive feedback, I will main my original score.

**Key Questions For Authors:**

Since correlation matrices are obtained from SPD matrices through variance normalization, could the authors clarify the precise theoretical relationship between the proposed correlation geometries and standard SPD geometries? In particular, can ECM, LECM, or PHCM be understood as constrained, quotient, or pullback counterparts of familiar SPD metrics such as LCM, LEM, or AIM, and what geometric information is preserved or lost in passing from $SPD(n)$ to $Cor(\Sigma)$?

**Limitations:**

Overall, most aspects of the paper, including the theoretical and empirical results, are quite good. My only real concern is the scale and versatility of the datasets. The particular datasets considered in the paper are relatively small and focused on specific application domains. In addition, since correlation matrices can be viewed as a subgroup-like subset within the broader landscape of SPD matrices and it would also be necessary to investigate whether the framework extends to other important settings such as quantum systems and dynamical systems. Because of this limitation in applicability, I would remain at a weak accept.

**Strengths And Weaknesses:**

The main novelty of the paper lies in elevating the correlation manifold itself to the level of a native architectural object and building a unified neural framework directly over it. Rather than proposing only a single metric-specific construction, the paper develops a family of correlation networks spanning ECM, LECM, OLM, LSM, and PHCM, enabling direct comparisons across latent geometries within a consistent architecture. The work also appears novel in its treatment of correlation MLR, where it aims to solve the margin-distance problem more directly, instead of relying on the relaxed formulation used in prior general-geometry MLR work, thereby making the construction more faithful to the underlying correlation geometry. In addition, the PHCM branch is not merely a routine transfer of SPD machinery, but a more distinctive construction that identifies correlation matrices through Cholesky factors and a poly-Poincare representation, from which correlation MLR, FC, and convolutional layers are built.

---

> ### Author Rebuttal · Authors · 2026-03-30
>
> We thank Reviewer $\textcolor{purple}{vmVB}$ for the encouraging feedback and constructive comments.
>
>
> ## 1. Relation to SPD: Geometry, invariance, and information trade-off
> - **Quotient set.** Full-rank correlation matrices $\mathrm{Cor}^{+}(n)$ are obtained from SPD matrices $\mathcal{S}_{++}^n$ by quotienting out positive diagonal variances:
> $$C=\mathrm{Cor}(\Sigma)=\mathbb{D}(\Sigma)^{-1 / 2} \Sigma \mathbb{D}(\Sigma)^{-1 / 2}.$$
>
> - **Non-quotient Riemannian geometries.** However, the five geometries used in our paper are not direct quotient counterparts of SPD metrics such as AIM, LEM, or LCM, nor simple restrictions of these SPD geometries to the unit-diagonal subset. The geometry most directly related to the SPD quotient viewpoint is the quotient-affine metric [a]. As noted in the introduction, this quotient geometry does not guarantee closed-form Riemannian logarithms and Fréchet means. This is why later work introduced more convenient intrinsic geometries on $\mathrm{Cor}^{+}(n)$ [b, c].
>
> - **Non-SPD pullback geometries.** As summarized in Tab. A, all five geometries are pullbacks from simpler prototype spaces rather than from SPD geometries. ECM, LECM, OLM, and LSM arise from Euclidean prototype spaces, whereas PHCM arises from a product of hyperbolic spaces.
>
> - **Correlation-specific invariances.** OLM and LSM additionally encode invariances that are natural on $\mathrm{Cor}^{+}(n)$ itself [c]. In particular, they are permutation-invariant (App. B.3.2), analogous to affine-invariance on $\mathcal{S}_{++}^n$ in the sense that both arise from the natural congruence action on their respective manifolds. LSM further satisfies inverse-consistency (App. B, Eq. (57)).
>
> - **Information trade-off.** For correlation matrices, the direct influence of diagonal variances is weakened, while pairwise dependence becomes more salient. This can be advantageous when covariance/SPD representations are dominated by large or highly fluctuating variances, which may bias the model away from informative inter-variable relations. In contrast, correlation encourages the model to focus more directly on pairwise correlations. This is the trade-off analyzed in App. H.5.
>
> Tab. A: Summary of correlation geometries. The full table is in Tab. 8.
> | Metric | Prototype space | Diffeomorphism | Properties |
> | --- | --- | --- | --- |
> | ECM | $\mathrm{LT}^{1}(n)$ | $\Theta : \mathrm{Cor}^{+}(n) \to \mathrm{LT}^{1}(n)$ | Null curvature |
> | LECM | $\mathrm{LT}^{0}(n)$ | $\log \circ \Theta : \mathrm{Cor}^{+}(n) \to \mathrm{LT}^{0}(n)$ | Null curvature |
> | OLM | $\mathrm{Hol}(n)$ | $\mathrm{Log}^{\circ} : \mathrm{Cor}^{+}(n) \to \mathrm{Hol}(n)$ | Permutation-invariance, null curvature |
> | LSM | $\mathrm{Row}_{0}(n)$ | $\mathrm{Log}^{\star} : \mathrm{Cor}^{+}(n) \to \mathrm{Row}_{0}(n)$ | Permutation-invariance, inverse-consistency, null curvature |
> | PHCM | $\mathbb{PH} S^{n-1}$ | $\mathrm{Chol} : \mathrm{Cor}^{+}(n) \to \mathbb{PH} S^{n-1}$ | Nonpositive sectional curvature |
>
> ## 2. Scale and versatility
> - **Scale.** The main paper includes NTU120, a representative large-scale benchmark. App. H.8 further studies computational scalability and shows that ECM is the most efficient.
> - **Versatility.** To further test versatility beyond radar and action recognition, we additionally followed SPDGCN [d] to conduct experiments on graph node classification benchmarks. We replaced only its final SPD classifier with CorMLR. For a fair comparison, we kept the backbone and all training hyperparameters unchanged across all experiments, including the learning rate, weight decay, and dropout. As shown in Tab. B, CorMLR often improved performance over the original SPDGCN.
> - **Future scope.** We agree that quantum systems and dynamical systems are meaningful future directions, and we will further investigate them in future work.
>
> Tab. B: 5-fold accuracy on node classification benchmarks.
> | Model | Disease | Airport | PubMed |
> | --- | --- | --- | --- |
> | SPDGCN | 94.64 ± 2.43 | 54.09 ± 2.20 | 75.87 ± 3.32 |
> | SPDGCN + CorMLR (ECM) | **96.85 ± 0.63** | 63.48 ± 2.20 | 76.66 ± 1.02 |
> | SPDGCN + CorMLR (LECM) | 96.07 ± 2.62 | 63.36 ± 1.52 | 76.70 ± 1.71 |
> | SPDGCN + CorMLR (OLM) | 93.01 ± 3.70 | 65.14 ± 2.69 | 75.82 ± 1.80 |
> | SPDGCN + CorMLR (LSM) | 95.47 ± 2.01 | 64.82 ± 1.64 | **78.48 ± 0.73** |
> | SPDGCN + CorMLR (PHCM) | 92.32 ± 3.35 | **65.73 ± 1.05** | 76.54 ± 1.26 |
>
> ## References
> > [a] A Riemannian Quotient Structure for Correlation Matrices with Applications to Data Science
> >
> > [b] Theoretically and Computationally Convenient Geometries on Full-Rank Correlation Matrices
> >
> > [c] Permutation-Invariant Log-Euclidean Geometries on Full-Rank Correlation Matrices
> >
> > [d] Modeling Graphs Beyond Hyperbolic: Graph Neural Networks in Symmetric Positive Definite Matrices

---

> > ### Author Rebuttal · Reviewer_vmVB · 2026-04-03
> >
> > All of my concerns are properly addressed. Thank you.

---

> > > ### Author Response · Authors · 2026-04-03
> > >
> > > Thank you for your acknowledgement and for taking the time to review our submission. We sincerely appreciate your effort and consideration.

---

### Decision · Program_Chairs · 2026-04-30

**Decision:**

Accept (regular)

**Comment:**

The paper proposes Riemannian networks over the manifold of full-rank correlation matrices, using five recently developed  geometries on this manifold. Numerical experiments comparing the proposed approach against existing SPD and Grassmannian networks are presented to demonstrate its effectiveness.

The scores are Weak Accept, Weak Accept, and Accept.

Reviewers generally appreciate the proposed networks using correlation matrices. On the tested datasets, the correlation networks lead to improvements over the SPD and Grassmannian networks.

I do share the concern by Reviewer RR5h that  the novelty of the work is limited, since all the metrics for correlation matrices were developed in prior work and are adapted for this setting. As pointed out by Reviewer  vmVB, most datasets being tested are quite small with the exception of NTU120. The authors did not present the state of the art for these datasets and nor provide a motivation for why correlation matrix representation should be used (Reviewer RR5h asked about correlation vs covariance, but there is also the bigger question of whether these approaches provide state of the art results on these datasets).